# Cancer cell plasticity defines response to immunotherapy in cutaneous squamous cell carcinoma

Laura Lorenzo-Sanz ⬛[1,13] ✉, Marta Lopez-Cerda ⬛[1], Victoria da Silva-Diz[1,12], Marta H. Artés[1], Sandra Llop[2], Rosa M. Penin[3], Josep Oriol Bermejo[4], Eva Gonzalez-Suarez ⬛[1,5], Manel Esteller ⬛[6,7,8,9], Francesc Viñals[1,9,10], Enrique Espinosa[7,11], Marc Oliva[2], Josep M. Piulats ⬛[1,2], Juan Martin-Liberal[2] & Purificación Muñoz ⬛[1,13] ✉

Immune checkpoint blockade (ICB) approaches have changed the therapeutic landscape for many tumor types. However, half of cutaneous squamous cell carcinoma (cSCC) patients remain unresponsive or develop resistance. Here, we show that, during cSCC progression in male mice, cancer cells acquire epithelial/mesenchymal plasticity and change their immune checkpoint (IC) ligand profile according to their features, dictating the IC pathways involved in immune evasion. Epithelial cancer cells, through the PD-1/PD-L1 pathway, and mesenchymal cancer cells, through the CTLA-4/CD80 and TIGIT/CD155 pathways, differentially block antitumor immune responses and determine the response to ICB therapies. Accordingly, the anti-PD-L1/TIGIT combination is the most effective strategy for blocking the growth of cSCCs that contain both epithelial and mesenchymal cancer cells. The expression of E-cadherin/Vimentin/CD80/CD155 proteins in cSCC, HNSCC and melanoma patient samples predicts response to anti-PD-1/PD-L1 therapy. Collectively, our findings indicate that the selection of ICB therapies should take into account the epithelial/mesenchymal features of cancer cells.

Cutaneous squamous cell carcinoma (cSCC) is the second most common skin cancer[1]. The most important risk factor for cSCC is chronic exposure to ultraviolet radiation, which leads to a high tumor mutational burden (TMB)[2]. Immunosuppression derived from solid-organ transplants or immunosuppressive therapies, as well as infection with high-risk human papillomavirus variants, elevate the risk of cSCC development by over 100-fold[1]. These observations suggest that immune surveillance is critical for preventing cSCC development.

Most cSCC patients are successfully treated by surgical excision, but a subset of patients develop recurrent and aggressive tumors,

[1]Oncobell Program, Bellvitge Biomedical Research Institute (IDIBELL), 08908 L'Hospitalet de Llobregat, Barcelona, Spain. [2]Medical Oncology Department, Catalan Institute of Oncology (ICO), 08908 L'Hospitalet de Llobregat, Barcelona, Spain. [3]Pathology Service, Bellvitge University Hospital/IDIBELL, 08908 L'Hospitalet de Llobregat, Barcelona, Spain. [4]Plastic Surgery Unit, Bellvitge University Hospital/IDIBELL, 08908 L'Hospitalet de Llobregat, Barcelona, Spain. [5]Molecular Oncology, Spanish National Cancer Research Centre (CNIO), 28029 Madrid, Spain. [6]Josep Carreras Leukaemia Research Institute (IJC), 08916 Badalona, Barcelona, Spain. [7]Centro de Investigación Biomédica en Red Cáncer (CIBERONC), ISCIII, 28029 Madrid, Spain. [8]Institució Catalana de Recerca i Estudis Avançats (ICREA), 08010 Barcelona, Spain. [9]Physiological Sciences Department, School of Medicine and Health Sciences, University of Barcelona (UB), 08908 Barcelona, Spain. [10]Program Against Cancer Therapeutic Resistance (ProCURE), Catalan Institute of Oncology (ICO)/IDIBELL, 08908 L'Hospitalet de Llobregat, Barcelona, Spain. [11]Medical Oncology Department, La Paz University Hospital, Autonomous University of Madrid (UAM), 28046 Madrid, Spain. [12]Present address: Rutgers Cancer Institute of New Jersey, Rutgers University, 08901 New Brunswick, NJ, USA. [13]These authors jointly supervised this work: Laura Lorenzo-Sanz, Purificación Muñoz. ✉e-mail: llorenzo@idibell.cat; p.munoz@idibell.cat

which have poor clinical outcomes[3]. Early cSCCs conserve epithelial differentiation features and are considered well or moderately differentiated (WD/MD-SCCs, G2 grade); however, advanced cSCCs lose epithelial differentiation traits, acquire poorly differentiated/mesenchymal features, and eventually become spindle-shaped (PD-SCCs and PD/S-SCCs; G3-G4 grade). These advanced cSCCs have a high risk of relapse and metastasis[4], and are commonly treated with radiotherapy and/or chemotherapy, which bestow limited clinical benefits[5]. Immunotherapy based on two programmed cell death protein 1 (PD-1) inhibitors (cemiplimab and pembrolizumab) has also been approved for the treatment of locally advanced and metastatic cSCCs. However, half of cSCC patients do not achieve tumor response or develop acquired resistance[6,7]. For this reason, determining the predictors of response to immune checkpoint inhibitors (ICIs) could help identify which cSCC patients will benefit from these therapies, which is why research on biomarkers is a clinical priority.

Our previous studies demonstrated that early mouse WD-SCCs retained epithelial differentiation traits. After long-term tumor growth, WD-SCCs progressed to MD/PD-SCCs containing well and poorly differentiated regions. Subsequent growth of these MD/PD-SCCs gave rise to PD/S-SCCs with poorly differentiated/mesenchymal features[8]. These findings proved that mouse cSCC progression is associated with an induction of the epithelial–mesenchymal transition (EMT) program, as described in other works[9–14], and that PD/S-SCCs are generated through the malignant progression of WD-SCCs. Furthermore, cancer cell features greatly affected EGFR-, PDGFR- and FGFR-targeted-therapy response because the mechanisms controlling cell proliferation, survival, and dissemination differed according to the epithelial or mesenchymal features of cancer cells[8,15,16]. Similar changes in cancer cell features and signaling pathways controlling cSCC growth were observed in early and advanced/high-risk cSCC patient samples[8,15,16], highlighting the relevance of our mouse cSCC model for evaluating the response to different therapies.

Several studies have indicated that cancer cell features also influence the tumor immune landscape[13,17–19] and that, in turn, these immune cells play an important role in tumor progression, invasion, and metastasis[20–22]. Indeed, tumors enriched in the EMT, focal adhesion, extracellular matrix remodeling, angiogenesis, inflammation, and hypoxia signatures are associated with increased infiltration of immunosuppressive cell populations (such as myeloid-derived suppressor cells (MDSCs), regulatory T (Treg) cells, and M2-like macrophages)[23,24] and expression of multiple inhibitory molecules such as immune checkpoint (IC) ligands[25,26]. Other reports have described the importance of epithelial–mesenchymal plasticity (EMP) and the progression through different hybrid epithelial/mesenchymal (E/M) states for modulating the immune system to support tumor heterogeneity, immune evasion, and therapy resistance[12,27–32], but it is unknown whether this diversity of E/M states could determine variable immune evasion responses during cSCC progression. To overcome this immune suppression, several IC-blocking antibodies have been developed to boost antitumor responses against cancer cells[33,34]. However, despite recent advances in the investigation of predictive biomarkers of response to anti-PD-1/PD-L1 therapies, neither these biomarkers nor the mechanisms involved in anti-PD-1/PD-L1 resistance, have yet been clearly identified.

Here, we show that cancer cells acquire E/M plasticity and change their IC ligand profile during mouse cSCC progression. Mouse epithelial and mesenchymal cancer cells inhibit the antitumor response of cytotoxic effector immune cells by using different IC pathways, revealing the critical role that cancer cell plasticity plays in defining the response to ICB therapies. Given that the diversity of cancer cell states and IC ligand expression in cSCC patient samples is similar to that described in mouse cSCCs, our results indicate that similar IC pathways may be involved in immune evasion in patient cSCCs. In addition, analysis of E-cadherin, Vimentin, CD80, and CD155 proteins in cSCC

patient samples shows their value as predictive biomarkers of anti-PD-1/PD-L1 response, which has also been corroborated in head and neck squamous cell carcinoma (HNSCC) and melanoma patient samples. Overall, our findings reveal that personalized ICB therapies should be selected on the basis of cancer cell features.

## Results

### Cancer cells acquire E/M plasticity during mouse cSCC progression

To characterize the dynamic changes of cancer cell features during mouse cSCC progression, we compared the α6-integrin+CD45− cancer cell compartment at different cSCC stages, which had previously been generated by orthotopic serial engraftments[8]. We observed that WD-SCCs were formed of epithelial α6-integrin+EpCAM+ cancer cells, MD/PD-SCCs contained a mixture of epithelial α6-integrin+EpCAM+ and mesenchymal α6-integrin+EpCAM− cancer cells, and PD/S-SCCs contained only mesenchymal α6-integrin+EpCAM− cancer cells (Fig. 1a). To investigate the possible plastic nature of these cancer cell populations, we isolated α6-integrin+EpCAM+ cancer cells from WD-SCCs and MD/PD-SCCs, and α6-integrin+EpCAM− cancer cells from MD/PD-SCCs, and stably transduced them with GFP. The engraftment of GFP+EpCAM+ cancer cells from WD-SCCs into immunocompetent syngeneic mice gave rise to tumors with a reduced percentage of GFP+EpCAM− cancer cells (Fig. 1b). In contrast, the engraftment of GFP+EpCAM+ cancer cells from MD/PD-SCCs gave rise to tumors with a variable percentage of GFP+EpCAM− cancer cells, and GFP+EpCAM− cancer cells from MD/PD-SCCs exclusively generated PD/S-SCCs (Fig. 1b). Interestingly, not all epithelial EpCAM+ cancer cells from MD/PD-SCCs switched to the mesenchymal state during cSCC growth, suggesting that this population is heterogeneous. EpCAM+ cancer cells from MD/PD-SCCs showed variable levels of EpCAM expression and were classified as EpCAM^high and EpCAM^low; by contrast, cancer cells from WD-SCCs exhibited a high level of EpCAM expression and were named full epithelial cancer cells (Fig. 1c and Supplementary Fig. 2). Their molecular characterization revealed that EpCAM^high cancer cells from MD/PD-SCCs expressed epithelial differentiation genes in a similar fashion to full epithelial cancer cells (Fig. 1d), although with induced expression of *Vim* and *Twist* compared with full epithelial cancer cells (Fig. 1e). Conversely, EpCAM^low cancer cells from MD/PD-SCCs did not show significant changes in the expression of *Krt14*, *Grhl2*, *dNp63*, *Ovol1* and *Ovol2* epithelial genes compared with full epithelial cancer cells, but exhibited diminished expression of *Cdh1*, *Grhl1* and *Epcam*, and upregulated the expression of *Vim* and several EMT transcription factors (TFs) to similar levels to those in mesenchymal EpCAM− cancer cells (Fig. 1d, e). These EpCAM− cancer cells from MD/PD-SCCs and full mesenchymal cancer cells from PD/S-SCCs showed downregulation of the expression of epithelial genes and marked upregulation of mesenchymal genes, revealing the mesenchymal nature of these populations (Fig. 1c−e).

We next compared the plasticity of these mouse cSCC cancer cells after engrafting them into immunocompetent syngeneic mice. Full epithelial cancer cells gave rise to WD-SCCs that contained mostly EpCAM^high cancer cells, whereas EpCAM− cell-derived tumors were comprised exclusively of EpCAM− cancer cells (Fig. 1f). It is of note that while EpCAM^high cancer cells showed a moderate ability to generate EpCAM^low and EpCAM− cancer cells, EpCAM^low cell-derived tumors contained mostly EpCAM− cancer cells (Fig. 1f). These findings indicate that hybrid E/M cancer cells with different plastic behavior appear at intermediate stages of mouse cSCC progression and that these differ from full epithelial cancer cells by their co-expression of epithelial and mesenchymal genes, and by their greater ability to switch into the mesenchymal state.

Finally, we characterized the presence of epithelial, hybrid E/M, and mesenchymal cancer cells by analyzing the expression of E-cadherin (Ecad, epithelial marker) and Vimentin (Vim, mesenchymal marker) in a subset of cSCC patient samples with different histopathological grades (G2-grade including MD-SCCs; G3-grade including

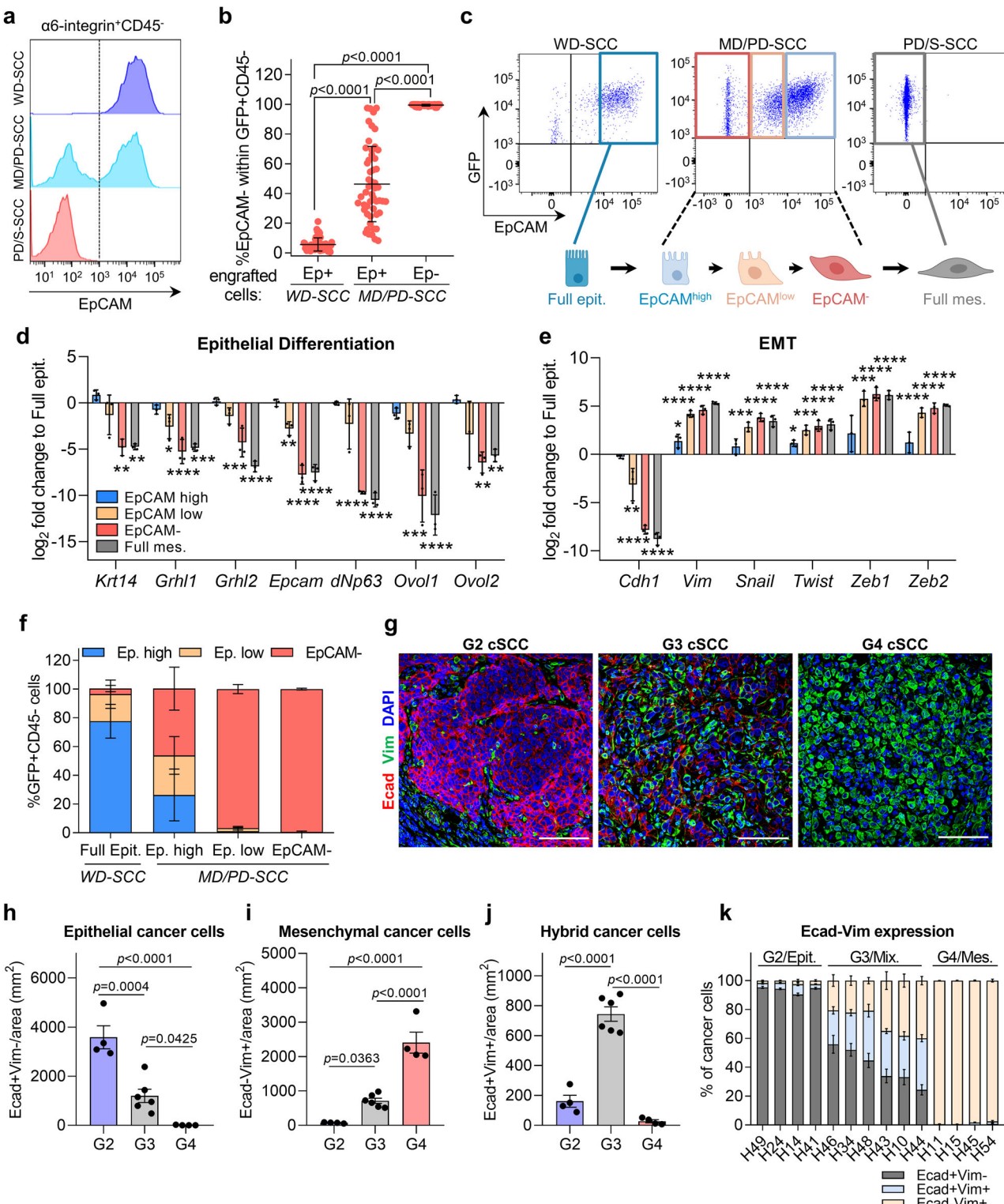

MD/PD-SCCs and PD-SCCs; G4-grade including PD/S-SCCs) (Fig. 1g–k). The frequency of epithelial Ecad⁺Vim⁻ cancer cells was lower in G3 cSCCs than in G2 cSCCs, and these cells were completely absent in G4 cSCCs (Fig. 1h). In addition, while Vimentin expression was most frequently detected in stromal cells of G2 cSCCs (Fig. 1g), it was also expressed in mesenchymal Ecad⁻Vim⁺ cancer cells of G3 cSCCs and, at high level, in G4 cSCC cancer cells (Fig. 1i). A population of hybrid Ecad⁺Vim⁺ cancer cells was specifically more abundant in G3 cSCCs (Fig. 1j), coinciding with the emerging hybrid E/M cancer cells identified at intermediate stages of mouse cSCC progression (Fig. 1d, e).

Taken together, our results demonstrate that early G2 cSCCs are mostly formed of epithelial Ecad⁺Vim⁻ cancer cells, G3 cSCCs contain epithelial, hybrid E/M, and mesenchymal cancer cells, and advanced/high-risk G4 cSCCs are mostly composed of mesenchymal Ecad⁻Vim⁺ cancer cells, allowing us to classify them more accurately according to their cancer cell features into epithelial, mixed, and mesenchymal cSCCs beyond their histopathological grade (Fig. 1k). Furthermore, these data highlight the utility of our mouse cSCC model to reproduce the epithelial, hybrid E/M, and mesenchymal cancer cell features found in cSCC patient samples.

**Fig. 1 | Epithelial, hybrid E/M, and mesenchymal cancer cells are detected in mouse and patient cSCCs. a** Representative flow cytometry profile of α6-integrin$^+$CD45$^-$EpCAM$^+$ and EpCAM$^-$ cancer cells within WD-SCCs, MD/PD-SCCs, and PD/S-SCCs, which had previously been generated by orthotopic serial engraftments[8]. **b** Percentage of mesenchymal GFP$^+$CD45$^-$EpCAM$^-$ cancer cells generated after engrafting the indicated GFP$^+$ cancer cells into immunocompetent syngeneic mice ($n$ = 52 tumors per group). **c** Flow cytometry strategy to isolate full epithelial, EpCAM$^{high}$, EpCAM$^{low}$, EpCAM$^-$ and full mesenchymal cancer cells from WD-SCCs, MD/PD-SCCs, and PD/S-SCCs based on EpCAM expression. **d, e** mRNA expression levels of **d** epithelial differentiation and **e** EMT genes in the indicated cancer cells relative to full epithelial cancer cells ($n$ = 3 biologically independent samples per group). *dNp63, Zeb2* (****$p$ < 0.0001), *Krt14* (**$p$ = 0.0010, **$p$ = 0.0011), *Grhl1* (*$p$ = 0.0156, ****$p$ < 0.0001, ***$p$ = 0.0002), *Grhl2* (***$p$ = 0.0005, ****$p$ < 0.0001), *Epcam* (**$p$ = 0.0030, ****$p$ < 0.0001), *Ovol1* (***$p$ = 0.0001, ****$p$ < 0.0001), *Ovol2* (**$p$ = 0.0038, **$p$ = 0.0087), *Cdh1* (**$p$ = 0.0034, ****$p$ < 0.0001), *Vim* (*$p$ = 0.0161, ****$p$ < 0.0001), *Snail* (***$p$ = 0.0004, ****$p$ < 0.0001), *Twist* (*$p$ = 0.0333, ***$p$ = 0.0002, ****$p$ < 0.0001), *Zeb1* (***$p$ = 0.0002, ****$p$ < 0.0001). **f** Percentage of EpCAM$^{high}$, EpCAM$^{low}$, and EpCAM$^-$ cancer cells within tumors generated after engrafting full epithelial ($n$ = 41), EpCAM$^{high}$ ($n$ = 32), EpCAM$^{low}$ ($n$ = 10), and EpCAM$^-$ ($n$ = 49) cancer cells. **g** Representative immunofluorescence images of Ecad$^+$ (red), Vim$^+$ (green), and DAPI nuclear (blue) staining in G2, G3, and G4 patient cSCCs. Scale bar, 100 μm. **h–j** Quantification of **h** epithelial Ecad$^+$Vim$^-$, **i** mesenchymal Ecad$^-$Vim$^+$, and **j** hybrid Ecad$^+$Vim$^+$ cancer cells per tumor area (mm$^2$) in G2 ($n$ = 4), G3 ($n$ = 6), and G4 ($n$ = 4) patient cSCCs. Each dot represents the average quantification of at least seven fields from different tumor regions. **k** Percentage of Ecad$^+$Vim$^-$, Ecad$^+$Vim$^+$, and Ecad$^-$Vim$^+$ cancer cells relative to total cancer cells in epithelial ($n$ = 4), mixed ($n$ = 6), and mesenchymal ($n$ = 4) patient cSCCs. Data are represented as the mean ± SD (**b, d–f**) or ± SEM (**h–k**), and $n$ values indicate independent tumors (**b, f, h–k**). $P$ values are determined by one-way ANOVA with Tukey's (**b, h–j**) or Dunnett's (**d, e**) multiple comparison tests. See Supplementary Fig. 2 for the gating strategy (**b, f**). Source data are provided as a Source Data file.

## Cancer cell features affect TME composition in mouse and patient cSCCs

To determine the impact of cancer cell features on tumor microenvironment (TME) composition, we characterized the immunophenotype during mouse cSCC progression. Epithelial cSCCs (>70% EpCAM$^+$ cancer cells), mixed cSCCs (10–70% EpCAM$^+$ cancer cells), and mesenchymal cSCCs (<10% EpCAM$^+$ cancer cells) were generated after engrafting full epithelial cancer cells from WD-SCCs, and EpCAM$^+$ (including EpCAM$^{high}$ and EpCAM$^{low}$ cells) and EpCAM$^-$ cancer cells from MD/PD-SCCs, respectively, into immunocompetent syngeneic mice (Supplementary Fig. 1a). Immune cell analyses revealed that mixed and mesenchymal cSCCs contained a higher infiltrate of T lymphocytes, CD8$^+$ cytotoxic T lymphocytes (CTLs), and NK cells than epithelial cSCCs (Supplementary Figs. 1b–d, 2 and 3a, b). However, the percentage of CTLs and NK cells co-expressing PD-1 along with LAG-3, TIM-3, CTLA-4, and TIGIT increased during mouse cSCC progression (Supplementary Figs. 1e–h, m–o and 2). Given that T- and NK-cell exhaustion is associated with increased co-expression of these IC receptors[35,36], our data indicated that mixed and mesenchymal cSCCs had more exhausted CTLs and NK cells than epithelial cSCCs. Accordingly, the expression of co-stimulatory receptors such as CD28 and CD226, and cytotoxic markers like GzmB and IFN-γ significantly decreased in both effector immune cells during mouse cSCC progression (Supplementary Figs. 1i–l, p–r and 2). We also detected that epithelial and mixed cSCCs presented greater Gr1$^+$ MDSC infiltration than mesenchymal cSCCs, whereas mesenchymal cSCCs contained more CD68$^+$ macrophages (Supplementary Fig. 3c–f). MDSC characterization showed that while Ly6C$^{lo}$Ly6G$^+$ PMN-MDSCs were the predominant MDSC population in epithelial and mixed cSCCs, similar percentages of PMN-MDSCs and Ly6C$^{hi}$Ly6G$^-$ M-MDSCs were present in mesenchymal cSCCs (Supplementary Figs. 1s, t and 2). In addition, most macrophages detected in epithelial and mixed cSCCs were CD206$^-$ M1-like macrophages, whereas CD206$^+$ or CD163$^+$ M2-like macrophages and FoxP3$^+$ Treg cells increased during mouse cSCC progression (Supplementary Figs. 1u, v, 2 and 3g–j).

The molecular characterization of M-MDSCs, M1-like macrophages, and M2-like macrophages isolated from epithelial and mesenchymal cSCCs revealed that M-MDSCs from mesenchymal cSCCs were more immunosuppressive than M-MDSCs from epithelial cSCCs since they showed higher expression of *Ptgs2, Pdl1, Il10,* and *Nos2* (Supplementary Fig. 1w), which would enable them to prevent the cytotoxic effects of CTLs and NK cells[37,38]. On the other hand, M1-like macrophages from mesenchymal cSCCs downregulated the expression of several pro-inflammatory cytokines (*Il23a, Tnfa, Il12b,* and *Il18*) and of some Th1 cell-attracting chemokines (*Cxcl9* and *Cxcl10*), and upregulated some M2/M-MDSC-attracting cytokines, such as *Il6* and *Ccl2* (Supplementary Fig. 1x). These data suggest that M1-like macrophages from mesenchymal cSCCs have less pro-inflammatory and tumoricidal functions than those isolated from epithelial cSCCs. M2-like macrophages from mesenchymal cSCCs showed a marked immunosuppressive signature characterized by an increased expression of immunosuppressive molecules (*Fizz1, Tgfb, Il10, Nos2, Arg1,* and *Gas6*), the pro-angiogenic factor *Vegfa*, and one Treg cell-attracting cytokine as *Ccl22* (Supplementary Fig. 1y), all of which may facilitate tumor progression[39,40].

We corroborated in cSCC patient samples that the frequency of GzmB$^+$ cells decreased and the frequencies of CD8$^+$PD-1$^+$, CD8$^+$LAG-3$^+$, CD8$^+$TIM-3$^+$ and CD8$^+$TIGIT$^+$ cells increased in mixed and mesenchymal patient cSCCs (Supplementary Fig. 4a–j), which are those tumors enriched in hybrid E/M and mesenchymal cancer cells (Fig. 1k). In addition, the infiltration of FoxP3$^+$ Treg cells and CD163$^+$ M2-like macrophages was greater in mixed than in epithelial patient cSCCs, and was even higher in mesenchymal patient cSCCs (Supplementary Fig. 4k–n). Overall, our data demonstrate that cancer cell features affect the frequency of tumor-infiltrating immune cells in mouse and patient cSCCs. cSCCs enriched in hybrid E/M and mesenchymal cancer cells show an increased frequency of immunosuppressive and exhausted immune cells, highlighting the importance of determining which immune evasion mechanisms contribute to the aggressive growth and enhanced metastasis observed in mice bearing mesenchymal cSCCs or in patients with advanced/high-risk cSCCs.

## Immune checkpoint ligand repertoire differs according to the epithelial, hybrid E/M, or mesenchymal features of cancer cells

To study the mechanisms involved in immune evasion, we examined whether mouse cSCC cancer cells might evade immune responses by expressing different IC ligands (Supplementary Fig. 2). Flow cytometry analysis indicated that PD-L1 and CD112 were the most expressed IC ligands by full epithelial and EpCAM$^{high}$ cancer cells, followed by CD80, Gal9, and CD155 (Fig. 2a–e). In contrast, EpCAM$^{low}$, EpCAM$^-$ and full mesenchymal cancer cells downregulated the expression of PD-L1, CD112, and Gal9, and upregulated the expression of CD80 and CD155, ligands of CTLA-4 and TIGIT, respectively (Fig. 2a–e). Interestingly, mixed and mesenchymal patient cSCCs recapitulated the IC ligand alterations described during mouse cSCC progression. The frequency of CD80$^+$ and CD155$^+$ cancer cells increased in mixed and even more so in mesenchymal patient cSCCs (Fig. 2f–i), being in the latter tumors where CD80 and CD155 expression was mainly observed in cancer cells that lost E-cadherin expression (Fig. 2j, k). The frequency of CD80$^+$Ecad$^+$ and CD155$^+$Ecad$^+$ cancer cells specifically increased in mixed patient cSCCs (Fig. 2j, k), suggesting that the upregulation of these IC ligands is associated with the enrichment of hybrid E/M cancer cells at intermediate cSCC stages. Collectively, our mouse and patient data reveal that IC ligand expression changes according to the epithelial, hybrid E/M, and mesenchymal features of cancer cells.

To further explore the relevance of cancer cell IC ligands in regulating CTL functions in the absence of additional stromal components, we isolated CD8[+] T cells from the spleens of mice bearing epithelial or mesenchymal cSCCs and activated them in vitro with CD3/CD28 antibodies in the presence of epithelial or mesenchymal cancer cells (Fig. 3a). We confirmed that the expression profile of IC ligands in

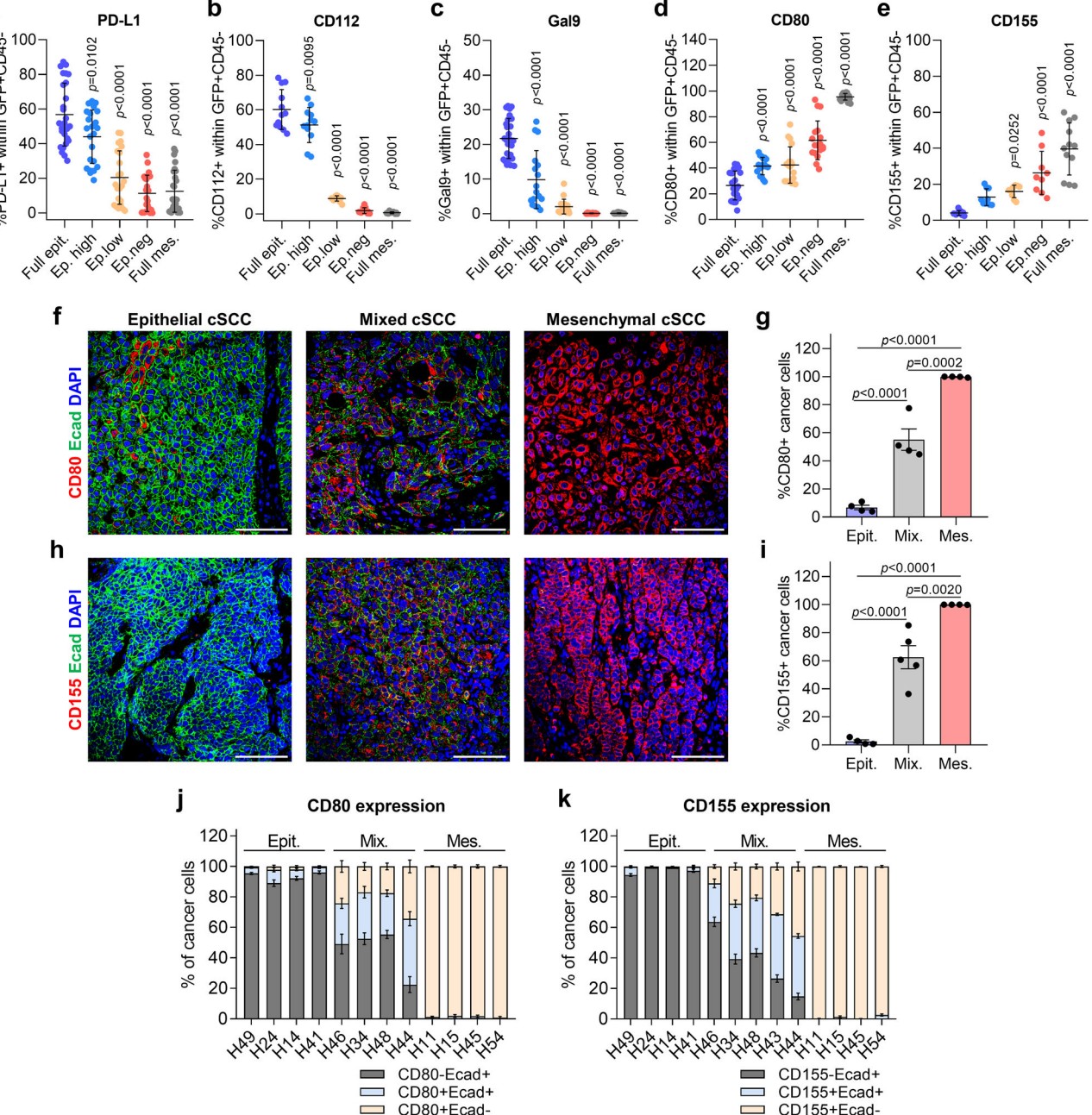

**Fig. 2 | IC ligand repertoire differs according to the epithelial, hybrid E/M, or mesenchymal features of cancer cells in mouse and patient cSCCs. a** Percentage of PD-L1[+] cells within full epithelial cancer cells from epithelial cSCCs ($n = 27$), EpCAM[high], EpCAM[low], and EpCAM[−] cancer cells from mixed cSCCs ($n = 23$), and full mesenchymal cancer cells from mesenchymal cSCCs ($n = 27$). **b** Percentage of CD112[+] cells within the indicated cancer cells ($n = 12$ tumors per group). **c** Percentage of Gal9[+] cells within full epithelial cancer cells from epithelial cSCCs ($n = 27$), EpCAM[high], EpCAM[low], and EpCAM[−] cancer cells from mixed cSCCs ($n = 17$), and full mesenchymal cancer cells from mesenchymal cSCCs ($n = 21$). **d** Percentage of CD80[+] cells within full epithelial cancer cells from epithelial cSCCs ($n = 24$), EpCAM[high], EpCAM[low], and EpCAM[−] cancer cells from mixed cSCCs ($n = 18$), and full mesenchymal cancer cells from mesenchymal cSCCs ($n = 21$). **e** Percentage of CD155[+] cells within full epithelial cancer cells from epithelial cSCCs ($n = 10$), EpCAM[high], EpCAM[low], and EpCAM[−] cancer cells from mixed cSCCs ($n = 9$), and full mesenchymal cancer cells from mesenchymal cSCCs ($n = 12$). **f, h** Representative immunofluorescence images of Ecad[+] (green), **f** CD80[+] or **h** CD155[+] (red), and DAPI nuclear (blue) staining in the indicated patient cSCCs. Scale bar, 100 µm. **g** Percentage of CD80[+] cancer cells relative to total cancer cells in the indicated patient cSCCs ($n = 4$ per group). **i** Percentage of CD155[+] cancer cells relative to total cancer cells in epithelial ($n = 4$), mixed ($n = 5$), and mesenchymal ($n = 4$) patient cSCCs. Each dot indicates the average quantification of at least five fields from different tumor regions. **j** Percentage of CD80[−]Ecad[+], CD80[+]Ecad[+], and CD80[+]Ecad[−] cancer cells relative to total cancer cells in the indicated patient cSCCs ($n = 4$ per group). **k** Percentage of CD155[−]Ecad[+], CD155[+]Ecad[+], and CD155[+]Ecad[−] cancer cells relative to total cancer cells in epithelial ($n = 4$), mixed ($n = 5$), and mesenchymal ($n = 4$) patient cSCCs. Data are represented as the mean ± SD (**a–e**) or ± SEM (**g, i, j, k**), and $n$ values indicate independent tumors (**a–e, g, i**). *P* values are determined by one-way ANOVA with Dunnett's (**a–e**) or Tukey's (**g, i**) multiple comparison tests. See Supplementary Fig. 2 for the gating strategy (**a–e**). Source data are provided as a Source Data file.

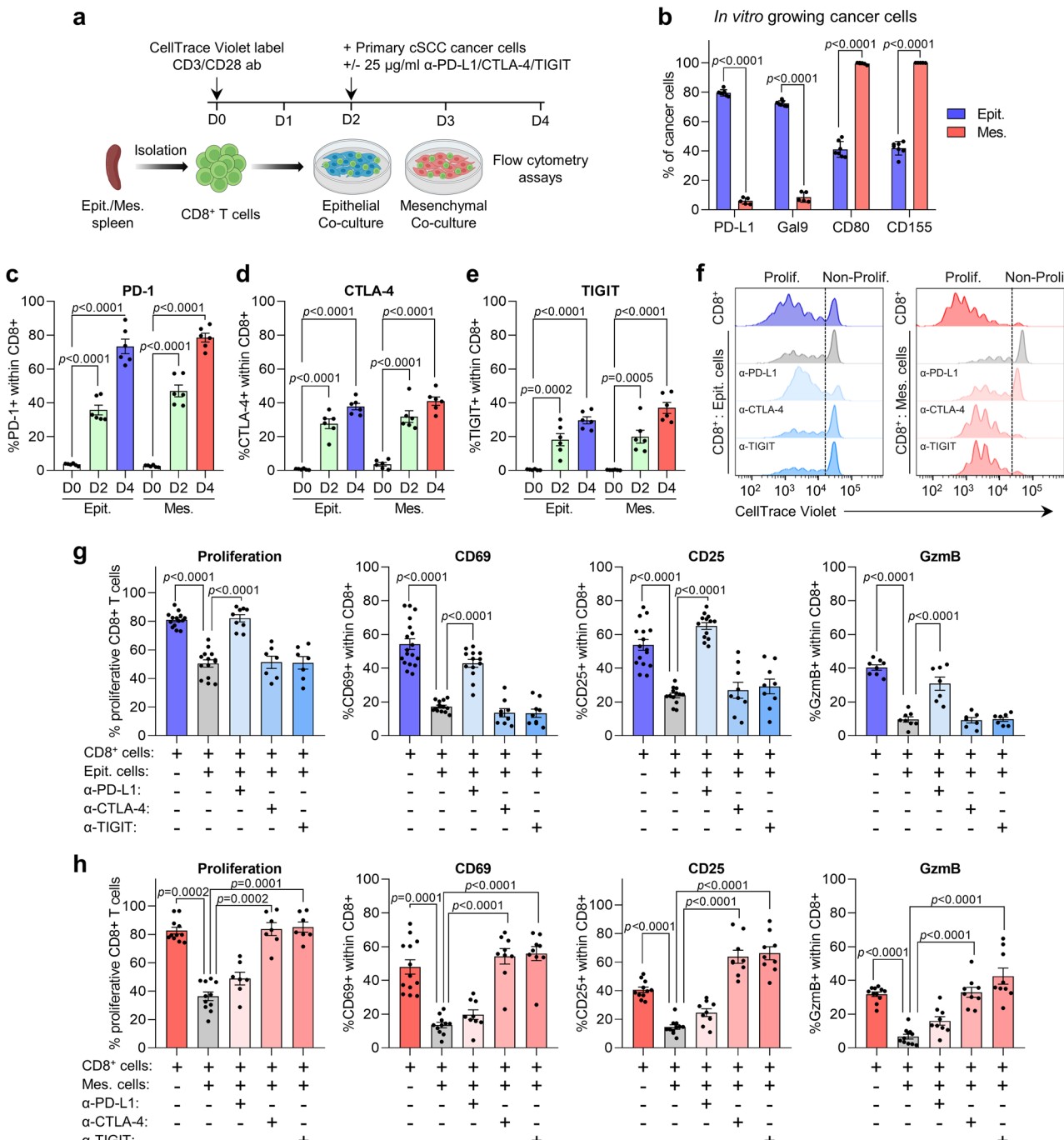

**Fig. 3 | Mouse epithelial and mesenchymal cancer cells activate different IC pathways to attenuate CD8⁺ T cell activity. a** Scheme of the experimental setup for epithelial and mesenchymal cancer cell co-cultures with CD3/CD28-activated CD8⁺ T cells isolated from the spleens of mice bearing epithelial or mesenchymal cSCCs for 4 days. Anti-PD-L1, anti-CTLA-4, and anti-TIGIT antibodies were added to the co-culture mediums on day 2. **b** Percentage of PD-L1⁺, Gal9⁺, CD80⁺ and CD155⁺ cells within epithelial ($n = 7$) and mesenchymal ($n = 5$) cancer cells growing in vitro. **c**–**e** Percentage of **c** PD-1⁺, **d** CTLA-4⁺, and **e** TIGIT⁺ cells within CD8⁺ T cells isolated from the spleens of mice bearing epithelial (Epit.) or mesenchymal (Mes.) cSCCs on days 0, 2 and 4 of in vitro culture ($n = 6$ per time point). **f** Representative CD8⁺ T cell proliferation as monitored by flow cytometry quantification of violet dye dilution when co-cultured with epithelial or mesenchymal cancer cells, with or without PD-L1, CTLA-4, and TIGIT-blocking antibodies. **g** Percentage of proliferative CD8⁺ T cells in the presence of epithelial cancer cells, without ($n = 14$) or with PD-L1, CTLA-4, and TIGIT-blocking antibodies ($n = 7$ per group). Percentage of CD69⁺ and

CD25⁺ CD8⁺ T cells in the presence of epithelial cancer cells, without ($n = 13$) or with PD-L1, CTLA-4, and TIGIT-blocking antibodies ($n = 9$ per group). Percentage of GzmB⁺ CD8⁺ T cells in the presence of epithelial cancer cells, without ($n = 8$) or with PD-L1, CTLA-4, and TIGIT-blocking antibodies ($n = 7$ per group). **h** Percentage of proliferative CD8⁺ T cells in the presence of mesenchymal cancer cells, without ($n = 11$) or with PD-L1, CTLA-4, and TIGIT-blocking antibodies ($n = 7$ per group). Percentage of CD69⁺, CD25⁺, and GzmB⁺ CD8⁺ T cells in the presence of mesenchymal cancer cells, without ($n = 11$) or with PD-L1, CTLA-4, and TIGIT-blocking antibodies ($n = 9$ per group). Data are represented as the mean ± SD (**b**) or ± SEM (**c**–**e**, **g**, **h**), and $n$ values indicate independent cancer cells (**b**) and $n$ values indicate independent cancer cells (**b**) or experiments (**c**–**e**, **g**, **h**). $P$ values are determined by unpaired two-sided Student's $t$-test (**b**), one-way ANOVA with Dunnett's multiple comparison test (**c**–**e**, **g**, and **h**: CD25 and GzmB), and Kruskal–Wallis with Dunn's multiple comparison test (**h**: proliferation and CD69). Source data are provided as a Source Data file.

primary cultures of epithelial and mesenchymal cancer cells was comparable to that observed in cancer cells growing in vivo (Fig. 3b), and that PD-1, CTLA-4, and TIGIT receptors were expressed by CD8[+] T cells after 2 and 4 days of in vitro CD3/CD28 activation (Fig. 3c–e). After 2 days of co-culture, epithelial and mesenchymal cancer cells both inhibited CD8[+] T cell proliferation and activity, as determined by violet dye dilution after subsequent cell divisions, the low level of expression of the T cell activation markers CD69 and CD25, and the reduced percentage of GzmB[+] CD8[+] T cells (Fig. 3f–h). Concomitantly, anti-PD-L1, anti-CTLA-4, and anti-TIGIT antibodies were added to the co-culture mediums to determine the relevance of these IC pathways in inhibiting CTL functions (Fig. 3a). Strikingly, anti-PD-L1 treatment markedly increased CTL proliferation and activity in the presence of epithelial cancer cells, but there were no significant effects after anti-CTLA-4 and anti-TIGIT treatments (Fig. 3f, g). However, anti-PD-L1 antibody showed no enhancement of CTL activity in the presence of mesenchymal cancer cells, while the impaired CTL activity was largely restored by anti-CTLA-4 and anti-TIGIT antibodies (Fig. 3f, h). These findings demonstrate that mouse epithelial and mesenchymal cancer cells activate different IC pathways to attenuate the cytotoxic activity of CD8[+] T cells. Indeed, while the PD-1-dependent IC pathway plays a significant role in blocking CTL activity by epithelial cancer cells, the CTLA-4 and TIGIT-dependent IC pathways are the most important for blocking CTL activity by mesenchymal cancer cells. Importantly, since cSCC patient samples recapitulate the same IC ligand alterations based on the epithelial and mesenchymal features of cancer cells, our data indicate that similar IC pathways may be involved in immune evasion in patient cSCCs and highlight our mouse cSCC model as a useful tool to assess the response to different ICB therapies.

### Cancer cell features influence the in vivo response of mouse cSCCs to ICB therapies

We next investigated which ICB therapies could most efficiently boost the in vivo antitumor response of mouse cSCCs depending on cancer cell features. We compared the response of epithelial (mostly composed of epithelial cancer cells), mixed (composed of epithelial and mesenchymal cancer cells), and mesenchymal cSCCs (mostly composed of mesenchymal cancer cells) to various ICIs (Fig. 4a). Consistent with the fact that epithelial cancer cells express high levels of PD-L1 and low levels of CD80 and CD155, epithelial mouse cSCCs responded similarly to anti-PD-L1 and anti-PD-1 therapies (Fig. 4b and Supplementary Fig. 5a), but they did not respond to anti-CTLA-4 and anti-TIGIT therapies (Fig. 4g–p and Supplementary Fig. 5h-s). Interestingly, anti-PD-L1 response was CD8[+] T cell-dependent rather than relying on NK cells (Fig. 4b and Supplementary Fig. 5a). In line with these results, anti-PD-L1 and anti-PD-1-treated epithelial mouse cSCCs showed a higher proportion of active CD8[+] T cells than control tumors, which were characterized by increased expression of activation markers like GzmB, CD69, and CD25, and reduced expression of the inhibitory receptors PD-1, TIGIT, and CTLA-4 (Fig. 4c, e and Supplementary Fig. 5b–e). Furthermore, no changes in the exhausted state of NK cells were detected between control and anti-PD-L1/PD-1-treated epithelial mouse cSCCs, despite an increase in their frequency (Fig. 4d, f and Supplementary Fig. 5f, g).

By contrast, mesenchymal mouse cSCCs, which express low levels of PD-L1, were resistant to anti-PD-L1 and anti-PD-1 therapies (Fig. 5a and Supplementary Fig. 6a). The absence of changes in the percentage of GzmB[+], CD69[+], CD25[+], PD-1[+]TIGIT[+] and PD-1[+]CTLA-4[+] cells within CD8[+] and NK cells indicated that anti-PD-L1 and anti-PD-1 therapies are not able to reverse the exhausted state of CTLs and NK cells in mesenchymal mouse cSCCs (Fig. 5b–e and Supplementary Fig. 6b–g). Surprisingly, anti-CTLA-4 and anti-TIGIT therapies significantly blocked the growth of mesenchymal mouse cSCCs (Fig. 5f, k and Supplementary Fig. 6h). Both treatments induced an increased infiltration of active CD8[+] T cells and NK cells (Fig. 5g–j, l, m and

Supplementary Fig. 6i–n), indicating that the activation of these IC pathways drives immune evasion in mesenchymal mouse cSCCs. To explore the extent to which this therapeutic effect was mediated by CTLs or NK cells, we performed CD8 and NK cell depletion experiments. Following successful CD8 and NK cell depletion (Fig. 5g, h, l, m), the delayed tumor growth conferred by anti-CTLA-4 and anti-TIGIT therapies was partially lost, suggesting that these responses are mediated by both CTLs and NK cells (Fig. 5f, k and Supplementary Fig. 6h). As expected from a homogeneous cancer cell composition, no changes in the relative content of epithelial EpCAM[+] and mesenchymal EpCAM[−] cancer cells vs. total cancer cells were observed after performing the ICB treatments on epithelial and mesenchymal mouse cSCCs (Figs. 4q–s and 5n, o).

These results prompted us to investigate whether mixed mouse cSCCs formed of epithelial and mesenchymal cancer cells should be treated with monotherapy or with a combination of different ICIs. Mixed mouse cSCCs treated with anti-PD-L1 antibody showed a reduction in tumor growth, accompanied by a significant increase in the percentage of active CD8[+] and NK cells (Fig. 6a–e, k-o and Supplementary Fig. 7a, c–f, k–o). Interestingly, anti-PD-L1 therapy favored the elimination of epithelial EpCAM[+] cancer cells and increased the frequency of mesenchymal EpCAM[−] cancer cells (Fig. 6p, r). Therefore, this initial good response to anti-PD-L1 therapy could turn into disease progression later, as the enriched mesenchymal cancer cells show resistance to this therapy (Fig. 5a). Mixed cSCC growth was also significantly reduced in response to anti-CTLA-4 and anti-TIGIT therapies (Fig. 6f, k and Supplementary Fig. 7b, k), and was associated with a higher percentage of active CD8[+] and NK cells in both ICB-treated mixed mouse cSCCs compared with control tumors (Fig. 6g–j, l–o and Supplementary Fig. 7g–j, l–o). In this case, anti-CTLA-4 and anti-TIGIT therapies not only led to a reduction in the percentage of mesenchymal EpCAM[−] cancer cells but also induced the enrichment of epithelial EpCAM[+] cancer cells (Fig. 6q, r). It is of particular note that the combination of anti-PD-L1 with anti-TIGIT reduced mixed cSCC growth and boosted CD8[+] and NK cell activity to a greater extent than did single ICB treatments (Fig. 6k–o and Supplementary Fig. 7k–o), highlighting that mouse cSCCs formed of epithelial and mesenchymal cancer cells should be treated with combined therapies to address both cancer cell components. Collectively, our data indicate that cancer cell features affect the response to ICB therapies and support the conclusion that the selection of these therapies should be based on the epithelial/mesenchymal features of cSCCs. In this regard, anti-PD-1/PD-L1 therapies elicit potent antitumor responses against epithelial cSCCs by reversing the exhausted state of CD8[+] T cells, mesenchymal cSCCs respond to anti-CTLA-4 and anti-TIGIT therapies in a CD8[+] and NK cell-dependent manner, and combined ICB therapies are more effective against cSCCs enriched in epithelial, hybrid E/M, and mesenchymal cancer cells by stimulating the activity of both CTLs and NK cells.

### Anti-PD-1/PD-L1 resistance in cSCC, HNSCC, and melanoma patient samples is associated with a higher frequency of hybrid E/M and mesenchymal cancer cells

To better understand whether cancer cell plasticity and the presence of hybrid E/M and mesenchymal cancer cells could influence anti-PD-1/PD-L1 response, we analyzed the cancer cell features in a retrospective cohort of pre-treatment samples from unresectable locally advanced and metastatic cSCC patients, and in another cohort of stage III/IV HNSCC patients (Supplementary Tables 1 and 2). We considered cSCC and HNSCC patients as responders if they achieved complete (CR) or partial response (PR) as the best response for at least 3 months, while non-responder patients were those who had stable (SD) or progressive disease (PD) as their best response, according to Response Evaluation Criteria in Solid Tumors (RECIST)[41]. In the cSCC cohort, of the seven patients who achieved CR or PR (median duration of response (DoR) of 21.5 months), two had subsequent PD with a median time to

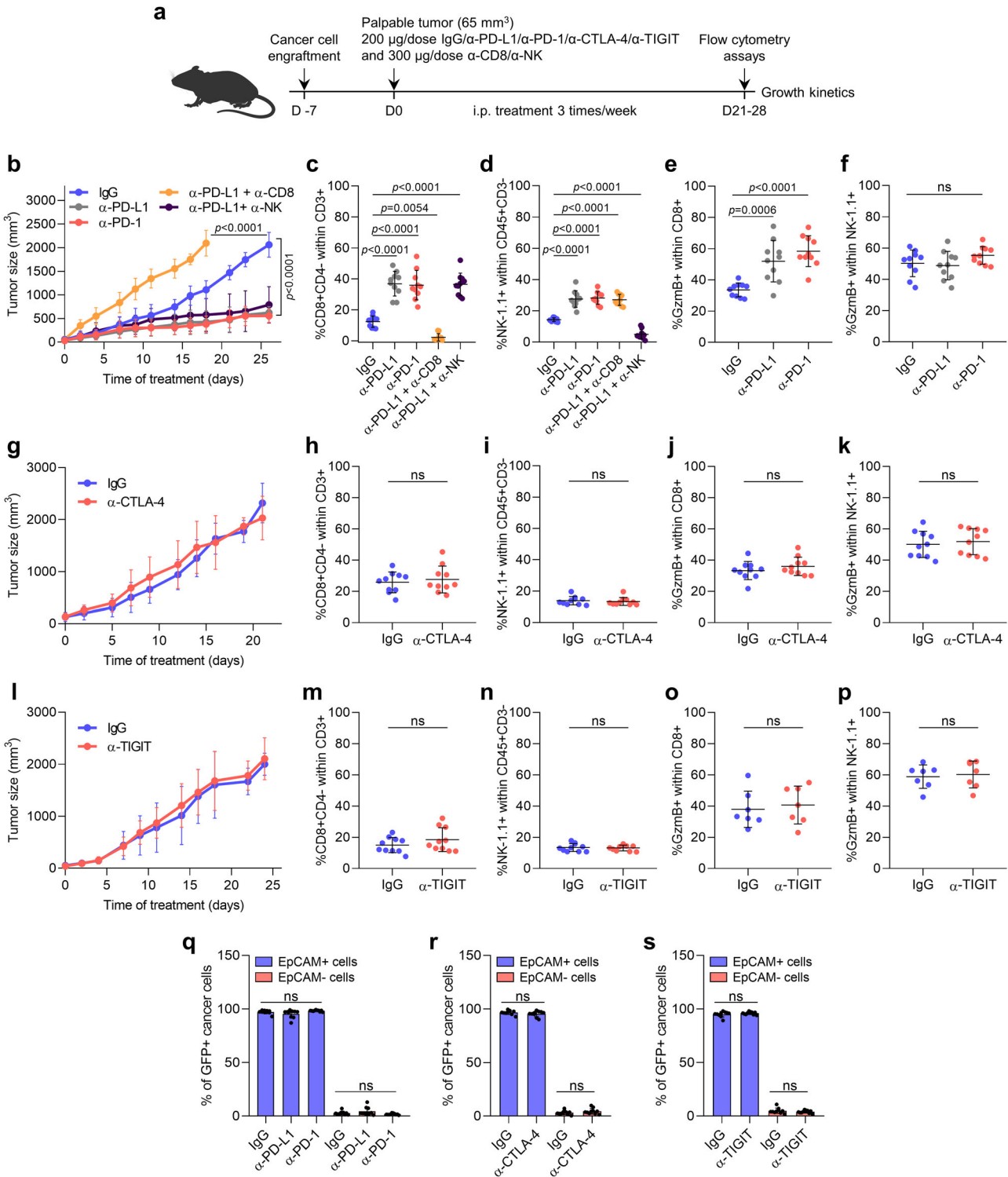

**Fig. 4 | Anti-PD-L1/PD-1 response is mediated by CD8+ T cells in mouse epithelial cSCCs.** **a** Experimental scheme for the treatment of mice bearing epithelial, mixed, and mesenchymal cSCCs with 200 μg/dose of IgG isotype control, anti-PD-L1, anti-PD-1, anti-CTLA-4, and anti-TIGIT antibodies, and 300 μg/dose of anti-CD8 and anti-NK1.1 antibodies (i.p. three times/week). All treatments started when engrafted tumors reached a volume of 65 mm³. **b** Growth kinetics of IgG control, anti-PD-L1, anti-PD-1, anti-PD-L1 + anti-CD8, and anti-PD-L1 + anti-NK-treated epithelial cSCCs (*n* = 10 per group). **c**–**f** Percentage of **c** CD8+ T cells, **d** NK cells, **e** GzmB+ CD8+ T cells, and **f** GzmB+ NK cells in the indicated epithelial cSCCs (*n* = 10 per group). **g** Growth kinetics of IgG control and anti-CTLA-4-treated epithelial cSCCs (*n* = 10 per group). **h**–**k** Percentage of **h** CD8+ T cells, **i** NK cells, **j** GzmB+ CD8+ T cells, and **k** GzmB+ NK cells in IgG control and anti-CTLA-4-treated epithelial

cSCCs (*n* = 10 per group). **l** Growth kinetics of IgG control and anti-TIGIT-treated epithelial cSCCs (*n* = 10 per group). **m**–**p** Percentage of **m** CD8+ T cells (*n* = 10 per group), **n** NK cells (*n* = 10 per group), **o** GzmB+ CD8+ T cells (*n* = 7 per group), and **p** GzmB+ NK cells (*n* = 7 per group) in IgG control and anti-TIGIT-treated epithelial cSCCs. **q**–**s** Percentage of GFP+EpCAM+ and GFP+EpCAM- cancer cells in the indicated epithelial cSCCs (*n* = 10 per group). All data are represented as the mean ± SD, and *n* values indicate independent tumors. *P* values are determined by two-way ANOVA test (**b**, **g**, **l**), one-way ANOVA with Dunnett's multiple comparison test (**c**–**f**, **q**), and unpaired two-sided Student's *t*-test (**h**–**k**, **m**–**p**, **r**, **s**). ns > 0.05: not significant. See Supplementary Fig. 2 for the gating strategy (**c**–**f**, **h**–**k**, **m**–**s**). Source data are provided as a Source Data file.

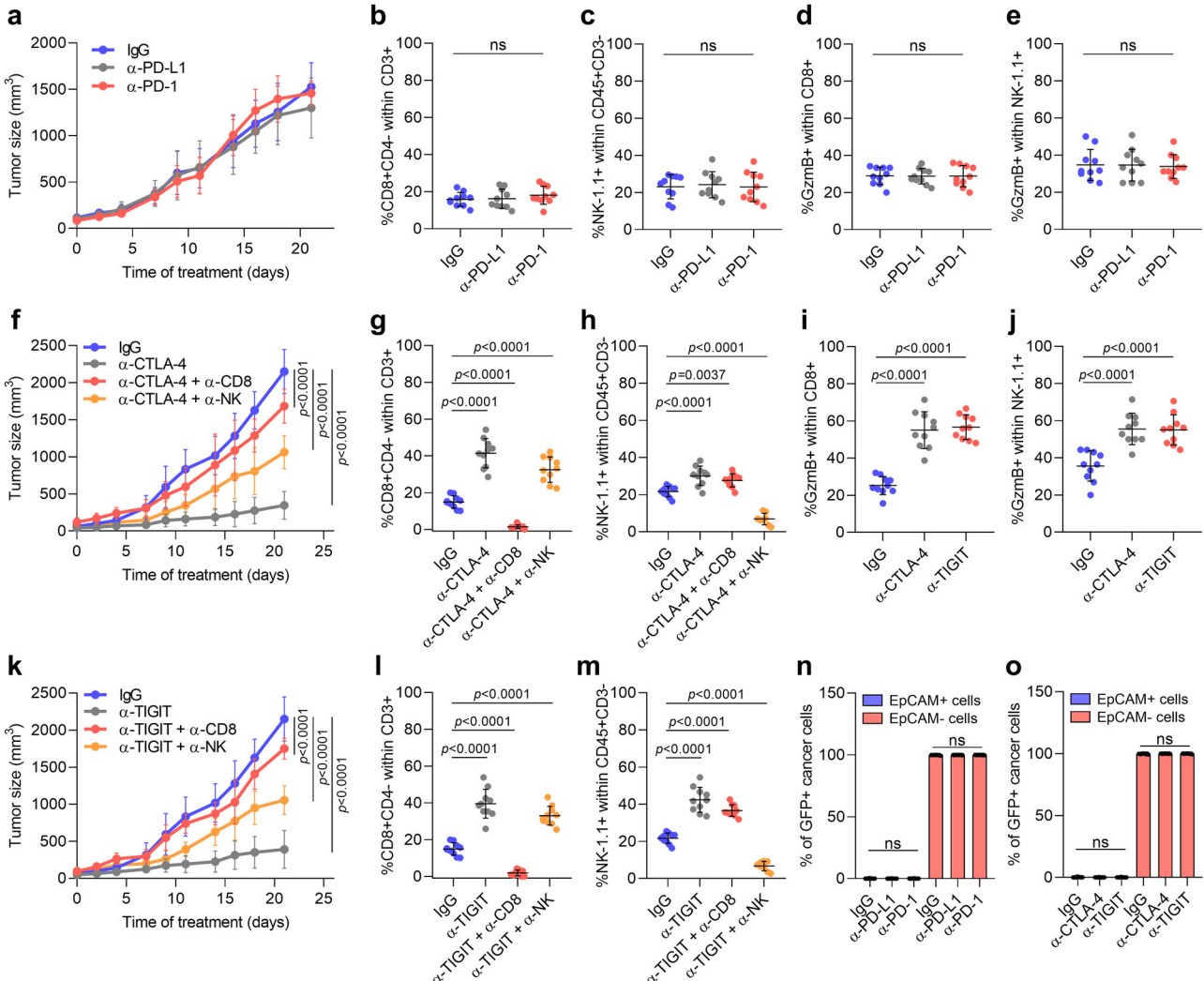

**Fig. 5 | Anti-CTLA-4 and anti-TIGIT responses are mediated by CD8+ and NK cells in mouse mesenchymal cSCCs. a** Growth kinetics of IgG control, anti-PD-L1, and anti-PD-1-treated mesenchymal cSCCs (*n* = 10 per group). **b–e** Percentage of **b** CD8+ T cells, **c** NK cells, **d** GzmB+ CD8+ T cells, and **e** GzmB+ NK cells in IgG control, anti-PD-L1, and anti-PD-1-treated mesenchymal cSCCs (*n* = 10 per group). **f, k** Growth kinetics of IgG control, (**f**) anti-CTLA-4, anti-CTLA-4 + anti-CD8, and anti-CTLA-4 + anti-NK or (**k**) anti-TIGIT, anti-TIGIT + anti-CD8, and anti-TIGIT + anti-NK-treated mesenchymal cSCCs (*n* = 10 per group). For better visualization, this experiment has been separated into two graphs in which the IgG control group is

the same. **g–j, l, m** Percentage of **g, l** CD8+ T cells, **h, m** NK cells, **i** GzmB+ CD8+ T cells, and **j** GzmB+ NK cells in the indicated mesenchymal cSCCs (*n* = 10 per group). **n, o** Percentage of GFP+EpCAM+ and GFP+EpCAM− cancer cells in the indicated mesenchymal cSCCs (*n* = 10 per group). All data are represented as the mean ± SD, and *n* values indicate independent tumors. *P* values are determined by two-way ANOVA test (**a, f, k**) and one-way ANOVA with Dunnett's multiple comparison test (**b–e, g–j, l–o**). ns > 0.05: not significant. See Supplementary Fig. 2 for the gating strategy (**b–e, g–j, l–o**). Source data are provided as a Source Data file.

progression of 21.5 months, and there were two deaths from progression-independent complications. The median time to progression of non-responder cSCC patients was 2.1 months, and there were four deaths due to PD and one from adverse events (Supplementary Table 1). In the HNSCC cohort, six patients achieved CR or PR, and the median DoR was 15.7 months. Of the six responder patients, four had subsequent PD with a median time to progression of 20.4 months, and there were two deaths due to PD and one death from adverse events. The median time to progression of non-responder HNSCC patients was 1.4 months, and there were eleven deaths due to PD (Supplementary Table 2).

Furthermore, as previous studies demonstrated that EMT induction in melanoma cancer cells promotes tumor growth and invasion[42–44], we also evaluated whether there was a correlation between cancer cell features and the efficacy of adjuvant anti-PD-1 therapy after resection in stage IIIC melanoma patients (Supplementary Table 3). In this clinical setting, we classified melanoma patients as

non-relapsed if they did not relapse within 18 months of starting adjuvant anti-PD-1 therapy, while relapsed patients were those who relapsed within that period. Of the five non-relapsed patients, one had a subsequent relapse at 33.8 months and died from PD. The median time to relapse of relapsed patients was 7.8 months, and there were four deaths due to PD (Supplementary Table 3).

Our analyses showed that anti-PD-1/PD-L1 responder and non-relapsed tumors had a higher percentage of epithelial Ecad+ cancer cells than non-responder and relapsed tumors (Fig. 7a–c, g, k and Supplementary Fig. 8a, f, k). Conversely, non-responder and relapsed tumors exhibited increased percentages of Vim+, CD80+, and CD155+ cancer cells (Fig. 7a, b, d–f, h–j, l–n and Supplementary Fig. 8b–d, g–i, l–n), indicating that cancer cell features might affect the efficacy of anti-PD-1/PD-L1 therapy in cSCC, HNSCC, and melanoma patients.

Given the potential clinical relevance of these results, we next studied whether there was an association between the percentage of Ecad+, Vim+, CD80+, CD155+, Ecad−Vim+, Ecad−CD80+ and Ecad−CD155+

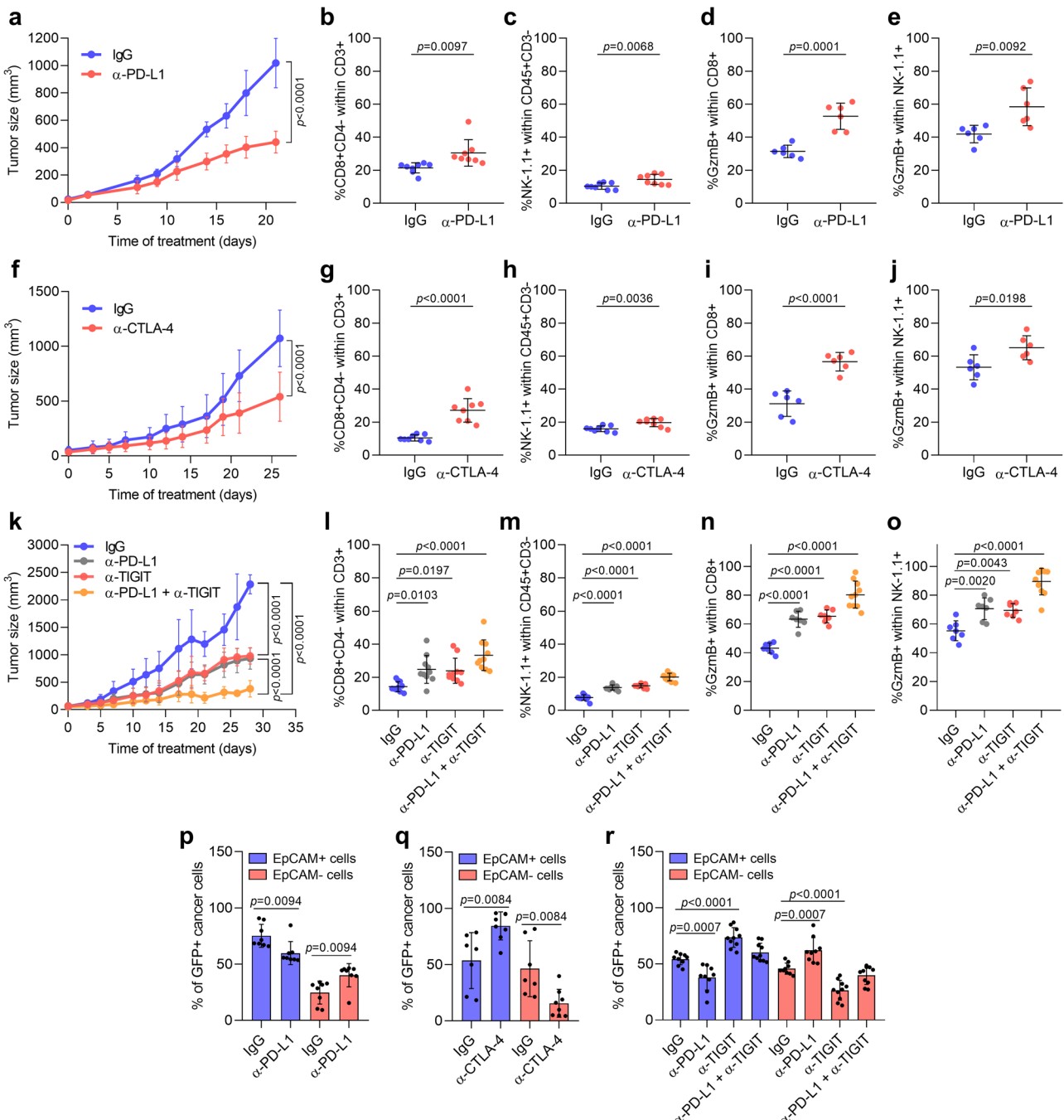

**Fig. 6 | Combined anti-PD-L1 and anti-TIGIT therapies suppress mixed mouse cSCC growth by targeting epithelial and mesenchymal cancer cells. a** Growth kinetics of IgG control and anti-PD-L1-treated mixed cSCCs (*n* = 8 per group). **b**–**e** Percentage of **b** CD8⁺ T cells (*n* = 8), **c** NK cells (*n* = 8), **d** GzmB⁺ CD8⁺ T cells (*n* = 6), and **e** GzmB⁺ NK cells (*n* = 6) in IgG control and anti-PD-L1-treated mixed cSCCs. **f** Growth kinetics of IgG control and anti-CTLA-4-treated mixed cSCCs (*n* = 8 per group). **g**–**j** Percentage of **g** CD8⁺ T cells (*n* = 8), **h** NK cells (*n* = 8), **i** GzmB⁺ CD8⁺ T cells (*n* = 6), and **j** GzmB⁺ NK cells (*n* = 6) in IgG control and anti-CTLA-4-treated mixed cSCCs. **k** Growth kinetics of IgG control, anti-PD-L1, anti-TIGIT, and anti-PD-L1 + anti-TIGIT-treated mixed cSCCs (*n* = 10 per group). **l**–**o** Percentage of **l** CD8⁺

T cells (*n* = 10), **m** NK cells (*n* = 10), **n** GzmB⁺ CD8⁺ T cells (*n* = 7), and **o** GzmB⁺ NK cells (*n* = 7) in the indicated mixed cSCCs. **p**–**r** Percentage of GFP⁺EpCAM⁺ and GFP⁺EpCAM⁻ cancer cells in the indicated mixed cSCCs (PD-L1 and CTLA-4 experiments: *n* = 8 per group; PD-L1/TIGIT experiment: *n* = 10 per group). All data are represented as the mean ± SD, and *n* values indicate independent tumors. *P* values are determined by two-way ANOVA test (**a**, **f**, **k**), unpaired two-sided Student's *t*-test (**b**–**e**, **g**–**j**, **p**, **q**), and one-way ANOVA with Dunnett's multiple comparison test (**l**–**o**, **r**). See Supplementary Fig. 2 for the gating strategy (**b**–**e**, **g**–**j**, **l**–**r**). Source data are provided as a Source Data file.

cancer cells and the risk of disease progression after anti-PD-1/PD-L1 therapy in cSCC and HNSCC patients or of relapse during adjuvant anti-PD-1 therapy in melanoma patients (Fig. 7o). Cox proportional hazards models showed that the higher the percentage of epithelial Ecad⁺ cancer cells, the lower the risk of progression/relapse (HR < 1.00, protective factor), while the higher the percentage of mesenchymal

Ecad⁻Vim⁺, Ecad⁻CD80⁺ and Ecad⁻CD155⁺ cancer cells, the higher the risk of progression/relapse (HR > 1.00, risk factors) in the cSCC, HNSCC, and melanoma cohorts (Fig. 7o). The correlation between a higher percentage of Vim⁺, CD80⁺, and CD155⁺ cancer cells and a higher risk of progression/relapse was not significant in all cohorts (Fig. 7o), but a trend was observed that should be validated with a larger number

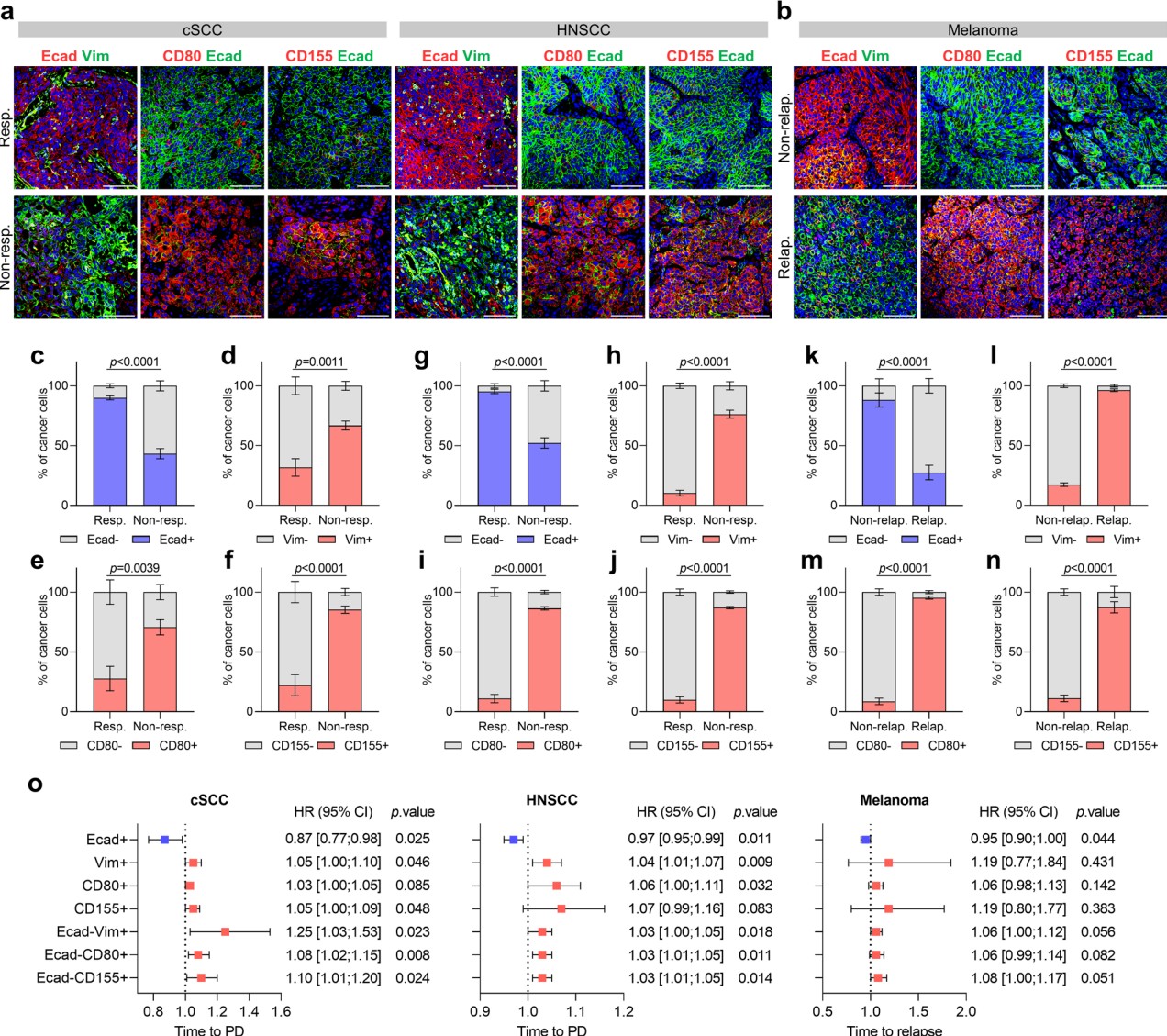

**Fig. 7 | Anti-PD-1/PD-L1 resistance in cSCC, HNSCC, and melanoma patient samples is associated with a higher frequency of hybrid E/M and mesenchymal cancer cells. a, b** Representative immunofluorescence images of Ecad⁺, CD80⁺ or CD155⁺ (red), Vim⁺ or Ecad⁺ (green), and DAPI nuclear (blue) staining in **a** anti-PD-1/PD-L1 responder and non-responder cSCCs (left panel) and HNSCCs (right panel), and **b** anti-PD-1 non-relapsed and relapsed melanomas. Scale bar, 100 μm. **c–f** Percentage (mean ± SEM) of **c** Ecad⁻/Ecad⁺, **d** Vim⁻/Vim⁺, **e** CD80⁻/CD80⁺, and **f** CD155⁻/CD155⁺ cancer cells relative to total cancer cells in anti-PD-1/PD-L1 responder and non-responder cSCCs (*n* = 7 per group). **g–j** Percentage (mean ± SEM) of **g** Ecad⁻/Ecad⁺, **h** Vim⁻/Vim⁺, **i** CD80⁻/CD80⁺, and **j** CD155⁻/CD155⁺ cancer cells relative to total cancer cells in anti-PD-1/PD-L1 responder and non-responder HNSCCs (*n* = 6 responders, *n* = 13 non-responders). **k–n** Percentage (mean ± SEM) of **k** Ecad⁻/

Ecad⁺, **l** Vim⁻/Vim⁺, **m** CD80⁻/CD80⁺, and **n** CD155⁻/CD155⁺ cancer cells relative to total cancer cells in anti-PD-1 non-relapsed and relapsed melanomas (*n* = 5 per group). **o** Forest plots showing the hazard ratios (HR; blue and red squares) ± 95% confidence intervals (CI; horizontal lines) of the association between the indicated variables and time to progression (PD) or time to relapse. Variables with HR < 1.00 represent protective factors, whereas HR > 1.00 indicates risk factors. Anti-PD-1/PD-L1 responder and non-responder cSCCs (*n* = 7 per group) and HNSCCs (*n* = 6 responders, *n* = 13 non-responders), and anti-PD-1 non-relapsed and relapsed melanomas (*n* = 5 per group). *P* values are determined by unpaired two-sided Student's *t*-test (**c–n**) and two-sided Cox proportional hazards models (**o**). Source data are provided as a Source Data file.

of patient samples. Indeed, according to Youden's index maximization criterion, we determined in the cSCC cohort that the presence of <62% Ecad⁺ cancer cells, and more than 55% Vim⁺ cancer cells, 36% CD80⁺ cancer cells, 74% CD155⁺ cancer cells, 37% Ecad⁻Vim⁺ cancer cells, 18% Ecad⁻CD80⁺ cancer cells, and 26% Ecad⁻CD155⁺ cancer cells might be risk factors for anti-PD-1/PD-L1 resistance (Supplementary Figs. 8a–e and 9a–d). In the HNSCC cohort, the presence of <71% Ecad⁺ cancer cells, >21% Vim⁺, CD80⁺ and CD155⁺ cancer cells, and >7% Ecad⁻Vim⁺, Ecad⁻CD80⁺ and Ecad⁻CD155⁺ cancer cells might be risk factors for anti-PD-1/PD-L1 resistance (Supplementary Figs. 8f–j and 9e–h). Finally, in the melanoma cohort, the presence of <49% Ecad⁺ cancer cells,

and more than 22% Vim⁺ cancer cells, 16% CD80⁺ cancer cells, 18% CD155⁺ cancer cells, and 9% Ecad⁻Vim⁺, Ecad⁻CD80⁺ and Ecad⁻CD155⁺ cancer cells might be risk factors for tumor relapse after adjuvant anti-PD-1 therapy (Supplementary Figs. 8k–o and 9i–l).

Taken together, our findings indicate that the enrichment of mesenchymal cancer cells and the change in the IC ligand repertoire towards CD80/CD155 expression act as predictive biomarkers of anti-PD-1/PD-L1 response in cSCC, HNSCC, and melanoma patient samples. Therefore, cSCCs, HNSCCs, and melanomas mostly composed of epithelial Ecad⁺ cancer cells may respond better to anti-PD-1/PD-L1 therapy, while those composed of epithelial Ecad⁺ and mesenchymal Vim⁺

cancer cells, and that do not respond to anti-PD-1/PD-L1 therapy, may be treated with anti-CTLA-4 or anti-TIGIT therapies, or even with combined ICB therapies, to improve their clinical outcomes. In conclusion, our results highlight the importance of using epithelial/ mesenchymal cancer cell heterogeneity as a biomarker to prospectively identify responsive patients to ICB treatments.

## Discussion

The use of anti-PD-1 therapies in cSCC and HNSCC, and anti-CTLA-4 alone or in combination with anti-PD-1 in melanoma have represented a major advance in cancer treatment[45]. However, it is not well understood why only a subset of patients respond to immunotherapies[46,47]. Indeed, 50% of locally advanced and metastatic cSCC patients and 65% of recurrent and metastatic (R/M) HNSCC patients remain unresponsive or develop acquired resistance to anti-PD-1 therapy[6,7,48–50]. On the other hand, the combination of ipilimumab (anti-CTLA-4) and nivolumab (anti-PD-1) significantly increases the response of about 60% of melanoma patients compared with monotherapy or chemotherapy[51,52]. Therefore, it is not only necessary to identify biomarkers of response to these treatments or of the benefit of monotherapy *vs.* combined strategies for cSCC, HNSCC, and melanoma patients, but also to understand whether cancer cell heterogeneity might explain how patients with similar tumor types show varying sensitivity to ICB therapies.

Using a mouse cSCC model that progresses from epithelial to mesenchymal cSCCs through mixed cSCCs, we have identified that the progressive induction of the EMT program is associated with the acquisition of cancer cell plasticity, as described in previous studies[9–14]. This situation generates epithelial, hybrid E/M, and mesenchymal cancer cell states, which are also detected in patient cSCCs. Since these hybrid E/M cancer cells with different degrees of expression of epithelial (e.g., EpCAM, E-cadherin) and mesenchymal (e.g., Vimentin) markers are more prone to progress to the mesenchymal state during mouse cSCC growth, their presence could be a risk factor for the increased relapse and metastasis observed in patients with advanced/ high-risk cSCCs. In this regard, other studies have demonstrated that E/M states are more efficient in reaching the circulation, colonizing, and forming metastases[12,53–56], and their presence has been linked to poor patient survival, immune evasion, and therapy resistance in other tumors such as breast, lung, ovarian, HNSCC, colorectal, pancreatic, and prostate cancers[57–60].

Here, we have characterized how epithelial/mesenchymal cSCC features affect the composition of the surrounding TME and the sensitivity to different ICB therapies, as it is well established that a dynamic interplay between cancer and immune cells promotes tumor growth and invasion, and hinders effective treatments[61–66]. Our data demonstrate that mouse and patient cSCCs enriched in hybrid E/M and mesenchymal cancer cells have a higher frequency of immunosuppressive cells (M-MDSCs, Treg cells, and M2-like macrophages) than epithelial cSCCs, consistent with other studies in which tumors enriched in the EMT signature are associated with an immunosuppressed TME[23,67–72]. Mixed and mesenchymal cSCCs also contain a higher frequency of CTLs and NK cells, which exhibit an exhausted phenotype compared with those in epithelial cSCCs. Since we have identified changes in TME composition depending on cancer cell features, future studies will be necessary to identify which factors derived from hybrid E/M and mesenchymal cancer cells may contribute to immunosuppression, in order to modulate them to promote an immunostimulatory environment in cSCCs.

It is well known that tumors can also evade antitumor immunity by upregulating the expression of IC ligands such as PD-L1, CD80, and CD155[73–78]. The interaction of these ligands with their respective IC receptors expressed by CTLs or NK cells produces a dysfunctional state known as exhaustion[79,80]. The blockade of these pathways results in a reversal of exhaustion[81–83]. In this work, using in vitro co-culture experiments and a preclinical cSCC model, we have identified a key

role of cancer cells directly evading immune attack and regulating CTL and NK-cell functions through differential expression of IC ligands, which highlights the role of EMP as an important immune evasion mechanism. Identifying the mechanisms of EMP-mediated immune evasion is important, as most patient cSCCs are heterogeneous with diverse E/M states, and the underlying mechanisms would also apply to other cancer types with an active EMT. Interestingly, our findings demonstrate that most epithelial cancer cells express PD-L1, whereas hybrid E/M and mesenchymal cancer cells reduce PD-L1 expression and upregulate CD80 and CD155 expression, which are the ligands of CTLA-4 and TIGIT receptors. In this regard, it was previously shown that TGF-β-responsive tumor-initiating cells become resistant to adoptive cytotoxic T cell transfer (ACT) immunotherapy by inducing the expression of CD80 in a mouse cSCC model[76]. Given that TGF-β is a well-characterized EMT inducer in several cancer cell types[84], these observations are consistent with the induced expression of CD80 detected in hybrid E/M and mesenchymal cSCC cells. On the other hand, although previous reports have shown that PD-L1 expression is induced in response to IFN-γ signaling[85,86], CTLs and NK cells are more exhausted and express less IFN-γ in mixed and mesenchymal cSCCs, which may preclude the IFN-γ-mediated stimulation of PD-L1 expression in hybrid E/M and mesenchymal cancer cells. Therefore, future studies will be necessary to identify the mechanisms that regulate the switch of IC ligand expression according to epithelial or mesenchymal cancer cell features, which may be used as potential therapeutic targets to enhance ICB therapy.

In addition, although other studies have reported that the acquisition of EMT-like properties induces the expression of PD-L1 by cancer cells[87–91], in this scenario, it is challenging to explain why the induction of the EMT program is associated with anti-PD-1 resistance[92], as immune evasion might be expected to be based on the PD-1/PD-L1 axis in these mesenchymal tumors. Importantly, our data reveal that based on their epithelial or mesenchymal features, mouse cSCC cancer cells inhibit the antitumor response of effector immune cells by using different IC pathways. Anti-PD-1/PD-L1 therapies elicit potent antitumor responses against epithelial cSCCs by reversing the exhausted state of CD8[+] T cells, whereas mesenchymal cSCCs expressing CD80 and CD155 are refractory to anti-PD-1/PD-L1 therapy and respond to anti-CTLA-4 and anti-TIGIT therapies in a CD8[+] and NK cell-dependent manner. Accordingly, the anti-PD-L1/TIGIT combination is the most effective strategy for blocking the growth of cSCCs that contain epithelial, hybrid E/M, and mesenchymal cancer cells through stimulation of both CTLs and NK cells, highlighting that mixed cSCCs should be treated with combined therapies to address both cancer cell components. These results reveal that E/M plasticity alters the immunomodulatory properties of cancer cells and drives their resistance to ICB therapies, indicating the importance of cancer cell heterogeneity as indicative of responsiveness.

Finally, although validation in a larger independent case-cohort is needed, we have identified E-cadherin, Vimentin, CD80, and CD155 proteins as predictive biomarkers of response to anti-PD-1/PD-L1 therapy, since TMB reflecting neoantigen diversity and PD-L1 score are not exclusionary predictive biomarkers of the clinical benefit of anti-PD-1/PD-L1-blocking antibodies in locally advanced and metastatic cSCCs, HNSCCs, and other tumor types[93–99]. In addition, our results highlight the prominent role of other surface receptors of CTLs and NK cells, such as TIGIT and CTLA-4, as actionable co-inhibitory signals beyond PD-1 in advanced/high-risk cSCCs enriched in hybrid E/M and mesenchymal cancer cells. In accordance with our findings, it has been recently reported that melanoma cells harboring a mesenchymal-like state are enriched in on-treatment lesions from refractory ICB patients[100]. Altogether, our study sheds light not only on anti-PD-1/PD-L1 resistance in cSCC, HNSCC, and melanoma patients but also on potential biomarkers to predict the response to this therapy and possible alternative treatments for these diseases.

## Methods

### Ethical regulations

This study complies with all ethical regulations. Clinical patient samples had approval from the Research Ethics Committee of the Bellvitge University Hospital (Barcelona, Spain), and conformed to the principles of the Declaration of Helsinki (PR392/20 for cSCC, PR381/19 and PR254/21 for HNSCC, and PR186/22 for melanoma). The treatment of the personal data was adjusted to the provisions of the European Data Protection Regulation. Two melanoma patient samples were provided by the IdiPAZ Biobank (PT20-0004), integrated into the Biobanks and Biomodels ISCIII Platform, and they were processed following standard operating procedures with the appropriate approval of the Ethics and Scientific Committees. All research involving animals was performed at the IDIBELL animal facility in compliance with the guidelines and protocols approved by the IDIBELL ethics committee (18003, DMAH10402), and in accordance with Spanish national regulations. For all experimental procedures, the maximal tumor volume allowed was 2 cm³, as approved by the IDIBELL ethics committee. In some cases, this limit was exceeded on the last day of measurement, and the mice were immediately euthanized.

### cSCC, HNSCC, and melanoma patient samples

cSCC (locally advanced and metastatic), HNSCC (stage III/IV), and melanoma (stage IIIC) patient samples were supplied by the Pathology Unit and Biobank of the Bellvitge University Hospital (Barcelona, Spain) and by the IdiPAZ Biobank (PT20-0004), and were fully anonymized. Age, gender, or ethnicity were not considered in the study design. Pathological review of tumor tissues was performed in the Pathology Service of the Bellvitge University Hospital as part of the standard clinical care.

In the cSCC and HNSCC cohorts, anti-PD-1/PD-L1 responders were defined as patients with complete or partial response of >3 months, while non-responders were defined as patients with stable or progressive disease as their best response (cSCC: $n = 7$ per group; HNSCC: $n = 6$ responders and 13 non-responders), according to Response Evaluation Criteria in Solid Tumors (RECIST)[41]. In the melanoma cohort, we considered patients as non-relapsed if they did not relapse within 18 months of starting adjuvant anti-PD-1 treatment, while relapsed patients were those who relapsed within that period ($n = 5$ per group). Some patients were treated within standard clinical practice and others within several clinical trials[6,7,101–103]. All available clinical information is included in Supplementary Tables 1–3. All patients were fully informed and provided written informed consent to participate in the study.

### Animals

A colony of C57BL/6 and FVB mice was maintained in-house for crossing. All animals used for experiments were 6–8-week-old male mice (C57BL6/FVB F1 background). In the interest of ensuring the reproducibility of tumor kinetics and growth, the animal experiments were conducted using only one gender. Mice were kept in a pathogen-free facility with a 12-h light/dark cycle at constant temperature ($22 \pm 2\,°C$), and with ad libitum access to food and water.

### Primary cSCC cancer cell cultures

Mouse cSCC cancer cells were derived from spontaneous or DMBA/TPA-induced tumors, which were previously generated by orthotopic serial engraftments[8]. After depleting red blood cells and CD31⁺ endothelial cells, primary α6-integrin⁺CD45⁻ cancer cells were isolated by FACS from WD-SCCs (full epithelial), MD/PD-SCCs (epithelial EpCAM⁺ and mesenchymal EpCAM⁻), and PD/S-SCCs (full mesenchymal) (Supplementary Fig. 2). Isolated cancer cells were then transduced with an MSCV-IRES-GFP lentivirus plasmid, thereby making it possible to identify them by the expression of green fluorescent protein (GFP). GFP⁺ full epithelial, GFP⁺EpCAM^high, GFP⁺EpCAM^low, and GFP⁺EpCAM⁻ cancer cells were sorted by FACS prior to engraftment into

immunocompetent syngeneic mice to generate cSCCs (Supplementary Fig. 2). Cells were cultured in basic DMEM-F12 medium (Life Technologies, 31331-093) supplemented with 1X B27 (Life Technologies, 17504-044) and 1% penicillin/streptomycin (P/S, Biowest, L0022-100), and were grown at 37 °C in a humidified 5% CO₂ incubator.

### Tumor-cell grafting and in vivo treatments

To generate epithelial (>70% EpCAM⁺ cancer cells), mixed (10–70% EpCAM⁺ cancer cells), and mesenchymal (<10% EpCAM⁺ cancer cells) cSCCs, FACS-isolated GFP⁺ full epithelial cancer cells from WD-SCCs, and epithelial GFP⁺EpCAM⁺ and mesenchymal GFP⁺EpCAM⁻ cancer cells from MD/PD-SCCs (10,000 cells) were mixed 1:1 with Matrigel Basement membrane matrix (Corning, 356234), and then subcutaneously engrafted into the back skin of 6–8-week-old immunocompetent syngeneic male mice (C57BL6/FVB F1 background). cSCC growth was monitored by caliper measurements three times per week, and tumor volume was calculated using the formula $V\,(mm^3) = \pi/6 \times L \times W^2$ ($L$: largest tumor diameter, $W$: perpendicular measurement). When tumors generated reached a volume of 65 mm³ (5 × 5 mm), mice were randomly assigned to a control or ICB treatment group and treated intraperitoneally three times per week with a 200 μg/dose of mouse IgG2b isotype control (clone MPC-11, BioXCell, BE0086), polyclonal rat IgG isotype control (BE0094), anti-PD-L1 (clone 10F.9G2, BioXCell, BE0101), anti-PD-1 (clone RMP1-14, BioXCell, BE0146), anti-CTLA-4 (clone UC10-4F10-11, BioXCell, BE0032), and anti-TIGIT (clone 1G9, BioXCell, BE0274) antibodies for 21–28 days. Depletion of CD8 and NK cells was achieved using 300 μg/dose of anti-CD8α (clone 2.43, BioXCell, BE0061) and anti-NK1.1 (clone PK136, BioXCell, BE0036) antibodies for 21–28 days. Once treatment was completed, tumors were excised and processed by flow cytometry assays. Mice were also checked for symptoms of poor health or discomfort during the treatment.

### cSCC cancer cell sorting and flow cytometry assays

For flow cytometry analysis and sorting, excised mouse cSCCs were mechanically minced and incubated in RPMI medium (Life Technologies, 61870044) with 10% FBS (Life Technologies, 10270106), 20 mM HEPES (Sigma, H3537), 1% antibiotic/antimycotic (Ab/Am, Biowest, L0010-100), 1600 U/ml collagenase type I (Sigma, C0130), and 70 U/ml dispase (Life Technologies, 17105-041), overnight at 37 °C. Cell suspensions were filtered and then depleted of red blood cells by incubating with ACK lysis buffer (Lonza, BP10-548E) for 10 min at room temperature. For endothelial cell depletion, cell suspensions were incubated with a rat anti-mouse CD31 antibody (1:100, BD Bioscience, 550274) for 30 min at 4 °C, and then with Dynabeads anti-rat IgG (1:33, Life Technologies, 11035) for 30 min at 4 °C. For cell-surface staining, cells were blocked with 1 mg/ml IgG (Sigma, I5381) and stained with a cocktail of cell-surface antibodies in staining buffer (5% FBS in PBS) for 30 min at 4 °C: from Biolegend, CD11b-APC 1:250 (M1/70, 101211), CD11b-PE/Cy7 1:250 (M1/70, 101215), CD152 (CTLA-4)-PE/Cy7 1:250 (UC10-4B9, 106313), CD155-PE/Cy7 1:200 (TX56, 131511), CD223 (LAG-3)-PE/Cy7 1:250 (C9B7W, 125225), CD226 (DNAM-1)-PE/Cy7 1:250 (10E5, 128811), CD25-PE/Cy7 1:200 (PC61, 102015), CD274 (PD-L1)-PE/Cy7 1:200 (10F.9G2, 124313), CD279 (PD-1)-APC/Cy7 1:250 (29F.1A12, 135223), CD28-PE/Cy7 1:250 (37.51, 102125), CD3ε-APC 1:200 (145-2C11, 100311), CD366 (TIM-3)-PE/Cy7 1:250 (B8.2C12, 134009), CD4-PE/Cy7 1:200 (RM4-5, 100528), CD49f (α6-integrin)-FITC 1:10 (GoH3, 313605), CD69-PE/Cy7 1:200 (H1.2F3, 104511), CD8a-PE 1:200 (53-6.7, 100707), CD80-PE/Cy7 1:250 (16-10A1, 104733), F4/80-APC/Cy7 1:200 (BM8, 123118), Galectin9-PE/Cy7 1:250 (108A2, 137913), Ly-6G/Ly-6C (Gr-1)-PE/Cy7 1:250 (RB6-8C5, 108415), Ly-6C-PE/Cy7 1:250 (HK1.4, 128017), Ly-6G-APC 1:250 (1A8, 127613), NK-1.1-PE 1:200 (PK136, 108707), TIGIT (Vstm3)-PE/Cy7 1:250 (1G9, 142107); from BD Bioscience, CD11b-PE 1:250 (M1/70, 557397); from eBioscience, CD206-APC 1:200 (MR6F3, 17-2061-80), CD326 (EpCAM)-APC-eF780 1:400 (G8.8, 47-5791-82); from TONBO, CD45-PE 1:350 (30-F11, 50-0451); from R&D Systems, CD112 (Nectin-2)-APC 1:200 (829038,

FAB3869A). Cells were then washed with 0.5% BSA, 2 mM EDTA in PBS, and resuspended in analysis buffer (2% FBS, 2 mM EDTA in PBS). Viability was assessed with DAPI (Thermo Scientific, 62248). For intracellular cell staining, cells were stimulated with Leukocyte Activation Cocktail with GolgiPlug™ (BD Bioscience, 550583) for 4 h at 37 °C, stained using the LIVE/DEAD™ Fixable Violet Dead Cell Stain Kit (1:1000, Life Technologies, L34963) for 30 min at 4 °C, and incubated with a cocktail of cell-surface antibodies for 30 min at 4 °C. Cells were then fixed with PFA 4% (Electron Microscopy Sciences, 15710-S) for 20 min at 4 °C, permeabilized with Permeabilization Buffer 1X (Life Technologies, 00-8333-56) for 15 min at 4 °C, and stained with antibodies recognizing intracellular antigens for 30 min at 4 °C (Granzyme B-PE/Cy7 1:200, NGZB, eBioscience, 25-8898-80; IFN-γ-PE/Cy7 1:200, XMG1.2, Biolegend, 505825). See Supplementary Fig. 2 for the gating strategies for cancer and immune cell populations. All antibodies used for flow cytometry are listed in Supplementary Table 4. Flow cytometry sorting and analysis were performed on BD FACSAria Fusion equipment, and data were analyzed with FlowJo v10.4.2 software.

### cSCC cancer cell : CD8⁺ T cell co-cultures

CD8⁺ T cells were isolated from the spleens of C57BL6/FVB F1 mice bearing epithelial or mesenchymal cSCCs. Briefly, spleens were mashed in PBS with 2% FBS using a 1-ml syringe and filtered through a 70-μm filter. Red blood cells were lysed with ACK lysis buffer for 5 min on ice. CD8⁺ T cells were isolated using the MojoSort™ Mouse CD8 T Cell Isolation Kit (Biolegend, 480008) according to the manufacturer's protocol, and then were labeled with 2.5 μM CellTrace™ Violet dye (Thermo Fisher, C34557) for 20 min at 37 °C. Purified CD8⁺ T cells were resuspended in a T cell medium (RPMI with 10% FBS, 1% non-essential amino acids, 1% Na-pyruvate, 1% L-glutamine, 1% P/S, 50 μM β-mercaptoethanol), and activated in vitro by incubating with 1 μg/ml of anti-CD3e (clone 145-2C11, eBioscience, 16-0031-82) and 1 μg/ml of anti-CD28 (clone 37.51, eBioscience, 16-0281-82) antibodies. Epithelial and mesenchymal cSCC cancer cells were added to CD8⁺ T cells at 1:1 ratio on day 2 after T cell activation, together with 25 μg/ml of anti-PD-L1 (clone 10 F.9G2), anti-CTLA-4 (clone UC10-4F10-11), or anti-TIGIT (clone 1G9) antibodies. After 2 days of co-culture, flow cytometry assays were performed to quantify the Violet dye dilution as a proxy for T cell activity and proliferation, as well as the expression of CD69, CD25, and GzmB as a proxy for CTL activity, as described above. Viability was assessed with 7-AAD Viability Staining Solution (1:100, Biolegend, 420403) or LIVE/DEAD™ Fixable Violet Dead Cell Stain Kit (1:1000, Life Technologies, L34963).

### Histology, immunofluorescence, and immunohistochemistry assays

Mouse cSCC samples were fixed with 4% formaldehyde (PanReac, 252931) overnight at 4 °C, paraffin-embedded, and sectioned at 4 μm. Paraffin cSCC, HNSCC, and melanoma sections from patients who had received anti-PD-1/PD-L1 therapy were provided by the Pathology Unit and Biobank of the Bellvitge University Hospital (Barcelona, Spain) and the IdiPAZ Biobank (Madrid, Spain). Paraffin-embedded tumor sections were deparaffinized with xylene and rehydrated with decreasing concentrations of ethanol, and antigens were retrieved in 10 mM sodium citrate (pH 6.0) or 10 mM TRIS/EDTA (pH 9.0), depending on the specifications of the primary antibody supplier.

For immunohistochemistry detection, endogenous peroxidase activity was quenched with 3% hydrogen peroxidase (Millipore, 1.07210.1000) for 10 min at room temperature. Tumor sections were then blocked with 5% horse serum in TBS for 2-3 h at room temperature and incubated overnight at 4 °C with the following primary antibodies diluted in TBS with 0.1% Tween20 and 3% horse serum: *m/h*CD163 (1:50, Abcam, ab182422), *m*CD8α (1:50, Cell Signaling, 98941), *m*FoxP3 (1:50, Cell Signaling, 12653), *h*FoxP3 (1:50, Cell Signaling, 98377), *h*Granzyme B (1:100, Abcam, ab4059). The next day, tumor sections were incubated with a secondary anti-rabbit Envision

System-HRP antibody (Dako, K4003) for 1 h at room temperature, followed by the DAB developing system (Dako, K3468). Finally, samples were counterstained with hematoxylin, mounted with DPX medium (Sigma, 06522), and visualized under light microscopy (Nikon Eclipse 80i and ZEISS Axioscan 7 Scanner).

For immunofluorescence detection, tumor sections were blocked with 5% horse serum in TBS for 2–3 h at room temperature and incubated overnight at 4 °C with the following primary antibodies diluted in TBS with 0.1% Tween20 and 3% horse serum: *h*CD155 (1:100, Cell Signaling, 13544), *h*CD8α (1:50, Abcam, ab17147), *h*CD80 (1:150, Abcam, ab254579), *h*E-cadherin (1:100, BD Bioscience, 610182), *h*LAG-3 (1:100, Cell Signaling, 15372), *h*PD-1 (1:100, Abcam, ab137132), *h*TIGIT (1:100, Cell Signaling, 99567), *h*TIM-3 (1:100, Cell Signaling, 45208), *h*Vimentin (1:100, Abcam, ab45939). Slides were then incubated with secondary antibodies conjugated with Alexa 568 or 647 (Invitrogen) for 1 h at room temperature, stained with DAPI (1:5000, Invitrogen, D3571) for 15 min, mounted with Vectashield medium (Vector Laboratories, H-1000-10), and imaged under a confocal microscope (Leica TCS SP5 and ZEISS LSM 980 with Airyscan2).

To visualize GFP directly, mouse cSCC samples were fixed with 4% formaldehyde for 30 min, washed with PBS for 30 min, embedded in optimal cutting temperature (OCT) compound (Sakura Finetek, 4583), and sectioned at 4 μm. Cryosections were permeabilized in TBS with 0.1% Triton X-100 (Sigma, 9036-19-5) for 15 min, blocked with 5% horse serum in TBS for 2–3 h at room temperature, and incubated overnight at 4 °C with the following primary antibodies diluted in TBS with 0.1% Tween20 and 3% horse serum: *m*CD68 (1:200, Abcam, ab125212) and *m*Gr-1 (1:200, R&D Systems, MAB1037). Samples were then incubated with secondary antibodies conjugated with Alexa 546 or 568 (Invitrogen) for 1 h at room temperature, stained with DAPI (1:5000, Invitrogen, D3571) for 15 min, mounted with Vectashield medium, and imaged under a Leica TCS SP5 confocal microscope.

The antibodies used for immunohistochemistry and immunofluorescence assays are listed in Supplementary Table 4. All images arising from this part of the work were analyzed using ImageJ v1.54d and ZEN Blue 3.6 software. For analyses on patient tumor sections, hematoxylin and eosin (H/E) samples were scanned to obtain an overview of the tissue structure (Supplementary Fig. 10), and pathology training was performed to differentiate cancer cells from stromal cells by nuclear atypia (crowded, pleomorphic, and often large and hyperchromatic) and cell size. Immunofluorescence samples were then scanned with a confocal microscope to visualize the distribution of the studied markers, and a significant number of images at 40x magnification (at least five images) were captured to accurately represent the intratumor heterogeneity of each sample. The expression of the analyzed markers within cancer cells was quantified from the magnified images (Supplementary Fig. 10).

### RNA purification and qRT-PCR

cDNA amplification by pico profiling was performed in the Functional Genomics Core of the Institute for Research in Biomedicine (IRB, Barcelona, Spain), as previously described[104]. qRT-PCR reactions were then performed on an Applied QuantStudio5 machine, mixing 4 ng of the total cDNA with specific gene primers and SYBR Green PCR Master Mix (Thermo Fisher, 4309155). Analyses were carried out in triplicate. mRNA expression was normalized relative to the expression of *Gapdh* and *Ppia* in all samples. mRNA levels are shown as the $\log_2$ fold change, in which the mean mRNA levels relative to two housekeeping genes were calculated. Gene-specific primers are listed in Supplementary Table 5. Data were analyzed with SDS 2.3 software.

### Statistical analysis and reproducibility

All statistical analyses and graphs were conducted using GraphPad Prism v8.0.1 and R software v4.0.5. Statistical tests used are described in each of the panels of the figure legends. Unpaired two-sided Student's *t-*

test for comparisons of two groups and one-way ANOVA with Tukey's or Dunnett's tests for comparisons of multiple groups were applied to continuous normal data. Non-parametric Kruskal–Wallis test for comparisons of multiple groups was used when data distribution failed normality tests. Significant differences in tumor growth over time were calculated by a two-way ANOVA test. In the graphs, data are represented as the mean ± standard deviation (SD) or standard error of the mean (SEM). Exact $p$ values are indicated in the figures. All experiments were replicated in at least three independent biological replicates unless otherwise indicated. The number of independent biological replicates for each experiment is indicated in the figure legends.

No statistical method was used to predetermine sample size in in vitro and in vivo experiments, but group sizes were determined based on the results of preliminary experiments. Group allocation was performed in a randomized fashion. The investigators were not blinded to allocation during outcome assessment.

The association between Ecad/Vim/CD80/CD155 variables and the efficacy of anti-PD-1/PD-L1 therapy was shown graphically by plotting the response/relapse variable and the percentage of Ecad[+], Vim[+], CD80[+], CD155[+], Ecad[−]Vim[+], Ecad[−]CD80[+] and Ecad[−]CD155[+] cancer cells, and then fitting the smooth curve obtained by logistic regression analysis (Supplementary Figs. 8 and 9). The R package ThresholdROC was used to calculate the best cut-off point for each variable (the value that best separates responder from non-responder cSCC and HNSCC patients, and non-relapsed from relapsed melanoma patients), according to Youden's index maximization criterion. Diagnostic accuracy measures (sensitivity and specificity) associated with that cut-off are shown (Supplementary Figs. 8 and 9).

Cox proportional hazards models were used to study the association between the percentage of Ecad[+], Vim[+], CD80[+], CD155[+], Ecad[−]Vim[+], Ecad[−]CD80[+], and Ecad[−]CD155[+] cancer cells and the outcome time to progression for cSCC and HNSCC cohorts, or time to relapse for the melanoma cohort. Results are reported as the hazard ratio (HR) ± 95% confidence interval (CI), and illustrated with a forest plot. The proportionality of risks in the Cox models was verified using Schoenfeld residuals.

### Reporting summary
Further information on research design is available in the Nature Portfolio Reporting Summary linked to this article.

## Data availability
Data are available within the Article, Supplementary Information or Source Data file. Source data are provided with this paper.

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

## Acknowledgements

L.L.-S. received an IDIBELL Fellowship and an EMBO Short-Term Fellowship (number 7192); M.L.-C. was supported by the FI program (2019FI_B_00265) of the Secretariat for Universities and Research of the Department of Business and Knowledge of the Government of Catalonia, with the support of the European Social Fund (ESF) "ESF, Investing in your future"; V.d.S.-D. was funded by a Spanish Ministry of Science and Innovation Fellowship. The research of P.M.'s group is supported by the Spanish Ministry of Economy and Competitiveness MINECO (SAF2017-84976R and PID2020-113495RB-I00; co-funded by the FEDER funds/European Regional Development Fund (ERDF) - A way to build Europe), Fundació Vallformosa and GESCO Family, and by the Catalan Department of Health (CERCA, Generalitat de Catalunya, 2017SGR595, 2021 SGR00769). J.M.-L. and P.M. acknowledge funding from Beca GEM (Grupo Español Multidisciplinar de Melanoma). We also thank the patients who enrolled in this study for their participation; the Tumor Biobank of the Bellvitge University Hospital and the IdiPAZ Biobank for their help collecting patients' tumor specimens; and the staff at the IDIBELL biostatistics, optical microscopy, and animal facilities, and at the UB/PCB flow cytometry facility for assistance. Figures 1c, 3a and 4a were created with BioRender.com, released under a Creative Commons Attribution-NonCommercial-NoDerivs 4.0 International license.

## Author contributions

L.L.-S. and P.M. conceptualized, designed, and supervised this work. L.L.-S. performed most of the experiments and data analysis, contributed to the interpretation and discussion of the results, and co-wrote the manuscript. M.L.-C. performed a substantial number of experiments and contributed to the discussion and writing. V.d.S.-D. helped generate the mouse cSCC progression model and contributed to the interpretation and formal analysis. M.H.A. helped carry out the mouse immunohistochemistry experiments and the formal analysis. S.L. contributed to the sample collection, interpretation, and formal analysis. R.M.P. and J.O.B. contributed to the sample collection, resources, and interpretation. E.G.-S. contributed to the formal analysis and interpretation. M.E. contributed to the resources and formal analysis. F.V. contributed to the formal analysis and interpretation. E.E. contributed to the sample collection and formal analysis. M.O. contributed to the sample collection, interpretation, and formal analysis. J.M.P. contributed to the resources and formal analysis. J.M.-L. secured funding and contributed to the sample collection, interpretation, and formal analysis. P.M. secured

funding, contributed to the interpretation of the results, and co-wrote the manuscript.

## Competing interests

M.E. declares research grants from Ferrer International and Incyte, and consulting fees from Quimatryx outside of this study. M.O. declares consulting or advisory arrangements with Merck, MSD and Transgene; research support (clinical trials) from Merck and Roche; that the institution receives clinical trial support from AbbVie, Ayala Pharmaceutical, MSD, ALX Oncology, Debiopharm International, Merck, ISA Pharmaceuticals, Roche Pharmaceuticals, Boehringer Ingelheim, Seagen, Gilead; and travel accommodation expenses from MSD, Merck. J.M.P. declares consulting or advisory roles for Janssen Oncology, Astellas Pharma, VCN Biosciences, Clovis Oncology, Roche/Genentech, Bristol-Myers Squibb, Merck Sharp & Dohme, BeiGene; research funding from Bristol-Myers Squibb, AstraZeneca/MedImmune, Merck Sharp & Dohme, Pfizer/EMD Serono, Incyte, Janssen Oncology; and travel, accommodation and other expenses from Janssen Oncology, Roche, Bristol-Myers Squibb. J.M.-L. has received lecturing fees from Astellas, Bristol-Myers Squibb, MSD, Novartis, Pierre Fabre, Pfizer, Roche, and Sanofi; advisory fees from Bristol-Myers Squibb, Highlight Therapeutics, Novartis, Pierre Fabre, Roche, Sanofi; and travel grants from Bristol-Myers Squibb, Merck, MSD, Novartis, Pierre Fabre, Pfizer, Roche, Ipsen. All other authors declare no potential competing interests.
