## [Peer Review File · Nature Communications]

Cancer Cell Plasticity Defines Response to Immunotherapy in Cutaneous Squamous Cell CarcinomaREVIEWERS' COMMENTS:

Reviewer #1 (Remarks to the Author): with expertise in skin cancer, immunology

In their study, Lorenzo-Sanz et al used tumor cell lines derived from a chemo-induced cutaneous carcinoma generated in transgenic K14-HPV16 mice with either an epithelial phenotype or a more aggressive mesenchymal phenotype to validate a mouse model that recapitulates the heterogeneity of human cancer cells from cutaneous squamous cell carcinoma. Using these tumor lines, they generated syngenic tumors by subcutaneous injection in mice. The characterization of these tumors revealed that the epithelial tumors were expressing the EPCAM and PD-L1 molecules contrary to the mesenchymal tumors. These latter tumors were expressing the stromal marker vimentin as well as the CD155 and the CD80 molecules. They showed that the tumor outgrowth was reduced under PD-L1 immunotherapy in mice bearing epithelial tumors but not mesenchymal tumors. Inversely, TIGIT- or CTLA4-based immunotherapies targeting CD155 and CD80, respectively reduced the outgrowth of mesenchymal tumors but not epithelial tumors. Then, they showed that an immunotherapy based on the combined administration of blocking anti-PD-L1 and anti-TIGIT antibodies in mice bearing mixed epithelial/mesenchymal tumors has a greater impact on tumor growth than monotherapies. Finally, they checked the relevance of their findings using baseline tumors from few patients that received an anti-PD1 immunotherapy (7 responders vs 7 non-responders). They proposed that the expression of mesenchymal markers may explain the resistance to anti-PD1 immunotherapy.

This study is a proof-of-concept study using a relevant mouse model of cutaneous squamous cell carcinoma and showing the anti-tumoral potential of a combined immunotherapy targeting both PD-L1/PD-1 and CD155/TIGIT immune checkpoint inhibitory pathways. In addition, the proposed tumor model could be useful for the discovery and/or validation of novel therapeutic targets for cSCC. Nevertheless, the originality of the study is limited as it extends previous pre-clinical and clinical studies in other cancers (as recently reviewed in Chu et al., *Mol Cancer* 2023 Jun 8;22(1):93).

Several points may help to improve the soundness of the data:

Fig. 2/4/7: As the data generated on human tumor sections are important for the conclusions of the study, the reviewer recommends to show annotated HES stainings to have a structural view of the tumor section. One slide before or after the one used for the

immunofluorescence could be used. In addition, if possible for the IF stainings, whole scans of tissue sections with boxes to indicate the fields used for the quantification should be indicated.

Fig. 3: Cancer cells are co-cultured with CTV-labelled splenic CD8 T cells. The expression of PD-1, TIGIT, CD28 and CTLA-4 by these CD8 T cells should be shown at D0 and D2. In addition, it is known that CD69 is an early marker of activation. It is surprising that roughly the same abundance of CD69+, CD25+ and Gzm+ cells (e.g. around 40% for CD8 T cells alone) were found. To make the data clearer, the reviewer suggestion is to show bidimensional plot of the expression of each marker (CD69/CD25/GzmB) against CTV dilution. In addition, the proliferation should be quantified. As the data are from several independent experiments, Standard Error of Mean should be used instead of Standard deviation.

Sup Fig. 1 and sup Fig. 3: Complete gating strategies used should be shown for a representative file or a concatenated file.

Sup. Fig. 1: As the data show that exhausted CD8 T cells mainly expressed PD-1 and exhausted NK cells the TIGIT marker, the infiltration of human cSCC by these populations should be also shown.

Fig. 5 : The authors should explain why they used anti-PD-L1 treatment rather than anti-PD1 in their mouse experiments. The two types of therapies can give different results and as mentioned by the authors, the expression of PD-L1 is not always a good predictive biomarker of response to anti-PD-1 treatment. Indeed, the patients used in the study (Fig. 7) were treated with an anti-PD-1 antibody.

Fig. 6: Identifying the cell types involved in the response to the combined immunotherapy would be informative and could help to better define the patients to treat.

Fig.7: PD-L1 marker should be added as a reference. Have the biomarkers proposed a better predictive value than PD-L-1 marker?

Reviewer #2 (Remarks to the Author): with expertise in skin cancer

Manuscript review Lorenzo-Sanz et al. Nature Communications

Key findings: Cutaneous squamous cell carcinoma is very heterogeneous in the degree of

tumor cells that exhibit mesenchymal characteristics. Response among cSCC patients to immune checkpoint inhibitors is also highly variable with few patients having sustained response to any one treatment. The authors demonstrate decreased expression of epithelial marker Epcam and Ecad and increased expression of mesenchymal marker Vimentin in poorly differentiated compared to well differentiated SCC, suggestive of epithelial to mesenchymal transition. Epcam+ and Epcam- cells have different checkpoint inhibitor ligand expression profiles and therefore different sensitivities to immune checkpoint blockade.

Quality of data: The first two figures do not adequately demonstrate EMT (see major comments below) and there is a general lack of correlative human clinical data. The latter figures mostly meet the claims that are made (see minor comments below).

Novelty and Citations of Previous Work: The authors fail to cite most of the relevant literature regarding EMT and EMT in cSCC in particular. EMT is a well established phenomenon in SCC and it has been shown to relate to tumor invasive potential (primary literature: Ji et al. 2020-PMID:32579974, Mizrahi et al. 2018-28925390, Yokoyama et al. 2001-11120485, Barrette et al. 2014-24628329, Toll et al. 2013-23928229; reviews: Thiery 2002-12189386, Fernandez-Figueras and Puig 2020-32350682). Additionally, other groups have previously demonstrated a relationship between EMT and immune checkpoint ligand expression (Shrestha et al. 2021-33925488). In addition, the technical approach uses fairly standard assays and don't include a significant technical novelty.

Overall, these studies raise the question of overall technical and conceptual novelty of the data in this manuscript and suggest it would be more appropriate for a specialty journal.

Major comments:

- The claim of epithelial/mesenchymal transition isn't fully supported by these data and has been previously demonstrated in SCC (though they haven't cited much of the EMT literature). Making that claim would require lineage tracing in the mouse model system. Though they do use a GFP label for cells isolated from mouse tumors, it is unclear to me how their GFP labeling works- are there mixed epithelial and mesenchymal cells in the

cultures that they apply the lentiviral plasmid to? If there are GFP labeled non-tumor mesenchymal cells one would not be able to distinguish expansion of mesenchymal non-cancer cells from EMT among cancer cells. The markers included in the flow cytometry results and in the IF images are insufficient to determine the origins of the mesenchymal cells.

- Likewise, in the human data (Fig 1a) all of the cells here are being called tumor cells, but on what basis? I am unclear of the gating that they have already applied to exclude non-tumor stromal cells from their samples. The methods section says that they have been depleted of red blood cells and endothelial cells, but otherwise are stromal cells included? I assume they are gated out based on the fact that there appear to be fibroblasts in the WD-SCC IF photos and there aren't any EPCAM- cells in the flow results, but I think the authors should include the details of how they gated.

- The strength of the paper is in the data they present in figures 3-7 in which they convincingly show that PDL1 is highly expressed among Epcam+ cells while CTLA4 and TIGIT ligands are more highly expressed among Epcam- cells. As a result, tumors with greater Epcam+ cell abundance are more sensitive to anti-PDL1 treatment, while tumors with greater Epcam- cell abundance are more sensitive to anti-CTLA4/anti-TIGIT treatment (Figure 5). Predictably, the proportion of Epcam+ cells is altered in tumors in response to these different therapies (Figure 6). And among human patients those that are responsive to anti-PD1 therapy have lower abundance of mesenchymal cells (Figure 7). While this is nicely shown, most of these conclusions have been shown previously in other systems.

Minor comments:

- I would like to see a more comprehensive panel of activation markers for the T cells in Figure 5 (like they showed in Figure 3f).

- When the authors claim that "anti-PD-L1 treatment was not able to reinvigorate the exhausted state of CD8+ T cells in mesenchymal mouse cSCC," I don't think they have sufficient data to claim reinvigoration especially in light of recent papers which demonstrate

that much of the effect of PD(L)1 treatments is due to the emergence of novel T cell clones (Yost et al. 2019).

- In Supplemental Figure 2f, I am not sure the differences in the CD68+ signal are supported by the photos so I am curious how they selected areas for quantification. This is also a concern for panels g-j.
- The authors claim that they have “show[n] elevated recruitment of immunosuppressive and exhausted immune cells,” is not fully supported- they can’t distinguish between recruitment and polarization of resident immune cells towards an immune-suppressive state.
- The details of the mouse model are not laid out- how do they achieve WD-, MD-, PD- SCC tumors in mice?
- Indicating Marker 1 and Marker 2 double positive cells as Marker1/Marker2 is confusing because it reads like a ratio.
- In Figure 2 (G4 especially), the IF images have so few markers and no landmarks, making it challenging to assess. Additional reference markers are required.

Reviewer #3 (Remarks to the Author): with expertise in skin cancer

The authors have performed a detailed immunophenotyping into epithelial, mesenchymal and hybrid cSCC subtypes. While the discovery of hybrid cSCC subtypes are not new, they are a relatively little studied cSCC subtype and the authors have brought new understanding into how these epithelial, mesenchymal and hybrid cSCC subtypes affect immune cell tumour microenvironment in both in vivo mouse models and in patient samples. Moreover, the authors explore IC ligand expression in the above subtypes and make the new finding that different epithelial/mesenchymal subtypes confer sensitivity/resistance to different Immune Checkpoint inhibitor (ICI) therapies. These results have translational relevance as they help us to understand how cSCC patients could be more effectively treated in the clinic.

Moreover, the authors show that a mixed population of epithelial/mesenchymal subtypes respond to monotherapy of ICI through expansion of the more resistant subtypes (potentially suggesting a more plastic adaptive role in this subtype?), thereby supporting the notion that mixed subtypes should be treated with combination therapy.

The epithelial and mesenchymal subtype categorisations and their link to cSCC progression is not new, and likewise the existence of hybrid epithelial/mesenchymal subtypes have been found, however fairly recently. Where the authors work adds novelty and value is in the deep immunophenotyping of the tumour microenvironment of these subtypes and the elegant cross-comparison between cSCC mouse models and using these to inform studies in patient samples. The implication for targeted therapy among these three phenotypes is also new. The work presented does support the overall claims, however, understandably sample size of patient samples is fairly limited. While the difference between patient samples is striking and in-line with the mouse models, to bolster the final claims about ICI treatment and cSCC subtypes, it would be strongly recommended if the authors could access some publicly available datasets (RNA-seq), if such a resource is available.

Overall, the writing is clear, the methodology is logical and sound and the interpretation reasonable. The sequential mouse model-to-patient data format is particularly compelling and pleasing. On the whole I recommend acceptance of the manuscript with some minor changes, which are required for quality control, or clarification of methods or statistics:

Data analysis/presentation:

1. The authors have not presented any of their gating strategies for flow in the supplementary materials. This is an absolute must for publication in Nature Communications.
2. I am not an immunologist or personally have extensive expertise in immunophenotyping so from my limited expertise I can say that the markers used seem appropriate and comprehensive for the interpretation extracted from the study. I would urge editors to deviate to other reviewers who are more proficient in this area however.
3. The authors have represented individual experiments/mice as individual dots on a graph, which is great. Please be clear if these are biological or technical replicates. For mouse I

suspect technical replicates within 1 biological replicate, this is ok but must be clear for the reader.

4. Be more considered of statistical tests used. There are many examples where $n < 10$ and the authors have used parametric tests. Consider non-parametric tests where the sample size is not appropriate to reasonably detect if the distribution is normal.

5. For tumour growth plots from mouse models, please state if all points represent cumulative growth from all tumours or if any were removed as outliers during the experiment.

6. For Figs 5 and 6, why are there different n-numbers for the %CD8+/CD4- within CD3+ and %GZMB+ within CD8+ versus total numbers of mouse tumours followed for tumour growth? Are some samples being discounted from the study? Please be clear about this.

7. For the Immune Checkpoint inhibitor tumour studies, I have never seen dosing begin at a palpable tumour size of 15mm^3 before. From all experience it is not possible to detect a tumour of this size accurately. Plus, in the methods the authors state the measurement is $3\text{mm} \times 3\text{mm}$ in size. This is not a 15mm^3 tumour. 3×3 would give you a 9mm^2 surface area. The authors could choose to report 3 measurements (height x width x length) and essentially calculate the volume of a cuboid, where $3 \times 3 \times 3$ would equal a volume of 27mm^3 . Or the authors could choose to input their measurements to calculate the volume of a sphere or hemisphere (depending on how they believe their tumours grow). Most usually $H \times W \times L$ is taken. Either way, such a tiny initial size is unusual and I have concerns for the accuracy of the onset of treatment. Especially if one subtype was inherently more stiff than another. Usually, people start dosing at around 75mm^3 . I would not advocate re-doing the mouse experiments on this basis, but rather being clearer about the size measurements and ultimately volume. Also presenting in the supplementary the time from cancer cell engraftment to dosing for each experiment.

8. Why is the Cox Regression analysis not performed on the Epcam-/ Vim+/ CD80+/ CD155+ signature as a whole in addition to individual genes?

9. Figure 7 would massively benefit from a Kaplan Meier plot. I realise this is not feasible with the limited patient samples available (7 per group), is there a possibility of using publicly accessible patient datasets to perform this analysis? It would be hugely compelling for the paper.

Text/context/interpretation:

1. I find it very intriguing in Fig 6 c, f, i, that there is a change in the proportion of the resistant/sensitive subtype depending on the treatment given, yet in fig 5 there is no change. My first-pass interpretation, perhaps overlooked by the authors, is that the mixed population is more plastic (ergo the title of the manuscript) and only the mixed population has the capacity for this plastic adaptation. Perhaps the authors could expand on this with a sentence or two in the discussion (if they likewise agree).

2. Lines 96-97 in regards to figure 1d-e:

“Conversely, EpCAMlow cancer cells from MD/PD-SCCs retained the expression of Krt14, Grhl1, Grhl2 and dNp63 epithelial markers,”

I find the above confusing as in the figures the bars go down in EpCAMlow cancer cells. Find a way to marry the text and the visual representation better, perhaps a better description is a gradient?

3. Lines 169-171:

“We corroborated in cSCC patient samples that the frequency of GzmB+ cells decreased and the frequencies of CD8+/PD-1+, CD8+/LAG-3+, CD8+/TIM-3+ and CD8+/TIGIT+ cells increased in mixed and mesenchymal patient cSCCs (Supplementary Fig. 4a-j)”

Agreed regarding the proportions of markers per CD8+ cells. But also from the pictures shown, the total number of CD8+ cells seems to peak in mixed samples. Is this a real thing or just the images shown, if it is real it deserves a mention.

4. Lines 193-194:

“The frequency of CD80+/Ecad+ and CD155+/Ecad+ cancer cells specifically increased in mixed patient cSCCs (Fig. 4e,f), suggesting that the expression of these IC ligands might be associated with the appearance of hybrid E/M cancer cells”

Careful with wording here, yes for CD155, there is absolutely a trend for expression only occurring upon appearance of hybrid cells. However for CD80, there are still 5-10% of epithelial cells that express CD80. Therefore perhaps...”suggesting that the upregulation of these IC ligands...” would be more appropriate.

5. Regarding Figure 4: Is there more heterogeneity in CD80 and CD155 expression in mixed

samples? With the caveat that IF staining is a semi-quantitative method.

Following minor revisions as stated I would strongly recommend publication of this manuscript in Nature Communications.

Reviewer #4 (Remarks to the Author): with expertise in cancer immunology

While immune checkpoint inhibitors (ICI) are used in a wide range of cancer types, not all patients can receive clinical benefits. Tumor cells evade antitumor immunity in a variety of ways, and there are needs for biomarkers to predict efficacies and therapies to overcome resistance. In cutaneous cell carcinoma (cSCC), anti-PD-1 inhibitor is used for advanced cases, but any combination therapies with ICIs have not been approved. The authors showed that the cSCC progression with mesenchymal differentiation features changed IC ligand profile of tumor cells in both a mouse model and clinical samples, and this change was also seen in anti-PD-1 non-responder cSCC as well. These results suggest that treatment with anti-PD-1 inhibitor in combination with anti-CTLA-4 inhibitor or anti-TIGIT inhibitor may give better outcomes in cSCC patients of any grade. Also, the degree of expression of these ligands can predict tumor progression and response to ICI, and this information is quite useful for cSCC patients. Although this manuscript is interesting with valuable data, there are several serious concerns.

Major concerns:

1. There are no mechanistic data on higher infiltration of lymphocytes in mixed and mesenchymal cSCCs and IC ligand profile changes during cSCC progression. I wonder why PD-L1 expression decreased in mesenchymal cSCCs even with higher infiltration of lymphocytes because PD-L1 is usually induced by IFN- γ from lymphocytes.
2. I have a question about CD80. CD80 is originally expressed in hematopoietic cells. I previously analyzed CD80/CD86 expression using CCLE datasets, showing that CD80 is seldom expressed in epithelial cancer cells.
3. Related to CD80, the authors should analyze CD28 expression during progression.
4. In Figure 3b-g, the authors used anti-CD3/CD28 mAbs for the stimulation. However, as this stimulation is non-specific, T cell activation can be non-specific for cancer cells. The

authors should conduct experiments in vivo instead of such questionable in vitro experiments.

5. CD155-TIGIT axis is also deeply involved with CD122/CD226. How were these molecules expressed in this mouse model?

Minor concerns:

1. Because Gal9 expression changed, the authors should investigate TIM-3 blockade, too.

2. In Figure 5, I want to know the treatment of mice bearing epithelial cSCCs with anti-CTLA-4 antibody. Although combination therapy with anti-PD-1 antibody and anti-CTLA-4 antibody has been used in other skin cancers, please indicate any reasons for choosing anti-TIGIT antibody, not anti-CTLA-4 antibody, for use with anti-PD-L1(PD-1) antibody.

3. Please provide more detailed information about in antibodies including Fc region. Particularly, ADCC activities should be considered in CTLA-4 antibodies.

4. TIGIT blockade is expected to be effective in lung cancer with high PD-L1 expression. However, the present data are somewhat conflicting. Please discuss.

5. EMT is known to occur in other skin cancers such as melanoma. It is desirable to verify whether the changes in IC ligand profile are specific to SCC or general.

First, we would like to thank the reviewers for their constructive and positive evaluation of our work. The reviewers' comments have been very helpful and have encouraged us to perform additional experiments to strengthen our findings. We believe that the changes made in the manuscript in response to the reviewer's suggestions have enhanced the clarity of the results and methodology. Below we detail how we have addressed the reviewer's concerns point-by-point:

Reviewer #1 (Remarks to the Author): with expertise in skin cancer, immunology

In their study, Lorenzo-Sanz et al. used tumor cell lines derived from a chemo-induced cutaneous carcinoma generated in transgenic K14-HPV16 mice with either an epithelial phenotype or a more aggressive mesenchymal phenotype to validate a mouse model that recapitulates the heterogeneity of human cancer cells from cutaneous squamous cell carcinoma. Using these tumor lines, they generated syngeneic tumors by subcutaneous injection in mice. The characterization of these tumors revealed that the epithelial tumors were expressing the EPCAM and PD-L1 molecules contrary to the mesenchymal tumors. These latter tumors were expressing the stromal marker Vimentin as well as the CD155 and the CD80 molecules. They showed that the tumor outgrowth was reduced under PD-L1 immunotherapy in mice bearing epithelial tumors but not mesenchymal tumors. Inversely, TIGIT- or CTLA-4-based immunotherapies targeting CD155 and CD80, respectively reduced the outgrowth of mesenchymal tumors but not epithelial tumors. Then, they showed that an immunotherapy based on the combined administration of blocking anti-PD-L1 and anti-TIGIT antibodies in mice bearing mixed epithelial/mesenchymal tumors has a greater impact on tumor growth than monotherapies. Finally, they checked the relevance of their findings using baseline tumors from few patients that received an anti-PD1 immunotherapy (7 responders vs. 7 non-responders). They proposed that the expression of mesenchymal markers may explain the resistance to anti-PD-1 immunotherapy.

This study is a proof-of-concept study using a relevant mouse model of cutaneous squamous cell carcinoma and showing the anti-tumoral potential of a combined immunotherapy targeting both PD-L1/PD-1 and CD155/TIGIT immune checkpoint inhibitory pathways. In addition, the proposed tumor model could be useful for the discovery and/or validation of novel therapeutic targets for cSCC. Nevertheless, the originality of the study is limited as it extends previous pre-clinical and clinical studies in other cancers (as recently reviewed in Chu et al., Mol Cancer 2023).

We would like to thank the positive evaluation of the reviewer regarding the use of our mouse cSCC model, which recapitulates the cancer cell heterogeneity found in patient cSCCs, to study what mechanisms might be involved in anti-PD-1/PD-L1 resistance and to assess the response to new therapeutic strategies. However, we disagree with the reviewer's statement about the "limited originality" of our study compared with previous preclinical and clinical studies. To clarify our response, we highlight below the main original contributions of our study. Although several

preclinical studies have demonstrated the potential benefits of dual anti-PD-1/PD-L1 and anti-TIGIT inhibition *vs.* monotherapy in several cancer types¹⁻⁷, **the mechanisms involved in immune checkpoint inhibitor (ICI) resistance have not yet been clearly identified**. In this regard, a notable proportion of patients still remain unresponsive or develop resistance to immune checkpoint blockade (ICB) therapies, and **no biomarkers currently precisely predict which patients will benefit from monotherapy *vs.* combined strategies**⁸⁻¹².

For this reason, the strength of our study is the finding that **epithelial-mesenchymal cancer cell plasticity is a mechanism of resistance to ICIs in cSCCs**. We have identified that cSCC cancer cells induce a differential IC ligand repertoire depending on their epithelial/hybrid/mesenchymal state, which plays a critical role in defining the response to monotherapy *vs.* combined ICB therapies. Therefore, our findings indicate that **cancer cell heterogeneity should be considered for the selection of ICB therapies** to ensure better efficacies and minimize adverse effects and toxicities. In addition, the identification of the **biomarkers** described in this manuscript could pave the way to **stratify patients into monotherapy *vs.* combined ICI therapies based on cancer cell features**. To the best of our knowledge, these findings have not been previously demonstrated in cSCCs or other tumor types.

Several points may help to improve the soundness of the data:

- Fig. 2/4/7: As the data generated on human tumor sections are important for the conclusions of the study, the reviewer recommends showing annotated H/E staining to have a structural view of the tumor section. One slide before or after the one used for the IF could be used. In addition, if possible, for the IF staining, whole scans of tissue sections with boxes to indicate the fields used for the quantification should be indicated.

Pathological review and annotation of all patient tumor tissues was performed at the Pathology Service of the Bellvitge University Hospital as part of the standard clinical care by an expert pathologist (Dr. Rosa M. Penin, co-author of this manuscript). Due to the large number of patient samples used in this study, we have included two epithelial (moderately differentiated subtype, MD: H49 and H41 tumors), two mixed (moderately/poorly differentiated subtype, MD/PD: H34 and H48 tumors) and two mesenchymal cSCCs (poorly differentiated/spindle subtype, PD/S, H11 and H15 tumors) in the Fig. Rev1 a-c (corresponding to the tumors shown in Fig. 1g-k). For analyses on patient tumor sections, we first scanned hematoxylin and eosin (H/E) samples to obtain an overview of the tissue structure, and then performed training at the Pathology Service of the Bellvitge University Hospital to differentiate cancer cells from stromal cells by nuclear atypia (crowded, pleomorphic, and often large and hyperchromatic) and cell size. Immunofluorescence samples were scanned with a confocal microscope to visualize the distribution of the studied markers (E-cadherin, Vimentin, CD80 and CD155).

a Epithelial patient cSCCs

H49

H41

Ecad Vim DAPI

b Mixed patient cSCCs

H34

H48

Ecad Vim DAPI

C Mesenchymal patient cSCCs

Fig. Rev1. Epithelial, hybrid E/M and mesenchymal cancer cells are detected in cSCC patient samples. a-c, Representative hematoxylin/eosin (H/E) and immunofluorescence images of Ecad⁺ (red), Vim⁺ (green) and DAPI (nuclear) staining in (a) epithelial, (b) mixed and (c) mesenchymal patient cSCCs (n = 2 per group). Boxed images enlarged to the right of each cSCC. Scale bar, 1000 μ m (main images), 100 μ m (magnified images).

Following pathology guidelines, a significant number of 40x magnified images (at least 10-20 images) were captured to accurately represent the intratumor heterogeneity of the sample. The expression of the analyzed markers within cancer cells was quantified from the magnified images. An improved explanation of this strategy has been included in the “Histology, immunofluorescence and immunohistochemistry assays” section of the methods (lines 606-613).

• Fig. 3: Cancer cells are co-cultured with CTV-labelled splenic CD8 T cells. The expression of PD-1, TIGIT, CD28 and CTLA-4 by these CD8 T cells should be shown at D0 and D2.

We have performed new *in vitro* experiments in which the expression of PD-1, CTLA-4, TIGIT, CD28 and CD226 by CD8⁺ T cells has been evaluated after their isolation from the spleen (D0), and after 48h (D2) and 96h (D4) of CD3/CD28 activation. Due to space limitations, the PD-1, CTLA-4 and TIGIT panels are included in the new Fig. 3c-e, and the CD28 and CD226 panels are shown to reviewers in Fig. Rev2 a,b. We have confirmed that PD-1, CTLA-4 and TIGIT

receptors, which are the targets of the antibodies used to block the different IC pathways in these assays, are expressed by CD8⁺ T cells after 2 and 4 days of CD3/CD28 activation.

Fig. Rev2

Fig. Rev2. CD28, CD226, CD69, CD25 and GzmB expression within CD8⁺ T cells increases after 2 and 4 days of *in vitro* CD3/CD28 activation. a-e, Percentage (mean \pm SEM) of (a) CD28⁺, (b) CD226⁺, (c) CD69⁺, (d) CD25⁺ and (e) GzmB⁺ cells within CD8⁺ T cells isolated from the spleens of mice bearing epithelial (Epit.) or mesenchymal (Mes.) cSCCs on days 0, 2 and 4 of *in vitro* culture. Each dot represents one independent experiment. Significant differences determined by one-way ANOVA with Dunnett's test; ** p < 0.0001. f, Representative bidimensional plots showing CD69 and CD25 expression vs. CellTrace Violet dilution within CD8⁺ T cells on days 0, 2 and 4 of *in vitro* culture.**

• Fig. 3: In addition, it is known that CD69 is an early marker of activation. It is surprising that roughly the same abundance of CD69⁺, CD25⁺ and GzmB⁺ cells (e.g., around 40% for CD8 T cells alone) were found. To make the data clearer, the reviewer suggestion is to show bidimensional plot of the expression of each marker (CD69/CD25/GzmB) against CTV dilution. In addition, the proliferation should be quantified. As the data are from several independent experiments, SEM should be used instead of SD.

As suggested by the reviewer, we have included representative bidimensional plots of CD69 and CD25 expression against CTV dilution on days 0, 2 and 4 in the Fig. Rev2f. At this stage, we cannot show similar analysis for GzmB because it was incompatible to use the CellTrace Violet

dye with the LIVE/DEAD Fixable Violet Dead Cell Stain Kit for intracellular flow cytometry analysis, as they use the same fluorescent label. In addition, and to complement the previous set of experiments, we have analyzed in the new *in vitro* experiments the expression of CD69, CD25 and GzmB by CD8⁺ T cells after their isolation from the spleen (D0), and after 48h (D2) and 96h (D4) of CD3/CD28 activation (Fig. Rev2 c-e). In accordance with the reviewer, we have identified that whereas roughly the same abundance of CD69⁺, CD25⁺ and GzmB⁺ CD8⁺ T cells are found on day 4 (new Fig. 3g,h), CD69 expression is induced earlier than CD25 and GzmB expression (shown to reviewers in Fig. Rev2 c-f).

We have also included in the new Fig. 3g,h the percentage of proliferative CD8⁺ T cells as monitored by flow cytometry quantification of Violet dye dilution when co-cultured with (g) epithelial or (h) mesenchymal cancer cells, with or without PD-L1, CTLA-4 and TIGIT-blocking antibodies. Finally, we have plotted all data in the new Fig. 3 and Fig. Rev2 as the mean \pm SEM instead of the SD, since the data are from several independent experiments, acknowledging the reviewer's suggestion. This information has been modified in the corresponding figure legends.

- Suppl. Fig. 1 and Suppl. Fig. 3: Complete gating strategies used should be shown for a representative file or a concatenated file.

Immunophenotype characterization during mouse cSCC progression was previously included in Supplementary Fig. 1 (CTLs and NK cells) and 3 (immunosuppressive immune cells), and in this new revised version of the manuscript it is all grouped in Supplementary Fig. 1. Complete flow cytometry gating strategies for cancer and immune cell populations are shown in the new Supplementary Fig. 9.

- Suppl. Fig. 1: As the data show that exhausted CD8 T cells mainly expressed PD-1 and exhausted NK cells the TIGIT marker, the infiltration of human cSCCs by these populations should be also shown.

After revisiting PD-1 and TIGIT expression by exhausted CD8⁺ T cells and NK cells following the reviewer's suggestion, we would like to emphasize that our mouse results show the percentage of co-expression of IC receptor pairs, which is a well-established feature of the exhausted phenotype^{13,14}, rather than monitoring the expression of single IC receptors. Therefore, our mouse data do not indicate that “*exhausted CD8⁺ T cells mainly expressed PD-1 and exhausted NK cells the TIGIT marker*”. In particular, we show that the percentage of CTLs and NK cells co-expressing PD-1 along with LAG-3, TIM-3, CTLA-4 and TIGIT increase during mouse cSCC progression (Supplementary Fig. 1e-h,m-o).

Given the importance of knowing the exhausted state of these effector immune populations in patient cSCCs, we already included in the previous version of manuscript the frequency of

GzmB⁺, CD8⁺PD-1⁺, CD8⁺LAG-3⁺, CD8⁺TIM-3⁺ and CD8⁺TIGIT⁺ cells in epithelial, mixed and mesenchymal patient cSCCs (Supplementary Fig. 3a-j). As suggested by the reviewer, we have performed new IF assays to identify exhausted NK cells in patient cSCCs. However, although we have tested several commercially available antibodies, we have not obtained specific labelling of NK cells with any of tested reagents. In order to overcome this situation, we consulted several colleagues with expertise in immuno-oncology who confirmed the challenging task of identifying NK cells by immunofluorescence assays on tissue sections. As per your reference, we list below the antibodies tested: anti-human CD56 (NCAM) antibody (Biolegend, 380702), anti-human NCAM1 (CD56) antibody (Cell Signaling, 3576S) and anti-human NKp46/NCR1 antibody (R&D Systems, AF1850).

- Fig. 5: The authors should explain why they used anti-PD-L1 treatment rather than anti-PD-1 in their mouse experiments. The two types of therapies can give different results and as mentioned by the authors, the expression of PD-L1 is not always a good predictive biomarker of response to anti-PD-1. Indeed, the patients used in the study (Fig. 7) were treated with an anti-PD-1 antibody.

We appreciate the appropriate remark made by the reviewer. At the beginning of these studies, anti-PD-1 therapy was not yet approved for the treatment of cSCC patients (Cemiplimab FDA approval in 2018 and Pembrolizumab FDA approval in 2020). Since we observed an elevated PD-L1 expression in epithelial but not in mesenchymal cancer cells (Fig. 2a), we decided to use the anti-PD-L1 antibody for *in vivo* experiments. As suggested by the reviewer, we have compared the response of epithelial and mesenchymal cSCCs to anti-PD-L1 and anti-PD-1 therapies. As observed in the new figures, epithelial cSCCs respond similarly to anti-PD-L1 and anti-PD-1 therapies (new Fig. 4b and Supplementary Fig. 4a), and mesenchymal cSCCs are resistant to both treatments (new Fig. 5a and Supplementary Fig. 5a). In addition, epithelial cSCCs treated with both anti-PD-L1 and anti-PD-1 antibodies show an increased frequency of active CD8⁺ T cells (new Fig. 4c,e and Supplementary Fig. 4b-e), and no changes were detected in the exhausted state of NK cells (new Fig. 4d,f and Supplementary Fig. 4f,g). Altogether, these results demonstrate that immune evasion of epithelial cSCCs relies on the PD-1/PD-L1 pathway as confirmed by the equal response to both antibodies, in contrast to mesenchymal cSCCs.

For the sake of transparency, in the revised version of the manuscript, we have replaced several previous *in vivo* data when new experiments have been performed to add new treatment groups and to analyze some additional immune markers requested by the reviewers (new Fig. 4, 5 and 6 corresponding to Fig.5 and 6 of the previous version). It was reassuring to note that updated datasets reproduce the previous data, confirming and strengthening our initial findings.

- Fig. 6: Identifying the cell types involved in the response to the combined immunotherapy would be informative and could help to better define the patients to treat.

Although the reviewer's concern is related to the combined therapy in mixed cSCCs (composed of epithelial and mesenchymal cancer cells), we considered that it would be better to address which cell types are involved in ICI response in tumors composed only of epithelial or mesenchymal cancer cells, as these tumors would allow us to unravel whether different immune populations are mediating the response based on their cancer cell features. To explore the extent to which anti-PD-L1, anti-CTLA-4 and anti-TIGIT effects against epithelial and mesenchymal cancer cells are mediated by CTLs or NK cells, we performed CD8 and NK cell depletion experiments in epithelial and mesenchymal cSCCs. These new experiments have been included in the revised version of the manuscript (new Fig. 4b-f, 5f-o and Supplementary Fig. 4a-g, 5h-n). Our results show that anti-PD-L1 response is CD8⁺ T cell-dependent rather than relying on NK cells in epithelial cSCCs (new Fig. 4b-f and Supplementary Fig. 4a-g). Furthermore, the delayed tumor growth conferred by anti-CTLA-4 and anti-TIGIT therapies in mesenchymal cSCCs is partially lost after CD8 and NK cell depletion, indicating that these responses are mediated by both CTLs and NK cells (new Fig. 5f-m and Supplementary Fig. 5h-n). Given that anti-PD-L1, anti-CTLA-4 and anti-TIGIT therapies also reverse the exhausted state of both CD8⁺ T cells and NK cells in mixed cSCCs (new Fig. 6 and Supplementary Fig. 6), we suggest that both populations could play a key role in the response of cSCCs enriched in epithelial, hybrid E/M and mesenchymal cancer cells (mixed cSCCs), even though we have not performed depletion experiments in this context.

- Fig. 7: PD-L1 marker should be added as a reference. Have the biomarkers proposed a better predictive value than PD-L1 marker?

To tackle this issue, we analyzed PD-L1 expression in paraffin cSCC sections from patients who had received anti-PD-1 therapy at the Pathology Service of the Bellvitge University Hospital (following the diagnostic protocol with 22C3 pharmDx, Agilent). We did not observe specific labeling in cancer or immune cells in these cSCC patient samples, whereas specific labeling was detected in the positive amygdala control processed in parallel in these assays (Fig. Rev3). In addition, we tested other commercially available antibodies (PD-L1 E1L3N, Cell Signaling, 13684) and, in the same way, we did not detect specific labeling. Therefore, in our hands, PD-L1 expression cannot be used as a predictive marker of anti-PD-1 response, highlighting the relevance of the biomarkers identified in this study as predictors of response to anti-PD-1 therapy. In line with these results, several studies have also shown that PD-L1 score is not an exclusionary predictive biomarker of the clinical benefit of anti-PD-1 blocking antibodies in locally advanced and metastatic cSCCs^{15,16}.

On the other hand, PD-L1 expression was monitored in HNSCC patient samples, as part of the standard clinical diagnosis (Supplementary Table 2). No correlation between PD-L1 expression

in pre-treatment HNSCC samples and response to anti-PD-1 therapy was detected in responder patients, as P4 and P5 patients with a PD-L1 CPS of 0 responded similarly than P1, P2, P3 and P6 patients with higher PD-L1 expression. Conversely, non-responder P7-P9 and P12 patients, with a high PD-L1 CPS score, showed no response to anti-PD-1 therapy. Importantly, the biomarkers described in this study showed a significant correlation with anti-PD-1 response.

Fig. Rev3. No specific PD-L1 labelling is observed in anti-PD-1 responder and non-responder cSCCs. Representative immunohistochemistry images of PD-L1 in anti-PD-1 responder (n = 7, P1-P7) and non-responder (n = 7, P8-P14) cSCCs. Scale bar, 1000 μ m. Amygdala tissue was used as a positive control.

Reviewer #2 (Remarks to the Author): with expertise in skin cancer

Key findings: Cutaneous squamous cell carcinoma is very heterogeneous in the degree of tumor cells that exhibit mesenchymal characteristics. Response among cSCC patients to immune checkpoint inhibitors is also highly variable with few patients having sustained response to any one treatment. The authors demonstrate decreased expression of epithelial marker Epcam and Ecad and increased expression of mesenchymal marker Vimentin in poorly differentiated compared to well differentiated SCC, suggestive of epithelial to mesenchymal transition. Epcam+ and Epcam- cells have different checkpoint inhibitor ligand expression profiles and therefore different sensitivities to immune checkpoint blockade.

Quality of data: The first two figures do not adequately demonstrate EMT (see major comments below) and there is a general lack of correlative human clinical data. The latter figures mostly meet the claims that are made (see minor comments below). Novelty and Citations of Previous Work: The authors fail to cite most of the relevant literature regarding EMT and EMT in cSCC in particular. EMT is a well-established phenomenon in SCC, and it has been shown to relate to tumor invasive potential (primary literature: Ji et al. 2020-PMID:32579974, Mizrahi et al. 2018-28925390, Yokoyama et al. 2001-11120485, Barrette et al. 2014-24628329, Toll et al. 2013-23928229; reviews: Thiery 2002-12189386, Fernandez-Figueras and Puig 2020-32350682). Additionally, other groups have previously demonstrated a relationship between EMT and IC ligand expression (Shrestha et al. 2021-33925488). In addition, the technical approach uses fairly standard assays and do not include a significant technical novelty. Overall, these studies raise the question of overall technical and conceptual novelty of the data in this manuscript and suggest it would be more appropriate for a specialty journal.

After evaluating the reviewer's concerns about the relationship between EMT and invasion or IC ligand expression, we believe that the main message of our study may not have been sufficiently clear in the manuscript. We agree with the reviewer that EMT is a well-established phenomenon in cSCC¹⁷⁻²¹ and, as mentioned in the original version of the manuscript, our laboratory has also contributed to this field^{22,23}. However, in this study, we have used a mouse cSCC model associated with epithelial/mesenchymal plasticity (EMP) to recapitulate the epithelial/hybrid/mesenchymal cancer cell features found in cSCC patient samples, but not in the context of providing more information on the relationship between EMT and cSCC progression, invasion, or metastasis. The use of our mouse cSCC model based on EMT-induced cancer cell plasticity is important to demonstrate the impact of cancer cell heterogeneity on TME composition and in the response/resistance to ICIs, which is the strength of our manuscript.

On the other hand, as mentioned in the discussion of the manuscript, several studies have reported that the acquisition of EMT-like properties induces the expression of PD-L1 by cancer cells²⁴⁻³⁰.

However, in this scenario, it is still not fully understood why the induction of the EMT program is associated with anti-PD-1/PD-L1 resistance, as observed by our group and other independent laboratories³¹. Other studies have also described a correlation between the EMT phenotype and the expression of IC molecules such as PD-1, PD-L1, PD-L2, CTLA-4, OX40L or CD80^{25,32}, but no impact on therapy response has been reported as a consequence of this altered IC ligand profile. Importantly, our *in vitro* and *in vivo* data demonstrate that cSCC cancer cells change the IC ligand profile according to their features, and inhibit the antitumor response of effector immune cells by using different IC pathways. For this reason, the strength of our study is that the dynamic IC ligand switch in an EMT-mediated process to block CD8⁺ and NK cell activity could be used as a novel therapeutic target for cSCCs, as it plays an important role in ICI resistance. Our findings have also prompted efforts to identify other biomarkers of anti-PD-1/PD-L1 response (Vimentin, CD80 and CD155), since TMB and PD-L1 score are not exclusionary predictive biomarkers of the clinical benefit of anti-PD-1-blocking antibodies in cSCCs and other tumor types^{8,9,15,16,33–35}.

Major comments:

- The claim of epithelial/mesenchymal transition is not fully supported by these data and has been previously demonstrated in SCC (though they have not cited much of the EMT literature). Making that claim would require lineage tracing in the mouse model system. Though they do use a GFP label for cells isolated from mouse tumors, it is unclear to me how their GFP labeling works- are there mixed epithelial and mesenchymal cells in the cultures that they apply the lentiviral plasmid to? If there are GFP labeled non-tumor mesenchymal cells one would not be able to distinguish expansion of mesenchymal non-cancer cells from EMT among cancer cells.

As suggested by the reviewer, we have included some additional relevant EMT citations in the introduction and discussion sections of the revised version of the manuscript (see revised text in lines 43 and 376). However, in this study, we have only used a mouse cSCC model associated with epithelial/mesenchymal plasticity (Fig. 1c-f) to recapitulate the epithelial, hybrid E/M and mesenchymal cancer cell features found in cSCC patient samples (Fig. 1g-k) and to assess the response to new therapeutic strategies (Fig. 4, 5 and 6), but not in the context of providing more information on the relationship between EMT and cSCC progression, invasion, or metastasis.

In addition, as suggested by the reviewer, we have included a more detailed explanation of how cSCCs have been generated in the revised results section (lines 79-92), and in the “Primary cSCC cancer cell cultures” and “Tumor-cell grafting and *in vivo* treatments” sections of the revised methods (lines 482-487 and 491-497), as well as the flow cytometry gating strategy to identify cancer cell populations in the new Supplementary Fig. 9. To characterize the dynamic changes of cancer cell features during mouse cSCC progression, we isolated $\alpha 6$ -integrin⁺CD45⁻ cancer cells from mouse WD-SCCs (full epithelial cancer cells), MD/PD-SCCs (plastic epithelial EpCAM⁺

and mesenchymal EpCAM⁻ cancer cells) and PD/S-SCCs (full mesenchymal cancer cells), which were previously generated by orthotopic serial engraftments²² (please see new Supplementary Fig. 9). Isolated cancer cells were then transduced with an MSCV-IRES-GFP lentivirus plasmid, thereby making it possible to identify them by the expression of GFP. Epithelial (>70% EpCAM⁺ cancer cells), mixed (10-70% EpCAM⁺ cancer cells) and mesenchymal cSCCs (<10% EpCAM⁺ cancer cells) were generated after sorting and engrafting GFP⁺ full epithelial cancer cells, epithelial GFP⁺EpCAM⁺ and mesenchymal GFP⁺EpCAM⁻ cancer cells, respectively, into immunocompetent syngeneic mice (please see new Supplementary Fig. 9). For this reason, the identification of epithelial and mesenchymal cancer cells in the generated cSCCs was performed within the GFP⁺CD45⁻ cancer cell compartment, and we use GFP expression as a tracer of E/M cancer cell plasticity. On the other hand, it is important to highlight that stromal (non-tumor mesenchymal cells) were never included in our studies, as cSCCs were depleted of red blood cells and CD31⁺ endothelial cells, and cancer cells were gated as α 6-integrin⁺CD45⁻EpCAM⁺ or EpCAM⁻ cells.

- The markers included in the flow cytometry results and in IF images are insufficient to determine the origins of the mesenchymal cells.

Regarding this point, we disagree with the reviewer as GFP expression in epithelial EpCAM⁺ cancer cells engrafted into mice can determine the epithelial origin of the mesenchymal cancer cells that emerge in the generated cSCCs (Fig. 1b,f). For example, as shown in Fig. 1f, epithelial GFP⁺EpCAM^{high} cancer cells (we ensured that these cancer cells expressed EpCAM prior to engraftment) were engrafted into immunocompetent mice that, in turn, generated mesenchymal GFP⁺EpCAM⁻ cancer cells. On the other hand, the origin of these mesenchymal cancer cells is not the main focus of this manuscript. Rather, our aim is to evaluate how the expression of IC ligands changes within epithelial and mesenchymal cancer cells and the impact of these changes on the response to ICI-based therapies.

- Likewise, in the human data (Fig. 1a) all of the cells here are being called tumor cells, but on what basis? I am unclear of the gating that they have already applied to exclude non-tumor stromal cells from their samples. The methods section says that they have been depleted of red blood cells and endothelial cells, but otherwise are stromal cells included? I assume they are gated out based on the fact that there appear to be fibroblasts in the WD-SCC IF photos and there are not any EpCAM⁻ cells in the flow results, but I think the authors should include the details of how they gated.

As suggested by the reviewer, we have included a more detailed explanation of how cSCCs have been generated in the methods section. Mouse cSCCs were depleted of red blood cells and CD31⁺ endothelial cells, and cancer cells were gated as α 6-integrin⁺CD45⁻EpCAM⁺ or EpCAM⁻ cells

(see new Supplementary Fig. 9). Isolated $\alpha 6$ -integrin⁺ cancer cells were then transduced with an MSCV-IRES-GFP lentivirus plasmid, thereby making it possible to identify them by the expression of GFP.

- The strength of the paper is in the data they present in figures 3-7 in which they convincingly show that PD-L1 is highly expressed among Epcam⁺ cells while CTLA-4 and TIGIT ligands are more highly expressed among Epcam⁻ cells. As a result, tumors with greater Epcam⁺ cell abundance are more sensitive to anti-PD-L1 treatment, while tumors with greater Epcam⁻ cell abundance are more sensitive to anti-CTLA-4/anti-TIGIT treatment (Figure 5). Predictably, the proportion of Epcam⁺ cells is altered in tumors in response to these different therapies (Figure 6). And among human patients those that are responsive to anti-PD-1 therapy have lower abundance of mesenchymal cells (Figure 7). While this is nicely shown, most of these conclusions have been shown previously in other systems.

We thank the positive comments made by the reviewer of our results included in the Figures 3-7. However, to the best of our knowledge, this manuscript is the first study reporting the changes in the expression of CD155 and CD80 ligands within cancer cells based on the epithelial and mesenchymal features of cSCCs, and their relationship with resistance to anti-PD-1/PD-L1 therapy. Otherwise, we would be very grateful if the reviewer could point us to a reference where same conclusions to our findings have been previously described.

A previous study demonstrated that TGF- β -responsive tumor initiating cells induce CD80 expression and become resistant to the antitumor response of adoptive cytotoxic T cell transfer (ACT)-based immunotherapy by inducing CD80 expression³⁶. Indeed, treatment with anti-CTLA-4, anti-TGF- β or abrogation of CD80 expression allowed tumor initiating cancer cells to recover their sensitivity to ACT therapy. In this regard, since TGF- β is a well-characterized EMT inducer in several cancer cell types³⁷⁻³⁹, these observations are consistent with the induced expression of CD80 and CD155 IC ligands on hybrid E/M and mesenchymal cSCC cells. A paragraph explaining this point has been included in the revised discussion section (lines 413-419).

Minor comments:

- I would like to see a more comprehensive panel of activation markers for the T cells in Figure 5 (like they showed in Figure 3f).

We have included a more comprehensive panel of activation/exhausted markers for T and NK cells in the revised Figs. 4, 5 and 6 (GzmB), and Supplementary Figs. 4, 5 and 6 (CD69, CD25, PD1, TIGIT and CTLA-4) to assess the recovery of CTL and NK cell activity after performing ICB therapies in epithelial, mixed and mesenchymal cSCCs.

- When the authors claim that “anti-PD-L1 treatment was not able to reinvigorate the exhausted state of CD8⁺ T cells in mesenchymal mouse cSCC,” I do not think they have sufficient data to claim reinvigoration especially in light of recent papers which demonstrate that much of the effect of PD(L)1 treatments is due to the emergence of novel T cell clones (Yost et al. 2019).

As we agree with the reviewer with respect to the original wording used, we have changed the previous statement “*The absence of changes in the percentage of CD8⁺GzmB⁺ T cells indicated that anti-PD-L1 treatment was not able to reinvigorate the exhausted state of CD8⁺ T cells in mesenchymal mouse cSCCs*” to the revised sentence “*The absence of changes in the percentage of GzmB⁺, CD69⁺, CD25⁺, PD-1⁺TIGIT⁺ and PD-1⁺CTLA-4⁺ CD8⁺ and NK cells indicated that anti-PD-L1 and anti-PD-1 treatments were not able to reverse the exhausted state of CTLs and NK cells in mesenchymal mouse cSCCs*”. Please see the revised text in lines 250-253.

- In Supplemental Figure 2f, I am not sure if the differences in the CD68⁺ signal are supported by the photos so I am curious how they selected areas for quantification. This is also a concern for panels g-j.

Immunofluorescence samples were scanned with a confocal microscope to visualize the distribution of the studied markers. Following pathology guidelines, a significant number of 40x magnified images (at least 10-20 images) were captured along the tumor section to accurately represent the intratumor heterogeneity of the sample. The expression of the analyzed markers within cancer cells was quantified from the magnified images. As suggested by the reviewer, an improved explanation of this strategy has been included in the “Histology, immunofluorescence and immunohistochemistry assays” section of the methods (lines 606-613). The final quantification results are the ones that recapitulate the heterogeneity of each sample. We agree that it is sometimes challenging to select a representative image that accurately reflects the quantitative results. Following the reviewer’s suggestions, we have changed the representative images for CD8, Gr1, CD68, CD163 and FoxP3 labeling in the revised Supplementary Fig. 2a-j.

- The authors claim that they have “show[n] elevated recruitment of immunosuppressive and exhausted immune cells,” is not fully supported- they cannot distinguish between recruitment and polarization of resident immune cells towards an immune-suppressive state.

As we agree with the reviewer that we cannot distinguish between recruitment or polarization of resident immune cells towards an immunosuppressive state during cSCC progression, we have changed the wording in this statement to “*cSCCs enriched in hybrid E/M and mesenchymal cancer cells show an increased frequency of immunosuppressive and exhausted immune cells.*” Please see the revised text in lines 183-185.

- The details of the mouse model are not laid out- how do they achieve WD-, MD-, PD- SCC tumors in mice?

As suggested by the reviewer, we have included a more detailed explanation of how cSCCs have been generated in the “Primary cSCC cancer cell cultures” and “Tumor-cell grafting and *in vivo* treatments” sections of the methods (lines 482-487 and 491-497), as well as the flow cytometry gating strategy to identify cancer cell populations in the new Supplementary Fig. 9. To characterize the changes of cancer cell features during cSCC progression, we isolated $\alpha 6$ -integrin⁺CD45⁻ cancer cells from mouse WD-SCCs (full epithelial cancer cells), MD/PD-SCCs (plastic epithelial EpCAM⁺ and mesenchymal EpCAM⁻ cancer cells) and PD/S-SCCs (full mesenchymal cancer cells), which were previously generated by orthotopic serial engraftments²² (see new Supplementary Fig. 9). Isolated cancer cells were then transduced with an MSCV-IRES-GFP lentivirus plasmid, thereby making it possible to identify them by the expression of GFP. Epithelial/WD (>70% EpCAM⁺ cancer cells), mixed/MD (10-70% EpCAM⁺ cancer cells) and mesenchymal/PD cSCCs (<10% EpCAM⁺ cancer cells) were generated after sorting and engrafting GFP⁺CD45⁻ full epithelial cancer cells, epithelial GFP⁺CD45⁻EpCAM⁺ and mesenchymal GFP⁺CD45⁻EpCAM⁻ cancer cells, respectively, into immunocompetent syngeneic mice (see new Supplementary Fig. 9).

- Indicating Marker 1 and Marker 2 double positive cells as Marker1/Marker2 is confusing because it reads like a ratio.

We have renamed the double positive cells as Marker1Marker2 to avoid any confusion in the revised manuscript text, figures and figure legends. Please see the updated nomenclature throughout the revised manuscript.

- In Figure 2 (G4 especially), the IF images have so few markers and no landmarks, making it challenging to assess. Additional reference markers are required.

For analyses on patient tumor sections, we first scanned hematoxylin and eosin (H/E) samples to obtain an overview of the tissue structure, and then performed training at the Pathology Service of the Bellvitge University Hospital to differentiate cancer cells from stromal cells by nuclear atypia (crowded, pleomorphic, and often large and hyperchromatic) and cell size. This training has allowed us to evaluate the expression of different markers in cancer cells and not in stromal cells. An improved explanation of this strategy has been included in the revised “Histology, immunofluorescence and immunohistochemistry assays” section of the methods (lines 606-613).

Reviewer #3 (Remarks to the Author): with expertise in skin cancer

The authors have performed a detailed immunophenotyping into epithelial, mesenchymal and hybrid cSCC subtypes. While the discovery of hybrid cSCC subtypes are not new, they are a relatively little studied cSCC subtype and the authors have brought new understanding into how these epithelial, mesenchymal and hybrid cSCC subtypes affect immune cell tumor microenvironment in both *in vivo* mouse models and in patient samples. Moreover, the authors explore IC ligand expression in the above subtypes and make the new finding that different epithelial/mesenchymal subtypes confer sensitivity/resistance to different Immune Checkpoint inhibitor (ICI) therapies. These results have translational relevance as they help us to understand how cSCC patients could be more effectively treated in the clinic. Moreover, the authors show that a mixed population of epithelial/mesenchymal subtypes respond to monotherapy of ICI through expansion of the more resistant subtypes (potentially suggesting a more plastic adaptive role in this subtype?), thereby supporting the notion that mixed subtypes should be treated with combination therapy.

We would like to thank the reviewer for the positive evaluation of our work. Specially, we very much appreciate the reviewer recognition of our study to “bring new understanding into how epithelial/mesenchymal/hybrid cSCC subtypes affect the TME in both *in vivo* mouse models and in patient samples” and how we are providing the “new finding that different E/M subtypes confer resistance to different ICI therapies.” Additionally, we would like to deeply acknowledge the reviewer for positively valuing the translational relevance of our findings, and how our study may help to provide more effective combination therapies for patients presenting mixed subtypes.

The epithelial and mesenchymal subtype categorizations and their link to cSCC progression is not new, and likewise the existence of hybrid E/M subtypes have been found, however fairly recently. Where the authors work adds novelty and value is in the deep immunophenotyping of the tumor microenvironment of these subtypes and the elegant cross-comparison between cSCC mouse models and using these to inform studies in patient samples. The implication for targeted therapy among these three phenotypes is also new. The work presented does support the overall claims, however, understandably sample size of patient samples is fairly limited. While the difference between patient samples is striking and in-line with the mouse models, to bolster the final claims about ICI treatment and cSCC subtypes, it would be strongly recommended if the authors could access some publicly available datasets (RNA-seq), if such a resource is available.

Following the reviewer’s suggestion, we launched a search to explore whether some publicly RNA-seq datasets from cSCC patients treated with anti-PD-1/PD-L1 antibodies were available to analyze the clinical value of our predictive biomarkers (CD80 and CD155 proteins). Unfortunately, we have not found any RNA-seq dataset available. To strengthen our conclusions,

we have evaluated the expression of these biomarkers in 2 independent cohorts of HNSCC and melanoma patient samples (new Fig. 7 and Supplementary Fig. 7 and 8). Our results show that, similar to what we have previously described in cSCCs, the enrichment of mesenchymal Vim⁺ cancer cells is associated with a change in the IC ligand repertoire towards CD80/CD155 expression and with anti-PD-1 resistance in HNSCC and melanoma patients. In accordance, it has been recently reported that melanoma cells harboring a mesenchymal-like state are enriched in on-treatment lesions from refractory ICB patients⁴⁰.

Overall, the writing is clear, the methodology is logical and sound and the interpretation reasonable. The sequential mouse model-to-patient data format is particularly compelling and pleasing. On the whole I recommend acceptance of the manuscript with some minor changes, which are required for quality control, or clarification of methods or statistics:

Data analysis/presentation:

- The authors have not presented any of their gating strategies for flow in the supplementary materials. This is an absolute must for publication in Nature Communications.

We have included complete flow cytometry gating strategies for cancer and immune cell populations in the Supplementary Fig. 9 of the revised manuscript.

- I am not an immunologist or personally have extensive expertise in immunophenotyping so from my limited expertise I can say that the markers used seem appropriate and comprehensive for the interpretation extracted from the study. I would urge editors to deviate to other reviewers who are more proficient in this area however.

We have used well-established and validated markers in the cancer immunology field to define and analyze the different immune cell populations included in our study (Supplementary Fig. 9).

- The authors have represented individual experiments/mice as individual dots on a graph, which is great. Please be clear if these are biological or technical replicates. For mouse I suspect technical replicates within 1 biological replicate, this is ok but must be clear for the reader.

For all *in vivo* mouse data, each dot represents one single tumor. It is important to highlight that in order to comply with the 3Rs principle and following our protocols approved by the IDIBELL ethics committee (18003, DMAH10402), each mouse bears 2 tumors into the back skin and these have been considered as biological replicates.

- Be more considered of statistical tests used. There are many examples where $n < 10$ and the authors have used parametric tests. Consider non-parametric tests where the sample size is not appropriate to reasonably detect if the distribution is normal.

Following the reviewer's advice, we have tested data normal distribution when $n < 10$ using the Shapiro-Wilk ($n < 5$) or Kolmogorov-Smirnov ($5 < n < 10$) normality tests. Unpaired two-tailed Student's *t*-test for comparisons of two groups and one-way ANOVA with Tukey's or Dunnett's test for comparisons of multiple groups have been applied to continuous normal data. Non-parametric Kruskal-Wallis test for comparisons of multiple groups has been used when data distribution failed normality tests.

- For tumor growth plots from mouse models, please state if all points represent cumulative growth from all tumors or if any were removed as outliers during the experiment.

We have not removed outliers during *in vivo* experiments. To make the data clearer, we have included the growth kinetics of all individual tumors in the new Supplementary Fig. 4a,h,i, 5a,h and 6a,b,k of the revised manuscript.

- For Figs 5 and 6, why are there different n-numbers for the %CD8+/CD4- within CD3+ and %GzmB+ within CD8+ versus total numbers of mouse tumors followed for tumor growth.

We understand the reviewer's concern about the different n-numbers used for different analyses within the same experiment. These differences are because, in some cases, there was insufficient material to analyze all flow cytometry combinations. However, in all new *in vivo* experiments, we have ensured an n of 10 for all populations analyzed.

- For the Immune Checkpoint inhibitor tumor studies, I have never seen dosing begin at a palpable tumor size of 15mm^3 before. From all experience it is not possible to detect a tumor of this size accurately. Plus, in the methods the authors state the measurement is $3\text{mm} \times 3\text{mm}$ in size. This is not a 15mm^3 tumor. 3×3 would give you a 9mm^2 surface area. The authors could choose to report 3 measurements (height x width x length) and essentially calculate the volume of a cuboid, where $3 \times 3 \times 3$ would equal a volume of 27mm^3 . Or the authors could choose to input their measurements to calculate the volume of a sphere or hemisphere (depending on how they believe their tumors grow). Most usually $H \times W \times L$ is taken. Either way, such a tiny initial size is unusual and I have concerns for the accuracy of the onset of treatment. Especially if one subtype was inherently stiffer than another. Usually, people start dosing at around 75mm^3 . I would not advocate re-doing the mouse experiments on this basis, but rather being clearer about the size measurements and ultimately volume. Also presenting in the supplementary the time from cancer cell engraftment to dosing for each experiment.

We are grateful to the reviewer for raising this point and we apologize for the confusion generated with the tumor measurements. First, we would like to clarify that in the previous version of the manuscript there was an error as we initiate treatments when tumors reach a size of $5\text{mm} \times 5\text{mm}$ instead of $3\text{mm} \times 3\text{mm}$. We use the following formula to calculate the tumor volume: $V (\text{mm}^3) =$

$\pi/6 \times L \times W^2$ (L: largest tumor diameter, W: perpendicular measurement). Therefore, with a tumor size of 5mm x 5mm and, after applying our formula, the resulting volume is of about 65 mm³. We usually reached this 65 mm³ volume within one week of epithelial and mesenchymal cancer cell engraftment (see the scheme in Fig. 4a). We have included a more extended explanation of the calculation used to assess tumor volume in the revised methods section. Please see revised text in lines 497-500.

- Why is the Cox Regression analysis not performed on the Epcam-/Vim+/CD80+/CD155+ signature as a whole in addition to individual genes?

Although we consider that it would be very interesting to have these data, the Epcam⁻Vim⁺CD80⁺CD155⁺ signature as a whole in addition to individual genes is not possible because we did not analyze these markers within the same immunofluorescence experiment. Indeed, Cox analyses were performed on markers evaluated in the same assay.

- Figure 7 would massively benefit from a Kaplan Meier plot. I realize this is not feasible with the limited patient samples available (7 per group), is there a possibility of using publicly accessible patient datasets to perform this analysis? It would be hugely compelling for the paper.

We agree with the reviewer that our results would massively benefit from Kaplan-Meier curves indicating the survival probability of cSCC patients categorized as having high or low levels of expression of our biomarkers. However, even it has not been feasible to perform due to the limited patient samples analyzed in our study (7 per group), HR models showed that the higher the percentage of Vim⁺, CD80⁺ and CD155⁺ cancer cells, the higher the risk of progression/relapse (Fig. 7o). For this reason, we are designing a prospective clinical trial to demonstrate the clinical value of these biomarkers. Finally, as discussed above, we have not found any RNA-seq dataset from cSCC patients treated with anti-PD-1/PD-L1 antibodies to analyze the clinical value of our predictive biomarkers (CD80 and CD155 proteins).

Text/context/interpretation:

- I find it very intriguing in Fig. 6 c, f, i, that there is a change in the proportion of the resistant/sensitive subtype depending on the treatment given, yet in Fig. 5 there is no change. My first-pass interpretation, perhaps overlooked by the authors, is that the mixed population is more plastic (ergo the title of the manuscript) and only the mixed population has the capacity for this plastic adaptation. Perhaps the authors could expand on this with a sentence or two in the discussion (if they likewise agree).

To avoid misinterpretation of these results, we have expanded the explanation. In fact, in the case of tumors composed only of epithelial (WD-SCCs, Fig. 4) or mesenchymal (PD-SCCs; Fig. 5)

cancer cells, the treatment resulted in a reduction in tumor size without changes in the percentage of epithelial or mesenchymal cancer cells, as all cancer cells were either sensitive (died in response to therapy) or resistant (survive to therapy) (see Fig. 4q,r,s and 5n,o). In contrast, when tumors are composed of epithelial and mesenchymal cancer cells (mixed cSCCs, Fig. 6), anti-PD-L1 therapy allowed the elimination of epithelial cancer cells, leading in consequence to an enrichment of the mesenchymal cancer cell component (Fig. 6p,r). On the other hand, anti-TIGIT and CTLA-4 therapies led to opposite results, as mesenchymal cancer cells are sensitive to this therapy and died, resulting in an enrichment of TIGIT/CTLA-4-resistant epithelial cancer cells (see Fig. 6q,r).

In addition, we have changed the previous statement *“It is of note that no changes were observed in the total content of epithelial EpCAM⁺ and mesenchymal EpCAM⁻ cancer cells after performing the ICB treatments on epithelial and mesenchymal mouse cSCCs”* to the revised sentence *“As expected from a homogeneous cancer cell composition, no changes in the relative content of epithelial EpCAM⁺ and mesenchymal EpCAM⁻ cancer cells vs. total cancer cells were observed after performing the ICB therapies on epithelial and mesenchymal mouse cSCCs (Fig. 4q-s and 5n,o)”*. Please see the revised text in lines 262-265.

- Lines 96-97 in regards to figure 1d-e: *“Conversely, EpCAM^{low} cancer cells from MD/PD-SCCs retained the expression of Krt14, Grhl1, Grhl2 and dNp63 epithelial markers,”* I find the above confusing as in the figures the bars go down in EpCAM^{low} cancer cells. Find a way to marry the text and the visual representation better, perhaps a better description is a gradient?

As suggested by the reviewer, we have modified the text to better describe the changes in the expression of epithelial and mesenchymal markers genes within the different cancer cell populations. Please see the revised text in lines 96-104: *“Their molecular characterization revealed that EpCAM^{high} cancer cells from MD/PD-SCCs expressed epithelial differentiation genes in a similar fashion to EpCAM^{high} cancer cells from WD-SCCs (henceforth named full epithelial cancer cells) (Fig. 1d), although with induced expression of Vim and Zeb1 compared with full epithelial cancer cells (Fig. 1e). Conversely, EpCAM^{low} cancer cells from MD/PD-SCCs did not show significant changes in the expression of Krt14, Grhl1, Grhl2 and dNp63 epithelial genes compared with full epithelial cancer cells, but exhibited diminished expression of Cdh1, Epcam, Ovol1 and Ovol2, and upregulated the expression of Vim and several EMT transcription factors (TFs) to similar levels to those in EpCAM⁻ cancer cells (Fig. 1d,e)”*.

- Lines 169-171: *“We corroborated in cSCC patient samples that the frequency of GzmB⁺ cells decreased and the frequencies of CD8⁺/PD-1⁺, CD8⁺/LAG-3⁺, CD8⁺/TIM-3⁺ and CD8⁺/TIGIT⁺ cells increased in mixed and mesenchymal patient cSCCs (Supplementary Fig. 4a-*

j)” Agreed regarding the proportions of markers per CD8⁺ cells. But also, from the pictures shown, the total number of CD8⁺ cells seem to peak in mixed samples. Is this a real thing or just the images shown, if it is real, it deserves a mention.

The real changes are those presented in the quantification plots, as they recapitulate all the tumor cell heterogeneity more accurately. Following the reviewer’s suggestion, we have selected other representative images that better represents the quantitative results, and they have been included in the Supplementary Fig. 3c,e,g,i of the revised manuscript.

- Lines 193-194: “The frequency of CD80⁺/Ecad⁺ and CD155⁺/Ecad⁺ cancer cells specifically increased in mixed patient cSCCs (Fig. 4e,f), suggesting that the expression of these IC ligands might be associated with the appearance of hybrid E/M cancer cells” Careful with wording here, yes for CD155, there is absolutely a trend for expression only occurring upon appearance of hybrid cells. However, for CD80, there are still 5-10% of epithelial cells that express CD80. Therefore perhaps...”suggesting that the upregulation of these IC ligands...” would be more appropriate.

As suggested by the reviewer, we have changed the sentence to “*The frequency of CD80⁺Ecad⁺ and CD155⁺Ecad⁺ cancer cells specifically increased in mixed patient cSCCs (Fig. 2j,k), suggesting that the upregulation of these IC ligands is associated with the enrichment of hybrid E/M cancer cells at intermediate cSCC stages*”. Please see the updated text in lines 200-205 of the revised manuscript.

- Regarding Figure 4: Is there more heterogeneity in CD80 and CD155 expression in mixed samples? With the caveat that IF staining is a semi-quantitative method.

We believe that the variability of CD80/CD155 expression could be explained by the great heterogeneity of cancer cell features found in mixed cSCCs. In this regard, mixed cSCCs are composed of full epithelial cancer cells that express very little CD80/CD155, plastic/hybrid cancer cells that can express different levels of these IC ligands, and mesenchymal cancer cells with high CD80/CD155 expression. Therefore, depending on the cancer cell composition of mixed cSCCs, we will have a lower or a higher total expression of CD80 and CD155 ligands.

Following minor revisions as stated I would strongly recommend publication of this manuscript in Nature Communications.

We would like to thank again the reviewer for the very positive overall evaluation of our study. We believe that we have now been able to address the entirety of the comments in the revised version of the manuscript. We are confident that the current revised version of the manuscript has been improved and will be greatly welcomed by the broad readership of *Nature Communications*.

Reviewer #4 (Remarks to the Author): with expertise in cancer immunology

While immune checkpoint inhibitors (ICI) are used in a wide range of cancer types, not all patients can receive clinical benefits. Tumor cells evade antitumor immunity in a variety of ways, and there are needs for biomarkers to predict efficacies and therapies to overcome resistance. In cutaneous cell carcinoma (cSCC), anti-PD-1 inhibitor is used for advanced cases, but any combination therapies with ICIs have not been approved. The authors showed that the cSCC progression with mesenchymal differentiation features changed IC ligand profile of tumor cells in both a mouse model and clinical samples, and this change was also seen in anti-PD-1 non-responder cSCC as well. These results suggest that treatment with anti-PD-1 inhibitor in combination with anti-CTLA-4 inhibitor or anti-TIGIT inhibitor may give better outcomes in cSCC patients of any grade. Also, the degree of expression of these ligands can predict tumor progression and response to ICI, and this information is quite useful for cSCC patients. Although this manuscript is interesting with valuable data, there are several serious concerns.

We thank the reviewer for the positive evaluation of our study that in her/his view is “interesting with valuable data”, and for raising her/his concerns about several points that we have addressed below.

Major concerns:

- There are no mechanistic data on higher infiltration of lymphocytes in mixed and mesenchymal cSCCs and IC ligand profile changes during cSCC progression. I wonder why PD-L1 expression decreased in mesenchymal cSCCs even with higher infiltration of lymphocytes because PD-L1 is usually induced by IFN- γ from lymphocytes.

We thank the reviewer for making this observation. However, at this stage, we are not in a position to provide a comprehensive mechanistic data explaining why mixed and mesenchymal cSCCs exhibit a higher frequency of CD8⁺ T cells. Therefore, considering the interesting research question and the demanding experimental workload needed to tackle it, future experiments outside the scope of this manuscript will be performed focusing on the identification of mesenchymal cancer cell-derived cytokines that promote the recruitment of CD8⁺ T cells, as well as on the regulatory mechanisms controlling the expression of CD80 and CD155 IC ligands.

Regarding the comment on PD-L1 expression in mesenchymal cSCCs, although we observed a greater infiltration of CD8⁺ T cells in these tumors, we also showed that these cells were mostly exhausted and expressed less IFN- γ in mixed and mesenchymal cSCCs (Supplementary Fig. 11), which may preclude the stimulation of PD-L1 expression by this pathway. Given that the reviewer raised an interesting point that we did not discuss in the previous version of the manuscript, we have now added a paragraph in the revised discussion section to highlight that despite the higher

infiltration of CD8⁺ T cells, they were in an inactive state, and therefore, it might preclude the IFN- γ -mediated stimulation of PD-L1 expression in mesenchymal cSCCs. Please see revised text in the lines 420-423.

- I have a question about CD80. CD80 is originally expressed in hematopoietic cells. I previously analyzed CD80/CD86 expression using CCLE datasets, showing that CD80 is seldom expressed in epithelial cancer cells.

We agree with your observations, as the expression of CD80 is reduced in epithelial cancer cells and it is strongly induced in hybrid E/M and mesenchymal cancer cells. Please see the Fig. 2d where we specifically monitored the percentage of CD80⁺ cells within cancer cells ranging from full epithelial (with low levels of CD80) to full mesenchymal cancer cells (with high levels of CD80).

- Related to CD80, the authors should analyze CD28 expression during progression.

As suggested by the reviewer, we have analyzed CD28 expression in epithelial, mixed and mesenchymal cSCCs. These data have been included in the Supplementary Fig. 1i of the revised manuscript. Our results show that the expression of the co-stimulatory receptor CD28 significantly decreases in CD8⁺ T cells during mouse cSCC progression (see new Supplementary Fig. 1i). Given that CD80 is the ligand for both CD28 co-stimulatory and CTLA-4 inhibitory receptors, and that the percentage of CD28⁺ CD8⁺ T cells decreased and PD-1⁺CTLA-4⁺ CD8⁺ T cells increased during cSCC progression (Supplementary Fig. 1g), our results suggest that higher CD80 levels in mesenchymal cSCCs might be related to the inhibition of the antitumor response of CD8⁺ T cells through the CTLA-4/CD80 pathway, as also suggested by the higher affinity of CD80 for CTLA-4 than CD28⁴¹.

- In Figure 3b-g, the authors used anti-CD3/CD28 mAbs for the stimulation. However, as this stimulation is non-specific, T cell activation can be non-specific for cancer cells. The authors should conduct experiments *in vivo* instead of such questionable *in vitro* experiments.

We agree with the reviewer that *in vitro* CD8⁺ T cell activation may not be specific for cancer cells, but even under these conditions, we observed a cancer cell-mediated inhibition of CD8⁺ T cells and their activity was recovered upon treatment with ICIs.

As suggested by the reviewer and to complement the *in vitro* co-culture experiments, we have performed new *in vivo* experiments to test the effect of CD8 or NK cell depletion on the response to ICI therapy. These new experiments have been included in the revised version of the manuscript (new Fig. 4b-f, 5f-o and Supplementary Fig. 4a-g, 5h-n). Notably, we observed that anti-PD-L1 response is CD8⁺ T cell-dependent rather than relying on NK cells in epithelial cSCCs

(new Fig. 4b-f and Supplementary Fig. 4a-g). Furthermore, the delayed tumor growth conferred by anti-CTLA-4 and anti-TIGIT therapies in mesenchymal cSCCs is partially lost after CD8 and NK cell depletion, indicating that these responses are mediated by both CTLs and NK cells (new Fig. 5f-m and Supplementary Fig. 5h-n). Given that anti-PD-L1, anti-CTLA-4 and anti-TIGIT therapies also reverse the exhausted state of both CD8⁺ T cells and NK cells in mixed cSCCs (new Fig. 6 and Supplementary Fig. 6), we suggest that both populations could play a key role in the response of cSCCs enriched in epithelial, hybrid E/M and mesenchymal cancer cells (mixed cSCCs), even though we have not performed depletion experiments in this context.

- CD155-TIGIT axis is also deeply involved with CD122/CD226. How were these molecules expressed in this mouse model?

We appreciate the suggestion made by the reviewer. We have analyzed by flow cytometry the expression of CD112 in epithelial, hybrid E/M and mesenchymal cancer cells (see new Fig. 2b), as well as CD226 in CD8⁺ T cells (see new Supplementary Fig. 1j) and NK cells (see new Supplementary Fig. 1p). Interestingly, we have observed that the expression of CD112 follows an opposite profile to that of CD155. Indeed, CD122 expression is downregulated in hybrid E/M and mesenchymal cancer cells, which in turn strongly induced the CD155 expression.

Minor concerns:

- Because Gal9 expression changed, the authors should investigate TIM-3 blockade, too.

We agree with the reviewer that investigating the relevance of TIM-3 blockade is an interesting research line to further explore, given the expression of Gal9. However, for the revision process, we have concentrated our efforts in advancing our understanding of the anti-PD-1/PD-L1, anti-TIGIT and anti-CTLA-4 therapies. We have also performed additional multi-arm *in vivo* experiments such as the CD8 or NK cell depletions to assess their contribution to the response to ICB therapies. Weighting the interesting research question but also the demanding *in vivo* experimental workload required, future experiments will be performed outside the scope of this manuscript with the goal of dissecting the Gal9/TIM-3 axis.

- In Figure 5, I want to know the treatment of mice bearing epithelial cSCCs with anti-CTLA-4 antibody. Although combination therapy with anti-PD-1 antibody and anti-CTLA-4 antibody has been used in other skin cancers, please indicate any reasons for choosing anti-TIGIT antibody, not anti-CTLA-4 antibody, for use with anti-PD-L1(PD-1) antibody.

As requested by the reviewer, we have included an experiment to test the response of epithelial cSCCs to anti-CTLA-4 therapy. We have observed that epithelial cSCCs are resistant to anti-CTLA-4 therapy. New data have been included in the Fig. 4g-k and Supplementary Fig. 4h,j-o.

Regarding the point raised by the reviewer about the rationale for the combined therapy, we have chosen the combination of anti-PD-L1 + anti-TIGIT due to the favorable tolerability of these therapeutic antibodies. Blockade of CTLA-4, either as monotherapy or as combined therapy, is associated with higher toxicity and adverse events as compared with anti-TIGIT therapy. It should be noted that none of these combined therapies are currently approved for cSCC treatments.

- Please provide more detailed information about antibodies including Fc region. Particularly, ADCC activities should be considered in CTLA-4 antibodies.

In this study, we have used commercially available antibodies purchased from BioXCell, a trusted supplier of antibodies for *in vivo* ICB mouse treatments. The complete list of antibodies used for *in vivo* experiments are included in the methods section, please see lines 501-507. We have not obtained more information (specifically about Fc region) from these antibodies from BioXCell, but we would expect a similar ADCC activity from all of them, including the IgG isotype control. As a specific response was observed with anti-PD-1/PD-L1 antibodies or anti-CTLA-4 antibodies compared to the control IgG antibody in epithelial or in mesenchymal cSCCs, we may conclude that the observed effects are not mediated by ADCC activities.

- TIGIT blockade is expected to be effective in lung cancer with high PD-L1 expression. However, the present data are somewhat conflicting. Please discuss.

Lung squamous cell carcinoma (SCC), which is the second most frequent non-small cell lung cancer (NSCLC), remains difficult to treat and have limited effective treatments beyond chemoimmunotherapy. The results of CheckMate 227 and CITYSCAPE studies indicated that combined immunotherapy (anti-CTLA-4 + anti-PD-1 or anti-TIGIT + anti-PD-L1) show durable benefits in OS and PFS across all levels of PD-L1 expression⁴². The CITYSCAPE clinical trial shows very good clinical efficacy in patients with PD-L1 \geq 50% who were treated with anti-PD-L1 plus anti-TIGIT antibodies⁴³. In this regard, as EMT has been also described in lung SCCs⁴⁴, we hypothesize that a similar cancer cell heterogeneity may appear in some lung SCCs. In this scenario, a good response to anti-TIGIT and anti-PD-L1 combined therapy may be expected in those cases having an enrichment of hybrid E/M or mesenchymal cancer cells, as here described for cSCCs. The expression of PD-L1 may be associated with the more epithelial cancer cell component of lung tumors, whereas the response may be enhanced in the presence of hybrid E/M or mesenchymal cancer cells by the concomitant treatment with anti-TIGIT. For these cases, the expression of CD155 may be informative about the response to TIGIT therapy. However, experimental validation is needed to test this hypothesis.

- EMT is known to occur in other skin cancers such as melanoma. It is desirable to verify whether the changes in IC ligand profile are specific to SCC or general.

As we hypothesize that the described changes in IC ligand expression can be extended to other tumor types that present cancer cell heterogeneity and mesenchymal cancer cell states, we have analyzed the expression of Ecad, Vim, CD80 and CD155 in HNSCC and melanoma patient samples (see new Fig. 7 and Supplementary Fig. 7 and 8). We have observed that similarly to that here reported for cSCCs, melanoma and HNSCC cancer cells with mesenchymal features (loss of E-cadherin expression and increased expression of the mesenchymal Vimentin marker) were significantly enriched in pre-treatment samples from anti-PD-1/PD-L1 non-responder patients. Similarly, a significantly higher percentage of CD80⁺ and CD155⁺ cancer cells was detected in non-responder patients, indicating that the changes in the expression of these IC ligands may be link to the acquisition of EMT features in a general way.

1. Johnston, R. J. *et al.* The Immunoreceptor TIGIT Regulates Antitumor and Antiviral CD8+ T Cell Effector Function. *Cancer Cell* **26**, 923–937 (2014).
2. Chauvin, J.-M. *et al.* TIGIT and PD-1 impair tumor antigen-specific CD8⁺ T cells in melanoma patients. *J. Clin. Invest.* **125**, 2046–2058 (2015).
3. He, W. *et al.* CD155/TIGIT Signaling Regulates CD8+ T-cell Metabolism and Promotes Tumor Progression in Human Gastric Cancer. *Cancer Res.* **77**, 6375–6388 (2017).
4. Hung, A. L. *et al.* TIGIT and PD-1 dual checkpoint blockade enhances antitumor immunity and survival in GBM. *Oncot Immunology* **7**, e1466769 (2018).
5. Freed-Pastor, W. A. *et al.* The CD155/TIGIT axis promotes and maintains immune evasion in neoantigen-expressing pancreatic cancer. *Cancer Cell* **39**, 1342-1360.e14 (2021).
6. Banta, K. L. *et al.* Mechanistic convergence of the TIGIT and PD-1 inhibitory pathways necessitates co-blockade to optimize anti-tumor CD8+ T cell responses. *Immunity* **55**, 512-526.e9 (2022).
7. Chu, X., Tian, W., Wang, Z., Zhang, J. & Zhou, R. Co-inhibition of TIGIT and PD-1/PD-L1 in Cancer Immunotherapy: Mechanisms and Clinical Trials. *Mol. Cancer* **22**, 93 (2023).
8. Topalian, S. L., Taube, J. M., Anders, R. A. & Pardoll, D. M. Mechanism-driven biomarkers to guide immune checkpoint blockade in cancer therapy. *Nat. Rev. Cancer* **16**, 275–287 (2016).
9. Wolchok, J. D. *et al.* Overall Survival with Combined Nivolumab and Ipilimumab in Advanced Melanoma. *N. Engl. J. Med.* **377**, 1345–1356 (2017).
10. Cho, B. C. *et al.* Tiragolumab plus atezolizumab versus placebo plus atezolizumab as a first-line treatment for PD-L1-selected non-small-cell lung cancer (CITYSCAPE): primary and follow-up analyses of a randomised, double-blind, phase 2 study. *Lancet Oncol.* **23**, 781–792 (2022).
11. Niu, J. *et al.* First-in-human phase 1 study of the anti-TIGIT antibody vibostolimab as monotherapy or with pembrolizumab for advanced solid tumors, including non-small-cell lung cancer☆. *Ann. Oncol.* **33**, 169–180 (2022).
12. Rudin, C. M. *et al.* SKYSCRAPER-02: Tiragolumab in Combination With Atezolizumab Plus Chemotherapy in Untreated Extensive-Stage Small-Cell Lung Cancer. *J. Clin. Oncol.* **42**, 324–335 (2024).
13. Schnell, A., Bod, L., Madi, A. & Kuchroo, V. K. The yin and yang of co-inhibitory receptors: toward anti-tumor immunity without autoimmunity. *Cell Res.* **30**, 285–299 (2020).
14. He, X. & Xu, C. Immune checkpoint signaling and cancer immunotherapy. *Cell Res.* **30**, 660–669 (2020).
15. Migden, M. R. *et al.* Cemiplimab in locally advanced cutaneous squamous cell carcinoma: results from an open-label, phase 2, single-arm trial. *Lancet Oncol.* **21**, 294–305 (2020).
16. Gross, N. D. *et al.* Neoadjuvant Cemiplimab for Stage II to IV Cutaneous Squamous-Cell Carcinoma. *N. Engl. J. Med.* **387**, 1557–1568 (2022).
17. Toll, A. *et al.* Epithelial to mesenchymal transition markers are associated with an increased metastatic risk in primary cutaneous squamous cell carcinomas but are attenuated in lymph node metastases. *J. Dermatol. Sci.* **72**, 93–102 (2013).
18. Barrette, K. *et al.* Epithelial-mesenchymal transition during invasion of cutaneous squamous cell carcinoma is paralleled by AKT activation. *Br. J. Dermatol.* **171**, 1014–1021 (2014).
19. Pastushenko, I. *et al.* Identification of the tumour transition states occurring during EMT. *Nature* **556**, 463–468 (2018).

20. Navas, T. *et al.* Clinical Evolution of Epithelial–Mesenchymal Transition in Human Carcinomas. *Cancer Res.* **80**, 304–318 (2020).
21. Ji, A. L. *et al.* Multimodal Analysis of Composition and Spatial Architecture in Human Squamous Cell Carcinoma. *Cell* **182**, 497–514.e22 (2020).
22. Silva-Diz, V. da *et al.* Cancer Stem-like Cells Act via Distinct Signaling Pathways in Promoting Late Stages of Malignant Progression. *Cancer Res.* **76**, 1245–1259 (2016).
23. Bernat-Peguera, A. *et al.* FGFR Inhibition Overcomes Resistance to EGFR-targeted Therapy in Epithelial-like Cutaneous Carcinoma. *Clin. Cancer Res.* **27**, 1491–1504 (2021).
24. Taube, J. M. *et al.* Association of PD-1, PD-1 Ligands, and Other Features of the Tumor Immune Microenvironment with Response to Anti-PD-1 Therapy. *Clin. Cancer Res.* **20**, 5064–5074 (2014).
25. Mak, M. P. *et al.* A Patient-Derived, Pan-Cancer EMT Signature Identifies Global Molecular Alterations and Immune Target Enrichment Following Epithelial-to-Mesenchymal Transition. *Clin. Cancer Res.* **22**, 609–620 (2016).
26. Dongre, A. *et al.* Epithelial-to-Mesenchymal Transition Contributes to Immunosuppression in Breast Carcinomas. *Cancer Res.* **77**, 3982–3989 (2017).
27. Noman, M. Z. *et al.* The immune checkpoint ligand PD-L1 is upregulated in EMT-activated human breast cancer cells by a mechanism involving ZEB-1 and miR-200. *Oncolimmunology* **6**, e1263412 (2017).
28. Kim, S. T. *et al.* Comprehensive molecular characterization of clinical responses to PD-1 inhibition in metastatic gastric cancer. *Nat. Med.* **24**, 1449–1458 (2018).
29. Keenan, T. E., Burke, K. P. & Van Allen, E. M. Genomic correlates of response to immune checkpoint blockade. *Nat. Med.* **25**, 389–402 (2019).
30. Shrestha, R., Prithviraj, P., Bridle, K. R., Crawford, D. H. G. & Jayachandran, A. Combined Inhibition of TGF- β 1-Induced EMT and PD-L1 Silencing Re-Sensitizes Hepatocellular Carcinoma to Sorafenib Treatment. *J. Clin. Med.* **10**, 1889 (2021).
31. Hugo, W. *et al.* Genomic and Transcriptomic Features of Response to Anti-PD-1 Therapy in Metastatic Melanoma. *Cell* **165**, 35–44 (2016).
32. Cao, L. *et al.* Prognostic Role of Immune Checkpoint Regulators in Cholangiocarcinoma: A Pilot Study. *J. Clin. Med.* **10**, 2191 (2021).
33. Brahmer, J. *et al.* Nivolumab versus Docetaxel in Advanced Squamous-Cell Non–Small-Cell Lung Cancer. *N. Engl. J. Med.* **373**, 123–135 (2015).
34. El-Khoueiry, A. B. *et al.* Nivolumab in patients with advanced hepatocellular carcinoma (CheckMate 040): an open-label, non-comparative, phase 1/2 dose escalation and expansion trial. *The Lancet* **389**, 2492–2502 (2017).
35. Socinski, M. A. *et al.* Atezolizumab for First-Line Treatment of Metastatic Nonsquamous NSCLC. *N. Engl. J. Med.* **378**, 2288–2301 (2018).
36. Miao, Y. *et al.* Adaptive Immune Resistance Emerges from Tumor-Initiating Stem Cells. *Cell* **177**, 1172–1186.e14 (2019).
37. Massagué, J. TGF β in Cancer. *Cell* **134**, 215–230 (2008).
38. Scheel, C. *et al.* Paracrine and Autocrine Signals Induce and Maintain Mesenchymal and Stem Cell States in the Breast. *Cell* **145**, 926–940 (2011).
39. Moustakas, A. & Heldin, C.-H. Induction of epithelial–mesenchymal transition by transforming growth factor β . *Semin. Cancer Biol.* **22**, 446–454 (2012).

40. Pozniak, J. *et al.* A TCF4-dependent gene regulatory network confers resistance to immunotherapy in melanoma. *Cell* **187**, 166-183.e25 (2024).
41. van der Merwe, P. A., Bodian, D. L., Daenke, S., Linsley, P. & Davis, S. J. CD80 (B7-1) Binds Both CD28 and CTLA-4 with a Low Affinity and Very Fast Kinetics. *J. Exp. Med.* **185**, 393–404 (1997).
42. Alifu, M. *et al.* Checkpoint inhibitors as dual immunotherapy in advanced non-small cell lung cancer: a meta-analysis. *Front. Oncol.* **13**, (2023).
43. Rodriguez-Abreu, D. *et al.* Primary analysis of a randomized, double-blind, phase II study of the anti-TIGIT antibody tiragolumab (tira) plus atezolizumab (atezo) versus placebo plus atezo as first-line (1L) treatment in patients with PD-L1-selected NSCLC (CITYSCAPE). *J. Clin. Oncol.* **38**, 9503–9503 (2020).
44. Ancel, J. *et al.* Clinical Impact of the Epithelial-Mesenchymal Transition in Lung Cancer as a Biomarker Assisting in Therapeutic Decisions. *Cells Tissues Organs* **211**, 91–109 (2020).

REVIEWERS' COMMENTS

Reviewer #1 (Remarks to the Author):

The revisions made by the authors have considerably improved the soundness of the data presented.

minor and editing points:

section discussion (L420): showed has to be replaced by shown

A supplementary table with all the antibodies used for flow cytometry and immunohistochemistry/immunofluorescence analyses with their concentration use and the targeted species (mouse and/or human) would be appreciated

Reviewer #2 (Remarks to the Author):

In the revised manuscript Lorenzo-Sanz et al. have extensively addressed reviewer critiques through an edited manuscript (in yellow) and additional primary and supplementary figures. The study focuses on cutaneous murine K14-HPV16 mouse line surface phenotypes and their sensitivities to IC antibody therapies. The authors use extensively-passaged murine tumors with relatively stable surface phenotypes to show that those with high epithelial markers tend to respond to anti-PD1 / PDL1 blockade, and those with higher vimentin, CD155 or CD80 respond to anti-TIGIT antibodies. The strengths of the work in the initial review were the clear correlation between tumor surface phenotype and response to blockade in mice. This was further enhanced by additional human cSCC data (Rev Fig 1, which should be included in the published version), additional flow cytometry controls and analyses (new Figs 3-4, new Supp Fig 1, 4, 9). In addition, a strength is the enhanced correlation with tumor surface phenotype with IC response rather than using PD1/L1 immunoreactivity. I think the revised manuscript has addressed many of the technical issues.

Despite this work and the addition data in the revision, the overall conceptual novelty still

detracts from the impact of the work. As mentioned by other reviewers well differentiated SCC with a prominent epithelial component is expected to respond differently from poorly differentiated pre-clinical and clinical studies (i.e. Chu Mol Cancer 2023, Ji et al Cell 2020). The present work uses a viral SCC model and the isolation of tumor cell lines with stable surface phenotypes to convincingly confirm the previous work.

Detracting from the novelty and pointed out by reviewers is the lack of consideration that tumors are not ONLY epithelial or mesenchymal but are heterogeneous and evolving. Additional methodology not included in the original manuscript reveals the extensively passaged lines and It remains curious why at least a portion of the tumor did not express heterogeneity. The original and updated human data in Fig 6 supports the authors finding from their mouse data to the extent it was analyzed.

Reviewer #3 (Remarks to the Author):

I am satisfied that all my comments have been addressed.

I wholeheartedly recommend this study for publication and congratulate the authors on such a well-revised and informative study.

As a small note, in my copy the alpha symbols in supplementary figures 4 and 5 appear distorted.

Reviewer #4 (Remarks to the Author):

The authors responded to my queries insufficiently. Particularly, there are no mechanistic data. Figure S1c, cytotoxic CD8 T cells increased, which could induce PD-L1 expression in cancer cells. But the data are conflicting. In addition, very high CD80 expression data could be questionable (Fig. 2d). The authors should analyze gene expression after sorting cancer cells. CD80 DOES NOT induce CTLA-4-mediated immune suppression. CTLA-4 reportedly disturb the CD28 signaling pathways by strongly binding to CD80 or CD86, leading to immunosuppression. Thus, CD28 expression could be very important in this model. If CD80

or CD86 is expressed in cancer cells (i.e., hematological malignancies), these molecules induce CD28 expression in T cells in addition to CTLA-4. However, the authors' data are conflicting.

NCOMMS-23-19093A-Z by Lorenzo-Sanz et al., “Cancer Cell Plasticity Defines Response to Immunotherapy in Cutaneous Squamous Cell Carcinoma”

Reviewer #1 (Remarks to the Author)

The revisions made by the authors have considerably improved the soundness of the data presented.

We would like to thank the reviewer for their positive and encouraging comment regarding our revised manuscript, which she/he considers substantially improved. It is rewarding receiving this feedback. Your comments and constructive feedback have been invaluable in shaping the final version of our study.

minor and editing points:

section discussion (L420): showed has to be replaced by shown.

Thank you for spotting this minor typo. The sentence has been corrected to “*Previous reports have shown that PD-L1 expression is induced in response to IFN- γ signaling*” in the revised version of the manuscript (see revised text in line 405).

A supplementary table with all the antibodies used for flow cytometry and immunohistochemistry/immunofluorescence analyses with the concentration used and the targeted species (mouse and/or human) would be appreciated.

As suggested by the reviewer, we have included a Supplementary Table listing all the antibodies used for flow cytometry and immunohistochemistry/immunofluorescence analyses with the required information (see Supplementary Table 4). Additionally, in the “*cSCC cancer cell sorting and flow cytometry assays*” and “*Histology, immunofluorescence and immunohistochemistry assays*” sections of the methods, relevant antibody information including concentrations, target species and commercial reference codes is already indicated.

Reviewer #2 (Remarks to the Author)

- In the revised manuscript, Lorenzo-Sanz et al. have extensively addressed reviewer critiques through an edited manuscript (in yellow) and additional primary and supplementary figures. The study focuses on cutaneous murine K14-HPV16 mouse line surface phenotypes and their sensitivities to IC antibody therapies. The authors use extensively-passaged murine tumors with relatively stable surface phenotypes to show that those that have high epithelial markers tend to respond to anti-PD-1/PD-L1 blockade, and those with higher vimentin, CD155 or CD80 respond to anti-TIGIT antibodies. The strengths of the work in the initial review were the clear correlation between tumor surface phenotype and response to blockade in mice. This was further enhanced

by additional human cSCC data (Rev Fig 1, which should be included in the published version), additional flow cytometry controls and analyses (new Figs 3-4, new Supp Fig 1, 4, 9). In addition, a strength is the enhanced correlation with tumor surface phenotype with IC response rather than using PD-1/PD-L1 immunoreactivity. I think the revised manuscript has addressed many of the technical issues.

We would like to thank the reviewer for their positive and thorough evaluation of our revised manuscript, which, in their opinion, has comprehensively addressed the reviewer's concerns and benefited from the review process. Additionally, we are very grateful to the reviewer for highlighting some of the major strengths of our work and for their positive consideration regarding the extended patient data included in the revised version (Figure 7 and Supplementary Fig. 8,9). In reference to the reviewer's comment, we would like to point out that Fig. Rev1 was included in the rebuttal letter to explain how the quantification of cSCC patient samples was performed, but it does not provide additional patient data. However, given the interest of the reviewer for these results, we have included this figure in the revised version of our manuscript (see Supplementary Figure 10).

- Despite this work and the addition data in the revision, the overall conceptual novelty still detracts from the impact of the work. As mentioned by other reviewers well differentiated SCC with a prominent epithelial component is expected to respond differently from poorly differentiated pre-clinical and clinical studies (i.e. Chu Mol Cancer 2023, Ji et al Cell 2020).

After evaluating the reviewer's concern about the "*overall conceptual novelty*" of our study compared with previous preclinical and clinical studies^{1,2}, we would like to highlight why the main original contributions of our study differ from these previous reports. It is true that several preclinical studies have demonstrated the potential benefits of combined anti-PD-1/PD-L1 and anti-TIGIT or anti-CTLA-4 inhibition vs. monotherapy in several cancer types²⁻⁸. However, the mechanisms involved in immune checkpoint inhibitor (ICI) resistance have not yet been clearly identified, emphasizing the need for better characterize the immunosuppression in the TME. In this regard, a significant proportion of patients still remain unresponsive or develop resistance to immune checkpoint blockade (ICB) therapies, and no biomarkers currently precisely predict which patients will benefit from monotherapy vs. combined strategies⁹⁻¹³. For this reason, the strength of our study is the finding that epithelial-mesenchymal cancer cell plasticity is a mechanism of resistance to ICIs in cSCCs. As far as we know, we have identified for the first time that cSCC cancer cells dynamically change the IC ligand repertoire according to their epithelial, hybrid E/M and mesenchymal features, which plays a critical role in inhibiting the antitumor response of effector CD8⁺ T cells and NK cells and defines the response to monotherapy vs. combined ICB therapies. Therefore, our findings reveal that cancer cell heterogeneity should be considered for the selection of ICB therapies to ensure better efficacies

and minimize adverse effects and toxicities in cSCC, HNSCC and melanoma patients. We have added some sentences in the discussion section to further clarify this issue.

On the other hand, Ji et. al. reported the heterogeneity of cancer and stromal cell subpopulations in human cSCCs and demonstrated the expression of other IC ligands besides PD-L1 (B7-H3, LGALS9, C10orf54, TNFRSF14)¹, but no correlation with the epithelial, hybrid E/M or mesenchymal state of cancer cells, nor with the dynamics of CD80 and CD155 ligands, have been described. Furthermore, no information about the potential of these IC ligands as biomarkers to predict response to anti-PD-1/PD-L1 therapy in cSCC, HNSCC and melanoma patients is neither reported in these articles. In this regard, the identification of the biomarkers described in our manuscript could pave the way to stratify patients into monotherapy vs. combined ICI therapies based on cancer cell features. To the best of our knowledge, these findings have not been previously demonstrated in cSCCs or other tumor types. In fact, as a follow up study, we are designing a therapeutic clinical trial to validate our observation in prospective settings.

In conclusion, our study has identified various parameters that can predict responses to immunotherapies, and at the same time, can be therapeutically targeted to enhance the effectiveness of immunotherapies for cSCCs. Importantly, the residence of cSCC cells in alternating epithelial or mesenchymal plastic phenotypic states can also influence their immunomodulatory properties and susceptibilities to ICB therapies.

- The present work uses a viral SCC model and the isolation of tumor cell lines with stable surface phenotypes to convincingly confirm the previous work.

It is important to highlight that our mouse cSCC model was generated from tumors developed in K14-HPV16 mice expressing E6 and E7 oncoproteins from Human papillomavirus (HPV) 16 in basal keratinocytes under the K14 promoter^{14,15}. We showed that some of the altered genes described in the cancer cell signature associated with cSCC progression in our mouse model were also similarly deregulated in unrelated skin PD-SCCs, as unsupervised clustering correctly identified skin SCCs with an EMT-like status in other mouse SCC models¹⁶⁻²¹. Thus, a robust EMT induction occurs during mouse cSCC progression, along with a switch from epithelial to mesenchymal cancer cell features and enhanced metastasis capability. In addition, advanced human cSCCs recapitulate the molecular alterations described in mouse PD/S-SCCs²¹. Taken together, these results validate our mouse model to characterize cancer cell alterations during cSCC progression, to reproduce the cancer cell features found in cSCC patient samples, and to evaluate the response to different therapies.

- Detracting from the novelty and pointed out by reviewers is the lack of consideration that tumors are not ONLY epithelial or mesenchymal but are heterogeneous and evolving.

In agreement with the reviewer, we believe that one of the most important messages from our study is that cSCCs are mostly heterogeneous and that epithelial/mesenchymal states differently modulate the immune system to support immune evasion and therapy resistance. Our findings indicate that a large percentage of patient cSCCs are mixed and are composed of heterogeneous cancer cell populations (epithelial, hybrid/plastic and mesenchymal cancer cells, Fig. 1g-k). Furthermore, our previous and current studies had already demonstrated the ability of cSCC cells to evolve or progress from the epithelial to the mesenchymal state during cSCC progression in response to tumor microenvironment-derived factors (Fig. 1a-f). In this study, we clearly show how these dynamic cancer cell phenotypes impact on ICI response. For that reason, our findings highlight how monitoring these phenotypic states during treatment could contribute to the development of new therapeutic strategies to eliminate resistant cancer cell populations.

- Additional methodology not included in the original manuscript reveals the extensively passaged lines and it remains curious why at least a portion of the tumor did not express heterogeneity. The original and updated human data in Fig 6 supports the authors finding from their mouse data to the extent it was analyzed.

Although we did not include how our mouse cSCC model was generated in the original manuscript, we provided the reference of our previous study that describes it to a large extent²¹. However, for a better understanding and as suggested by the reviewer, we included a more detailed description of this methodology in the revised version of the manuscript. Importantly, cSCC lines are not continuously passaged and, once the model was established, isolated cancer cells were cryopreserved. These epithelial and mesenchymal cancer cells are specifically engrafted into immunocompetent mice when key experiments are needed. Despite several passages, WD-SCCs and full epithelial cancer cells retained their epithelial features, and PD/S-SCCs and full mesenchymal cancer cells conserved their mesenchymal features. Importantly, mixed cSCCs formed of epithelial, hybrid E/M and mesenchymal cancer cells were those that evolve to the mesenchymal state during tumor growth (Fig. 1f). As discussed above, these tumors, similar to what has been observed in patient cSCCs, exhibit strong heterogeneity and not all cancer cells are epithelial or mesenchymal, but may contain plastic cancer cells that may generate new hybrid E/M or mesenchymal cancer cells during tumor growth. Therefore, a combined composition of these cancer cell states can lead to the generation of highly heterogeneous cSCCs, even if some regions generated from full epithelial or mesenchymal cancer cells remain in an epithelial or mesenchymal state, respectively.

Reviewer #3 (Remarks to the Author)

I am satisfied that all my comments have been addressed. I wholeheartedly recommend this study for publication and congratulate the authors on such a well-revised and informative study.

We would like to thank the reviewer for the positive evaluation of the revised manuscript, and for your endorsement of publication of our manuscript in *Nature Communications*. We want to express our gratitude for your review of our manuscript. Your insights have truly enriched our work, and we are thankful for your time and expertise.

As a small note, in my copy the alpha symbols in supplementary figures 4 and 5 appear distorted.

Thank you for spotting this minor typo. We have reviewed the symbols in all figures and we have corrected them in the revised version of the manuscript.

Reviewer #4 (Remarks to the Author)

- The authors responded to my queries insufficiently. Particularly, there are no mechanistic data. Figure S1c, cytotoxic CD8⁺ T cells increased, which could induce PD-L1 expression in cancer cells. But the data are conflicting.

Regarding this concern mentioned by the reviewer, we would like to highlight that although we have observed that the frequency of CD8⁺ T cells increase during mouse cSCC progression (Supplementary Fig. 1c and Supplementary Fig. 3a,b), we also show that the percentage of CD8⁺ T cells co-expressing PD-1 along with LAG-3, TIM-3, CTLA-4 and TIGIT increases during mouse cSCC progression (Supplementary Fig. 1e-h). Given that T-cell exhaustion is associated with increased co-expression of these inhibitory IC receptors²²⁻²⁸, our data indicate that mixed and mesenchymal cSCCs have more exhausted CD8⁺ T cells than epithelial cSCCs. Accordingly, the expression of cytotoxic markers like GzmB and IFN- γ significantly decrease in CD8⁺ T cells during mouse cSCC progression (Supplementary Fig. 1k,l). In this regard, it has been described that the exhausted phenotype of CD8⁺ T cells is also characterized by a gradual loss of effector cytotoxicity as a consequence of reduced production of cytokines such as IFN- γ ²⁹. Importantly, in contrast to the reviewer's comment, CD8⁺ T cells are more exhausted and express less IFN- γ in mixed and mesenchymal mouse cSCCs, which may preclude the IFN- γ -stimulation of PD-L1 expression in cancer cells within these tumors. A little paragraph describing this possible scenario has been included in the discussion of the revised version of the manuscript (see lines 404-408).

It is known that IFN- γ binds to the interferon gamma receptor, leading to phosphorylation of Janus kinase 1 (JAK1) and Janus kinase 2 (JAK2), followed by recruitment of signal transducer and activator of transcription 1 (STAT1). JAKs phosphorylate tyrosine residues of STAT1 promoting its dimerization and translocation to the nucleus. Homodimers of phospho-STAT1 bind to gamma activated sequences (GAS) present in most IFN- γ inducible genes, such as interferon regulatory factor 1 (IRF1). IRF1 directly induces PD-L1 expression by binding to its promoter³⁰⁻³². To determine whether the lower PD-L1 expression detected in hybrid E/M and mesenchymal cancer cells may be mediated by an altered IFN- γ signaling pathway (Fig. 2a), we have *in vitro* treated

full epithelial and mesenchymal cancer cells with recombinant murine IFN- γ for 48h. Western blot analysis reveals the induction of STAT1 phosphorylation after IFN- γ treatment in full epithelial and mesenchymal cancer cells (Fig. Rev1a), indicating that both cancer cells exhibit proper signal transduction of the IFN- γ pathway. In addition, we have analyzed PD-L1 expression after IFN- γ treatment by flow cytometry (Fig. Rev1b,c). Interestingly, we have observed that *in vitro* stimulation with IFN- γ leads to increased PD-L1 expression on the cell surface of mesenchymal cancer cells (Fig. Rev1b,c), indicating that the levels of IFN- γ available *in vivo* from CD8⁺ T and NK cells in mesenchymal cSCCs might be insufficient to induce increased PD-L1 expression on mesenchymal cancer cells. In conclusion, our results indicate that PD-L1 expression during epithelial/mesenchymal cancer cell plasticity could be affected by a reduction of IFN- γ levels in the tumor microenvironment of mouse cSCCs (Fig. 2a). Although all these results are included in a figure for the reviewer (Fig. Rev1), we consider that adding these results in the manuscript may detract attention from the main message of our study. Furthermore, at this stage, we are not able to provide comprehensive mechanistic data explaining why mixed and mesenchymal cSCCs exhibit a higher frequency of CD8⁺ T cells. However, current experiments outside the scope of this manuscript are exploring the identification of mesenchymal cancer cell-derived cytokines that promote the recruitment of CD8⁺ T cells, as well as the regulatory mechanisms controlling the expression of CD80 and CD155 IC ligands. Some sentences have been included in the discussion to highlight the relevance of these future studies, as suggested by the reviewer (see lines 386-390 and 408-410).

Fig. Rev1. Epithelial and mesenchymal cancer cells upregulate PD-L1 upon IFN- γ treatment. a, Western blot image of pSTAT1 induction after IFN- γ treatment in full epithelial and mesenchymal cancer cells. β -actin was used as a loading control. **b,** Percentage (mean \pm SD) of PD-L1⁺ cells and PD-L1 median fluorescence intensity (MFI) in the indicated cancer cell populations after 0 and 10 ng/ml IFN- γ treatment. **c,** Representative PD-L1 MFI gated within full epithelial and mesenchymal cancer cells after 0 and 10 ng/ml IFN- γ treatment. Significant differences determined by unpaired two-sided Student's *t*-test; **p* < 0.05, ***p* < 0.01.

- In addition, very high CD80 expression data could be questionable (Fig. 2d). The authors should analyze gene expression after sorting cancer cells.

To further demonstrate to the reviewer an increase in CD80 expression during E/M cancer cell plasticity (Fig. 2d), we have conducted comprehensive assessments of CD80 expression using various techniques in the different cancer cell populations (full epithelial, EpCAM^{high}, EpCAM^{low}, EpCAM⁻ and full mesenchymal cancer cells). First, we have analyzed the median fluorescence intensity (MFI) of CD80 corresponding to Figure 2d by flow cytometry (Fig. Rev2a; CD80-PE/Cy7, 16-10A1, Biolegend #104733), as well as representative plots showing the percentage of CD80⁺ cancer cells and CD80 MFI gated within EpCAM^{high}, EpCAM^{low} and EpCAM⁻ cancer cells in epithelial, mixed and mesenchymal mouse cSCCs (Fig. Rev2c-e).

Fig. Rev 2

Fig. Rev2. CD80 ligand expression differs according to the epithelial, hybrid E/M or mesenchymal features of cancer cells in mouse cSCCs. a, Percentage (mean ± SD) of CD80⁺ cells and CD80 median

fluorescence intensity (MFI) within full epithelial, EpCAM^{high}, EpCAM^{low}, EpCAM⁻ and full mesenchymal cancer cell populations growing *in vivo*. **b**, mRNA expression levels (mean \pm SD) of *Cd80* in the indicated sorted cancer cells relative to full epithelial cancer cells (n = 3 per group). **c-e**, Gating strategy highlighting the identification of epithelial GFP⁺EpCAM⁺ (including GFP⁺EpCAM^{high} and GFP⁺EpCAM^{low} cells) and mesenchymal GFP⁺EpCAM⁻ cancer cells within the GFP⁺CD45⁻ cancer cell compartment in (**c**) epithelial, (**d**) mixed and (**e**) mesenchymal mouse cSCCs. The percentage of CD80⁺ cancer cells and CD80 MFI is gated within EpCAM^{high}, EpCAM^{low} and EpCAM⁻ cancer cells. Significant differences determined by one-way ANOVA with Tukey's test (**a,b**); **p* < 0.05, ****p* < 0.001, *****p* < 0.0001.

As suggested by the reviewer, we have also evaluated CD80 mRNA expression in sorted cancer cells from epithelial, mixed and mesenchymal cSCCs (Fig. Rev2b) after performing cDNA amplification by pico profiling³³. Both techniques corroborate a higher CD80 expression in hybrid E/M and mesenchymal cancer cells compared with epithelial cancer cells (Fig. Rev2).

Fig. Rev 3

Fig. Rev3. The frequency of CD80⁺ cancer cells increase in mixed and even more so in mesenchymal mouse cSCCs. **a**, Representative immunofluorescence images of GFP⁺ cancer cells (green), CD80⁺ (red), EpCAM⁺ or Vim⁺ (grey) and DAPI nuclear (blue) staining in epithelial, mixed

and mesenchymal mouse cSCCs (n = 3-5 per group). Scale bar, 100 μ m. **b-e**, Quantification (mean \pm SEM) of **(b)** GFP⁺CD80⁺Ep⁻, **(c)** GFP⁺CD80⁺Ep⁺, **(d)** GFP⁺CD80⁺Vim⁻ and **(e)** GFP⁺CD80⁺Vim⁺ cancer cells per tumor area (mm²) in the indicated mouse cSCCs. Each dot indicates the average of at least 10 fields from different tumor regions. **f**, Percentage (mean \pm SEM) of GFP⁺CD80⁻Ep⁻, GFP⁺CD80⁻Ep⁺, GFP⁺CD80⁺Ep⁺ and GFP⁺CD80⁺Ep⁻ cancer cells relative to total cancer cells in the indicated mouse cSCCs. **g**, Percentage (mean \pm SEM) of GFP⁺CD80⁻Vim⁻, GFP⁺CD80⁺Vim⁻, GFP⁺CD80⁺Vim⁺ and GFP⁺CD80⁻Vim⁺ cancer cells relative to total cancer cells in the indicated mouse cSCCs. Significant differences determined by one-way ANOVA with Tukey's test (b-e); *p < 0.05, ****p < 0.0001.

Finally, we have performed immunofluorescence assays to analyze the expression of EpCAM (Ep, epithelial marker), Vimentin (Vim, mesenchymal marker) and CD80 in epithelial, mixed and mesenchymal mouse cSCCs (Fig. Rev3a; mCD80, R&D Systems #AF740). Our results demonstrate that epithelial cSCCs are mainly composed of epithelial GFP⁺CD80⁺Ep⁺Vim⁻ cancer cells and mesenchymal cSCCs are mainly formed of mesenchymal GFP⁺CD80⁺Ep⁻Vim⁺ cancer cells (Fig. Rev3b,e,f,g). The frequency of GFP⁺CD80⁺Ep⁺ cancer cells is specifically increased in mixed cSCCs (Fig. Rev3c), suggesting that the upregulation of this IC ligand might be associated with the enrichment of hybrid/plastic cancer cells. Importantly, mixed and mesenchymal patient cSCCs recapitulate the CD80 alterations described during mouse cSCC progression (Fig. 2f,g,j), which we believe is a robust and definitive validation that the reported CD80 expression dynamics hold true both in cSCC patients and in our murine cSCC model.

Consistent with our results, Miao et al. also demonstrated in a skin cancer model for squamous cell carcinoma that CD80 was induced in the cancer stem cells (SCs) in response to the enhanced TGF- β within the tumor microenvironment³⁴. There, upon engaging CTLA-4, CD80-expressing SCs directly dampen cytotoxic CD8⁺ T cell activity. In this regard, since TGF- β is a well-characterized EMT inducer in several cancer cell types³⁵⁻³⁷, these observations are consistent with the induced expression of CD80 IC ligand on hybrid E/M and mesenchymal cSCC cells in our cSCC model (Fig. 2d). Given that these results support the induced CD80 expression in hybrid E/M and mesenchymal cancer cells shown in Fig. 2d and the constriction of space in the figures, we have not included these new results in the revised version of the manuscript. However, we have included a paragraph in the discussion section to further discuss the induction of CD80 expression in mesenchymal cSCCs (see lines 400-404).

- CD80 DOES NOT induce CTLA-4-mediated immune suppression. CTLA-4 reportedly disturb the CD28 signaling pathways by strongly binding to CD80 or CD86, leading to immunosuppression. Thus, CD28 expression could be very important in this model. If CD80 or CD86 is expressed in cancer cells (i.e., hematological malignancies), these molecules induce CD28 expression in T cells in addition to CTLA-4. However, the authors' data are conflicting.

As far as we know, it has been described that CD80 is mostly considered to be stimulatory and it can work through CD28 to activate CD8⁺ T cells. However, CD80's affinity is higher for CTLA-4 than for CD28, and its interaction with CTLA-4 have an inhibitory effect³⁸⁻⁴¹. Importantly, when cytotoxic T cells receive their activating stimulus in peripheral lymph nodes and then infiltrate a solid tumor, they display markedly elevated CTLA-4³⁹. Thus, within the tumor, the CD80 on cancer cells is more likely to engage CTLA-4 than CD28, thereby dampening the effectiveness of CTLs. Consistent with our results (Fig. 3h, Fig. 5f-j and Supplementary Fig. 6h-n), another study showed that CTLA-4-blocking treatment significantly boost CD8⁺ T cell activities in the presence of CD80⁺ SCCs³⁴. In addition, when CTLs express high levels of CTLA-4, CD80 preferentially engage with CTLA-4 on activated CTLs, but if this interaction is blocked, then CD80 binds to CD28 and becomes stimulatory³⁴. Since Tregs express high levels of CTLA-4 and are potent suppressors of cytotoxic T cells⁴², we suggest that, in mesenchymal cSCCs (Supplementary Fig. 2i,j), they may also participate in the competition for CD80 with CD28⁺CD8⁺ T cells or contribute to CD80 degradation within the recipient cell by trans-endocytosis⁴³, thus favoring T cell exhaustion.

On the other hand, based on previous studies, we suggest that upon T cell activation by Ag-TCR and CD80:CD28 interactions, CTLA-4 expression is induced, but also intracellular CTLA-4 is translocated to the plasma membrane, which rapidly increases CTLA-4 expression at the cell membrane^{44,45}. This scenario induces a reduction in CD8⁺ T cell activity by decreasing the local availability of CD80 for CD28, due to the increased affinity of CTLA-4 for CD80^{38,46}. In addition, an increase in CTLA-4 may lead to a decrease in CD28 expression on CD8⁺ T cells of mesenchymal cSCCs due to CD28 internalization and degradation, as previously reported⁴⁷. It has also been proposed that CTLA-4 may differentially inhibit T cells depending on the strength of the TCR signal they receive. Under conditions of low CD80 expression, up-regulation of small amounts of CTLA-4 during the initial phases of T cell activation could easily suppress the subsequent generation of CD28-mediated stimulatory signals due to its higher avidity for CD80 ligands. Under conditions where CD80 is not limiting (as in mesenchymal cSCCs), TCR and CD28 may be primarily responsible for determining whether a cell enters the cell cycle, while CTLA-4 regulates the extent of subsequent divisions. By more efficiently up-regulating CTLA-4 expression and mobilizing it to the T cell-APC interface, the strongest TCR signals can be preferentially inhibited by CTLA-4⁴⁵.

Altogether, these events could explain a reduction of locally available CD80 for CD28 and the exhaustion of CD8⁺ T cells observed in mesenchymal cSCCs. However, we have not data to compare the simultaneous expression of CD28 and CTLA4 in a specific subset of CD8⁺ T cells in epithelial and mesenchymal cSCCs. We consider that the induced expression of CD80, the

reduced percentage of CD28⁺ T cells and the increased percentage of CTLA-4⁺ T cells observed in mesenchymal cSCCs may be the results of a balance of above-described processes.

1. Ji, A. L. *et al.* Multimodal Analysis of Composition and Spatial Architecture in Human Squamous Cell Carcinoma. *Cell* **182**, 497-514.e22 (2020).
2. Chu, X., Tian, W., Wang, Z., Zhang, J. & Zhou, R. Co-inhibition of TIGIT and PD-1/PD-L1 in Cancer Immunotherapy: Mechanisms and Clinical Trials. *Molecular Cancer* **22**, 93 (2023).
3. Johnston, R. J. *et al.* The Immunoreceptor TIGIT Regulates Antitumor and Antiviral CD8⁺ T Cell Effector Function. *Cancer Cell* **26**, 923–937 (2014).
4. Chauvin, J.-M. *et al.* TIGIT and PD-1 impair tumor antigen-specific CD8⁺ T cells in melanoma patients. *J Clin Invest* **125**, 2046–2058 (2015).
5. He, W. *et al.* CD155/TIGIT Signaling Regulates CD8⁺ T-cell Metabolism and Promotes Tumor Progression in Human Gastric Cancer. *Cancer Research* **77**, 6375–6388 (2017).
6. Hung, A. L. *et al.* TIGIT and PD-1 dual checkpoint blockade enhances antitumor immunity and survival in GBM. *OncoImmunology* **7**, e1466769 (2018).
7. Freed-Pastor, W. A. *et al.* The CD155/TIGIT axis promotes and maintains immune evasion in neoantigen-expressing pancreatic cancer. *Cancer Cell* **39**, 1342-1360.e14 (2021).
8. Banta, K. L. *et al.* Mechanistic convergence of the TIGIT and PD-1 inhibitory pathways necessitates co-blockade to optimize anti-tumor CD8⁺ T cell responses. *Immunity* **55**, 512-526.e9 (2022).
9. Topalian, S. L., Taube, J. M., Anders, R. A. & Pardoll, D. M. Mechanism-driven biomarkers to guide immune checkpoint blockade in cancer therapy. *Nat Rev Cancer* **16**, 275–287 (2016).
10. Wolchok, J. D. *et al.* Overall Survival with Combined Nivolumab and Ipilimumab in Advanced Melanoma. *N Engl J Med* **377**, 1345–1356 (2017).
11. Cho, B. C. *et al.* Tiragolumab plus atezolizumab versus placebo plus atezolizumab as a first-line treatment for PD-L1-selected non-small-cell lung cancer (CITYSCAPE): primary and follow-up analyses of a randomised, double-blind, phase 2 study. *The Lancet Oncology* **23**, 781–792 (2022).
12. Niu, J. *et al.* First-in-human phase 1 study of the anti-TIGIT antibody vibostolimab as monotherapy or with pembrolizumab for advanced solid tumors, including non-small-cell lung cancer☆. *Annals of Oncology* **33**, 169–180 (2022).
13. Rudin, C. M. *et al.* SKYSCRAPER-02: Tiragolumab in Combination With Atezolizumab Plus Chemotherapy in Untreated Extensive-Stage Small-Cell Lung Cancer. *JCO* **42**, 324–335 (2024).
14. Arbeit, J. M., Münger, K., Howley, P. M. & Hanahan, D. Progressive squamous epithelial neoplasia in K14-human papillomavirus type 16 transgenic mice. *J Virol* **68**, 4358–4368 (1994).
15. Coussens, L. M., Hanahan, D. & Arbeit, J. M. Genetic predisposition and parameters of malignant progression in K14-HPV16 transgenic mice. *Am J Pathol* **149**, 1899–1917 (1996).
16. Malanchi, I. *et al.* Cutaneous cancer stem cell maintenance is dependent on β-catenin signalling. *Nature* **452**, 650–653 (2008).
17. Schober, M. & Fuchs, E. Tumor-initiating stem cells of squamous cell carcinomas and their control by TGF-β and integrin/focal adhesion kinase (FAK) signaling. *Proceedings of the National Academy of Sciences* **108**, 10544–10549 (2011).
18. Lapouge, G. *et al.* Skin squamous cell carcinoma propagating cells increase with tumour progression and invasiveness. *EMBO J* **31**, 4563–4575 (2012).
19. White, R. A. *et al.* Epithelial stem cell mutations that promote squamous cell carcinoma metastasis. *J Clin Invest* **123**, 4390–4404 (2013).
20. Wong, C. E. *et al.* Inflammation and Hras signaling control epithelial–mesenchymal transition during skin tumor progression. *Genes Dev.* **27**, 670–682 (2013).
21. Silva-Diz, V. da *et al.* Cancer Stem-like Cells Act via Distinct Signaling Pathways in Promoting Late Stages of Malignant Progression. *Cancer Res* **76**, 1245–1259 (2016).

22. Fourcade, J. *et al.* Upregulation of Tim-3 and PD-1 expression is associated with tumor antigen-specific CD8⁺ T cell dysfunction in melanoma patients. *Journal of Experimental Medicine* **207**, 2175–2186 (2010).
23. Matsuzaki, J. *et al.* Tumor-infiltrating NY-ESO-1-specific CD8⁺ T cells are negatively regulated by LAG-3 and PD-1 in human ovarian cancer. *Proceedings of the National Academy of Sciences* **107**, 7875–7880 (2010).
24. Sakuishi, K. *et al.* Targeting Tim-3 and PD-1 pathways to reverse T cell exhaustion and restore anti-tumor immunity. *Journal of Experimental Medicine* **207**, 2187–2194 (2010).
25. Schnell, A., Bod, L., Madi, A. & Kuchroo, V. K. The yin and yang of co-inhibitory receptors: toward anti-tumor immunity without autoimmunity. *Cell Research* **30**, 285–299 (2020).
26. He, X. & Xu, C. Immune checkpoint signaling and cancer immunotherapy. *Cell Research* **30**, 660–669 (2020).
27. Kubli, S. P., Berger, T., Araujo, D. V., Siu, L. L. & Mak, T. W. Beyond immune checkpoint blockade: emerging immunological strategies. *Nat Rev Drug Discov* **20**, 899–919 (2021).
28. Gu, X. *et al.* Itaconate promotes hepatocellular carcinoma progression by epigenetic induction of CD8⁺ T-cell exhaustion. *Nat Commun* **14**, 8154 (2023).
29. Wherry, E. J. T cell exhaustion. *Nature Immunology* **12**, 492–499 (2011).
30. Dong, H. *et al.* Tumor-associated B7-H1 promotes T-cell apoptosis: A potential mechanism of immune evasion. *Nat Med* **8**, 793–800 (2002).
31. Lee, S.-J. *et al.* Interferon regulatory factor-1 is prerequisite to the constitutive expression and IFN- γ -induced upregulation of B7-H1 (CD274). *FEBS Letters* **580**, 755–762 (2006).
32. Garcia-Diaz, A. *et al.* Interferon Receptor Signaling Pathways Regulating PD-L1 and PD-L2 Expression. *Cell Reports* **19**, 1189–1201 (2017).
33. Gonzalez-Roca, E. *et al.* Accurate Expression Profiling of Very Small Cell Populations. *PLOS ONE* **5**, e14418 (2010).
34. Miao, Y. *et al.* Adaptive Immune Resistance Emerges from Tumor-Initiating Stem Cells. *Cell* **177**, 1172–1186.e14 (2019).
35. Massagué, J. TGF β in Cancer. *Cell* **134**, 215–230 (2008).
36. Scheel, C. *et al.* Paracrine and Autocrine Signals Induce and Maintain Mesenchymal and Stem Cell States in the Breast. *Cell* **145**, 926–940 (2011).
37. Moustakas, A. & Heldin, C.-H. Induction of epithelial–mesenchymal transition by transforming growth factor β . *Seminars in Cancer Biology* **22**, 446–454 (2012).
38. van der Merwe, P. A., Bodian, D. L., Daenke, S., Linsley, P. & Davis, S. J. CD80 (B7-1) Binds Both CD28 and CTLA-4 with a Low Affinity and Very Fast Kinetics. *Journal of Experimental Medicine* **185**, 393–404 (1997).
39. Alegre, M.-L., Frauwirth, K. A. & Thompson, C. B. T-cell regulation by CD28 and CTLA-4. *Nat Rev Immunol* **1**, 220–228 (2001).
40. Walker, L. S. K. & Sansom, D. M. The emerging role of CTLA4 as a cell-extrinsic regulator of T cell responses. *Nat Rev Immunol* **11**, 852–863 (2011).
41. Matheu, M. P. *et al.* Imaging regulatory T cell dynamics and CTLA4-mediated suppression of T cell priming. *Nat Commun* **6**, 6219 (2015).
42. Zhang, R. *et al.* An obligate cell-intrinsic function for CD28 in Tregs. *J Clin Invest* **123**, 580–593 (2013).
43. Qureshi, O. S. *et al.* Trans-Endocytosis of CD80 and CD86: A Molecular Basis for the Cell-Extrinsic Function of CTLA-4. *Science* **332**, 600–603 (2011).
44. Linsley, P. S. *et al.* Intracellular Trafficking of CTLA-4 and Focal Localization Towards Sites of TCR Engagement. *Immunity* **4**, 535–543 (1996).
45. Egen, J. G. & Allison, J. P. Cytotoxic T Lymphocyte Antigen-4 Accumulation in the Immunological Synapse Is Regulated by TCR Signal Strength. *Immunity* **16**, 23–35 (2002).
46. Thompson, C. B. & Allison, J. P. The Emerging Role of CTLA-4 as an Immune Attenuator. *Immunity* **7**, 445–450 (1997).
47. Berg, M. & Zavazava, N. Regulation of CD28 expression on CD8⁺ T cells by CTLA-4. *Journal of Leukocyte Biology* **83**, 853–863 (2008).